# Dueling Over Dessert, Mastering the Art of Repeated Cake Cutting

**Simina Brânzei**
Purdue University
simina.branzei@gmail.com

**MohammadTaghi Hajiaghayi**
University of Maryland
hajiaghayi@gmail.com

**Reed Phillips**
Purdue University
phill289@purdue.edu

**Suho Shin**
University of Maryland
suhoshin@umd.edu

**Kun Wang**
Purdue University
wang5675@purdue.edu

## Abstract

We consider the setting of repeated fair division between two players, denoted Alice and Bob, with private valuations over a cake. In each round, a new cake arrives, which is identical to the ones in previous rounds. Alice cuts the cake at a point of her choice, while Bob chooses the left piece or the right piece, leaving the remainder for Alice. We consider two versions: *sequential*, where Bob observes Alice's cut point before choosing left/right, and *simultaneous*, where he only observes her cut point after making his choice. The simultaneous version was first considered in Aumann and Maschler (1995).

We observe that if Bob is almost myopic and chooses his favorite piece too often, then he can be systematically exploited by Alice through a strategy akin to a binary search. This strategy allows Alice to approximate Bob's preferences with increasing precision, thereby securing a disproportionate share of the resource over time.

We analyze the limits of how much a player can exploit the other one and show that fair utility profiles are in fact achievable. Specifically, the players can enforce the equitable utility profile of $(1/2, 1/2)$ in the limit on every trajectory of play, by keeping the other player's utility to approximately $1/2$ on average while guaranteeing they themselves get at least approximately $1/2$ on average. We show this theorem using a connection with Blackwell approachability.

Finally, we analyze a natural dynamic known as fictitious play, where players best respond to the empirical distribution of the other player. We show that fictitious play converges to the equitable utility profile of $(1/2, 1/2)$ at a rate of $O(1/\sqrt{T})$.

# 1 Introduction

Cake cutting is a model of fair division Steinhaus (1948), where the cake is a metaphor for a heterogeneous divisible resource such as land, time, memory in shared computing systems, clean water, greenhouse gas emissions, fossil fuels, or other natural deposits (Procaccia (2013)). The problem is to divide the resource among multiple participants so that everyone believes the allocation is fair. There is an extensive literature on cake cutting in mathematics, political science, economics (Robertson and Webb (1998); Brams and Taylor (1996); Moulin (2003)) and computer science (Brandt et al. (2016)), with a number of protocols implemented (Goldman and Procaccia (2014)).

Traditional approaches to cake cutting often consider single instances of division. However, many real-world scenarios require a repeated division of resources. For instance, consider the recurring task of allocating classroom space in educational institutions each quarter or that of repeatedly dividing computational resources (such as CPU and memory) among the members of an organization. These settings reflect the reality of many social and economic interactions, necessitating a model that not only addresses the fairness of a single division, but also the dynamics and strategies that emerge among participants over repeated interactions.

Repeated fair division is a classic problem first considered by Aumann and Maschler (1995), where two players—denoted Alice and Bob—have private valuations over the cake and interact in the following environment. Every day a new cake arrives, which is the same as the ones in previous days. Alice cuts the cake at a point of her choosing, while Bob chooses either the left piece or the right piece, leaving the remainder to Alice. Aumann and Maschler (1995) considered the simultaneous setting, where both players take their actions at the same time each day, and analyzed the payoffs achievable by Bob when he can have one of two types of valuations.

In this paper, we provide the first substantial progress in this classic setting. We further analyze the simultaneous version from Aumann and Maschler (1995) and also go beyond it, by considering the sequential version where Bob has the advantage of observing Alice's chosen cut point before making his selection. Tactical considerations remain pivotal in the sequential version, which is none other than the repeated *Cut-and-choose* protocol with strategic players.

A key observation in our study is the strategic vulnerability inherent in repeated Cut-and-choose. At a high level, if Bob consistently chooses his preferred piece, then he can be systematically exploited by Alice through a strategy akin to a binary search. This strategy allows Alice to approximate Bob's preferences with increasing precision, thereby securing a disproportionate share of the resource over time. To fight back Alice's attempt to exploit him, Bob could deceive her by being unpredictable, thus hiding his preferences. While this behavior has the potential to reduce Alice's share of the cake, it could also come at the price of affecting Bob's own payoff guarantees in the long term.

Our analysis of the repeated cake cutting game formalizes the intuition that Alice can exploit a (nearly) myopic Bob that often chooses his favorite piece. This outcome, where Alice gains more value, is not entirely fair, as it leaves her happier than Bob. The fairness notion of *equitability* addresses this imbalance, embodying the idea that players should be equally happy. Formally, it requires that Alice's value for her allocation should equal Bob's value for his allocation. Achieving equitability is particularly important in scenarios with potential for conflict, such as splitting an inheritance.

We show that achieving equitable outcomes in the repeated interaction is in fact possible. Specifically, each player has a strategy that guarantees the other player receives no more than approximately $1/2$ on average, while securing at least approximately $1/2$ for themselves. This approaches the equitable utility profile of $(1/2, 1/2)$ in the limit. We obtain this result by using a connection with Blackwell approachability (1956). Moreover, we consider a natural dynamic known as fictitious play (Brown (1951)), where players best respond to the empirical frequency of the other player's past actions. We show that fictitious play converges to the equitable utility profile of $(1/2, 1/2)$ at a rate of $O(1/\sqrt{T})$.

## 1.1 Model

**Cake cutting model for two players.** The cake is modelled as the interval $[0, 1]$. There are players $N = \{A, B\}$, where $A$ stands for Alice and $B$ for Bob. Each player $i$ has a private value density function $v_i : [0, 1] \to \mathbb{R}_+$. A *piece* of cake is a measurable set $S \subseteq [0, 1]$. The value of player $i$ for $S$ is $V_i(S) = \int_{x \in S} v_i(x) \; dx$. Atoms are worth zero and the valuations are normalized so that

$V_i([0,1]) = 1 \ \forall i \in [n]$. We require bounded densities, i.e. there exist fixed arbitrary $\delta, \Delta > 0$ such that $\delta \leq v_i(x) \leq \Delta$ for all $x \in [0,1]$.

An allocation $Z = (Z_A, Z_B)$ is a partition of the cake among the players such that each player $i$ receives piece $Z_i$, the pieces are disjoint, and $\bigcup_{i \in N} Z_i = [0,1]$. The valuation (aka utility or payoff) of player $i$ at an allocation $Z$ is $V_i(Z_i)$. An allocation $Z$ is *equitable* if the players are equally happy with their pieces, meaning $V_A(Z_A) = V_B(Z_B)$.

Let $m_A$ be Alice's midpoint of the cake and $m_B$ Bob's midpoint. Alice's *Stackelberg value*, denoted $u_A^*$, is the utility she receives when she cuts the cake at $m_B$ and Bob chooses his favorite piece, breaking ties in Alice's favor. The midpoints and Alice's Stackelberg value are depicted in Figure 1.

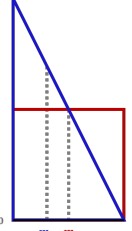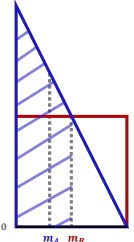

Figure 1: Densities for Alice (blue) and Bob (red). Figure (a) shows Alice's midpoint $m_A$ and Bob's midpoint $m_B$. The shaded area in Figure (b) is Alice's Stackelberg value.

**Repeated cake cutting.** Each round $t = 1, 2, \ldots, T$, the next steps take place:

- A new cake arrives, which is identical to the ones in previous rounds.
- Alice cuts the cake at a point $a_t \in [0,1]$ of her choice. Bob chooses either the left piece or the right piece, then Alice takes the remainder.

We consider two versions: *sequential*, where Bob observes Alice's cut point $a_t$ before choosing left/right, and *simultaneous*, where he only observes her cut point after making his choice.

A pure strategy is a map from the history observed by a player to the next action to play. A mixed strategy is a probability distribution over pure strategies.

## 1.2 Our Results

Our results will examine how players fare in the repeated game over $T$ rounds. Given a history $H$, Alice's Stackelberg regret is $\text{Reg}_A(H) = \sum_{t=1}^{T} [u_A^* - u_A^t(H)]$, where $u_A^*$ is Alice's Stackelberg value and $u_A^t(H)$ is Alice's utility in round $t$ under history $H$.

Suppose Alice uses a mixed strategy $S_A$ and Bob uses a mixed strategy $S_B$. Then $S_A$ ensures Alice's Stackelberg regret is at most $\gamma$ against $S_B$ if $\text{Reg}_A(H) \leq \gamma$ for all $T$-round histories $H$ that could have arisen under the strategies $(S_A, S_B)$. Precise definitions for strategies/regret and the remaining notation needed for the full proofs can be found in Section 3.

**Alice exploiting Bob**

We start with an observation about the sequential setting. If Bob chooses his favorite piece in each round, then Alice can exploit him by running binary search until identifying his midpoint within a small error and then cutting near it for the rest of time. This will lead to Alice getting essentially her Stackelberg value in all but $O(\log T)$ rounds, while Bob will get $1/2$ in all but $O(\log T)$ rounds.

**Proposition 1.** *If Bob plays myopically in the sequential setting, then Alice has a strategy that ensures her Stackelberg regret is $O(\log T)$.*

This exploitation phenomenon holds more generally: if Bob's strategy has bounded regret with respect to the standard of selecting his preferred piece in every round in hindsight, then Alice can almost get her Stackelberg value in each round. Her Stackelberg regret is a function of Bob's regret guarantee, as quantified in the next theorem.

**Theorem 1** (Exploiting a nearly myopic Bob)**.** *Let $\alpha \in [0, 1)$. Suppose Bob plays a strategy that ensures his regret is $O(T^\alpha)$ in the sequential setting. Let $\mathcal{B}^\alpha$ denote the set of all such Bob strategies.*

- *If Alice knows $\alpha$, she has a strategy $S_A = S_A(\alpha)$ that ensures her Stackelberg regret is $O(T^{\frac{\alpha+1}{2}} \log T)$. The exponent is sharp: Alice's Stackelberg regret is $\Omega(T^{\frac{\alpha+1}{2}})$ for some Bob strategy in $\mathcal{B}^\alpha$.*

- *If Alice does not know $\alpha$, she has a strategy $S_A$ that ensures her Stackelberg regret is $O\left(\frac{T}{\log T}\right)$. The exponent is sharp: if $S_A$ guarantees Alice Stackelberg regret $O(T^\beta)$ against all Bob strategies in $\mathcal{B}^\alpha$ for some $\beta \in [0, 1)$, then $S_A$ has Stackelberg regret $\Omega(T)$ for some Bob strategy in $\mathcal{B}^\beta$.*

In contrast, in the simultaneous setting, Alice may not approach her Stackelberg value on *every* trajectory of play. In order to get her Stackelberg value in any given round, Alice needs to cut near Bob's midpoint and Bob needs to pick the piece he prefers, say $R$. However, if Bob deterministically commits to picking $R$, he will be completely exploited by an Alice who cuts at 1, breaking any reasonable regret guarantee he might have. Indeed, any Bob with a deterministic strategy (possibly using different actions over the rounds) has a corresponding Alice who can completely exploit him. Therefore, any Bob strategy with a good regret guarantee would behave randomly, making it impossible for Alice to reliably get her Stackelberg value on every trajectory. For this reason, we focus on the sequential setting when studying how Alice can exploit Bob.[1]

**Equitable payoffs.**

Motivated by Theorem 1, we examine the general limits of how much each player can exploit the other and whether fair outcomes are achievable, in both the sequential and simultaneous settings.

Given a history $H$, player $i$ is said to get an average payoff of $\gamma$ if $\left(\frac{1}{T}\right) \sum_{t=1}^{T} u_i^t(H) = \gamma$, where the left hand side is not expected utility, but rather the observed total utility averaged over $T$ rounds.

We say a utility profile $(u_A, u_B)$ is *equitable* if $u_A = u_B$. In the single round setting, $u_A$ and $u_B$ will naturally represent the utilities of the players at an allocation. In the repeated setting, $u_A$ and $u_B$ will represent the time-average utilities of the players.

The next theorems show that each player can keep the other player at $1/2$ while guaranteeing $1/2$ for themselves. This type of behavior is reminiscent of spiteful bidding in auctions (Tang and Sandholm (2012)), where a buyer's utility diminishes if other bidders are too satisfied.

**Theorem 2** (Alice enforcing equitable payoffs; informal)**.** *In both the sequential and simultaneous settings, Alice has a pure strategy $S_A$, such that for every Bob strategy $S_B$:*

- *on every trajectory of play, Alice's average payoff is at least $1/2 - o(1)$, while Bob's average payoff is at most $1/2 + o(1)$. More precisely, $\frac{u_A}{T} \geq \frac{1}{2} - \Theta\left(\frac{1}{\sqrt{T}}\right)$ and $\frac{u_B}{T} \leq \frac{1}{2} + \Theta\left(\frac{1}{\ln T}\right)$, where $u_i$ is the cumulative payoff of player $i$ over the time horizon $T$.*

A key ingredient in the proof of Theorem 2 is a connection with Blackwell's approachability theorem Blackwell (1956). Generally speaking, Blackwell approachability can be used by a player to limit the payoff of the other player in a certain region of the utility profile. However, the main challenge is that there are uncountably many types of Bob and so Alice cannot apply the strategy from Blackwell directly. Instead, Alice's strategy constructs a countably infinite set of representatives, which allows us to adapt Blackwell's argument to this setting.

We show a symmetric theorem for Bob in the sequential setting, while in the simultaneous setting Bob's guarantee only holds in expectation.

**Theorem 3** (Bob enforcing equitable payoffs; informal)**.**

- In the sequential setting: *Bob has a pure strategy $S_B$, such that for every Alice strategy $S_A$, on every trajectory of play, Bob's average payoff is at least $1/2 - o(1)$, while Alice's average payoff is at most $1/2 + o(1)$. More precisely, $\frac{u_B}{T} \geq \frac{1}{2} - \frac{1}{\sqrt{T}}$ and $\frac{u_A}{T} \leq \frac{1}{2} + \Theta\left(\frac{1}{\sqrt{T}}\right)$.*

---

[1]One may explore a weaker regret benchmark for Bob of always picking the better of left or right in hindsight, which we believe would be an interesting direction for future work.

- In the simultaneous setting: *Bob has a mixed strategy $S_B$, such that for every Alice strategy $S_A$, both players have average payoff $1/2$ in expectation.*

When Alice and Bob play such strategies against each other, they approach an equitable utility profile of $(1/2, 1/2)$. If just one player follows such a strategy, then the best the other player can do is to ensure the safety value of $1/2$ for themselves, thus achieving the utility profile $(1/2, 1/2)$.

**Fictitious play**

Fictitious play is a classic learning rule where at each round, each player best responds to the empirical frequency of play of the other player. Fictitious play was introduced in Brown (1951). Convergence to Nash equilibria has been shown for zero-sum games (Robinson (1951)) and special cases of general-sum games (Nachbar (1990); Monderer and Shapley (1996b,a)).

In the cake cutting model, learning rules such as fictitious play are more meaningful in the simultaneous setting, where there is uncertainty for both players due to the simultaneous actions. The precise definition of the fictitious play dynamic is in Section 6, while an example of trajectories for an instance with random valuations and uniform random tie-breaking can be found in Figure 2.

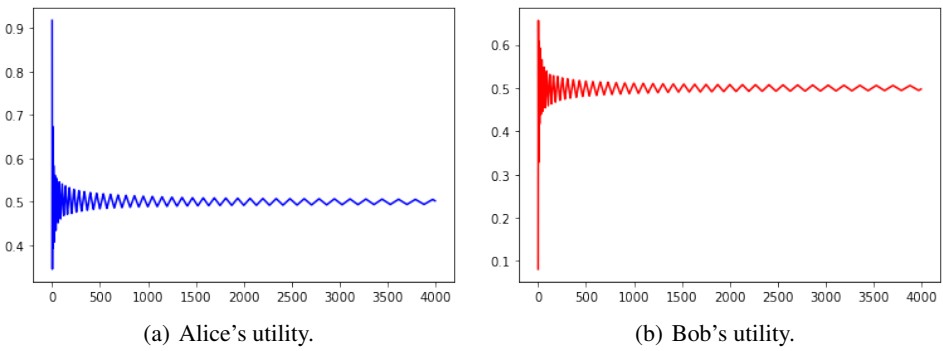

(a) Alice's utility.      (b) Bob's utility.

Figure 2: Illustration of Alice's and Bob's average payoff in a randomly generated instance of valuations. The X axis shows the time and the Y axis shows the average payoff up to that round.

The convergence properties of fictitious play can be characterized as follows.

**Theorem 4** (Fictitious Play; informal). *When both Alice and Bob run fictitious play, the average payoff of each player converges to $1/2$ at a rate of $O(1/\sqrt{T})$.*

**Roadmap to the paper.** Related work is in Section 2. Formal notation and preliminaries can be found in Section 3. An overview of how Alice can exploit a nearly myopic Bob can be found in Section 4, with formal proofs in Appendix A. An overview of how players can enforce equitable payoffs can be found in Section 5, with formal proofs in Appendix B. Fictitious play can be found in Section 6, with formal proofs in Appendix C. Concluding remarks can be found in Section 7.

## 2 Related Work

**Cake cutting and fairness notions.** The cake cutting model is due to Steinhaus (1948). Standard fairness notions include proportionality, equitability, envy-freeness (Even and Paz (1984); Dubins and Spanier (1961); Edward Su (1999); Stromquist (1980); Alon (1987)). For surveys, see Robertson and Webb (1998); Brams and Taylor (1996); Moulin (2003); Brandt et al. (2016); Procaccia (2013).

In the Robertson-Webb (RW) query model for cake cutting (Woeginger and Sgall (2007)), a mediator asks the players enough queries about their preferences until it can output a fair allocation. For studies on the query complexity of cake cutting, see Even and Paz (1984); Woeginger and Sgall (2007); Edmonds and Pruhs (2006); Procaccia (2009); Aziz and Mackenzie (2016); Amanatidis et al. (2018); Cheze (2020); Stromquist (2008); Deng et al. (2012); Goldberg et al. (2020a); Filos-Ratsikas et al. (2022); Segal-Halevi (2018); Deligkas et al. (2021); Goldberg et al. (2020b); Brânzei and Nisan (2022, 2019); Filos-Ratsikas et al. (2020); Alon and Graur (2020); Filos-Ratsikas et al. (2021).

Mossel and Tamuz (2010); Branzei and Miltersen (2015) studied truthful cake cutting in the RW query model, and Chen et al. (2013); Bu et al. (2023); Bei et al. (2022); Tao (2022) in the direct revelation model. The equilibria of cake cutting protocols were considered in Nicolò and Yu (2008); Brânzei and Miltersen (2013); Brânzei et al. (2016); Goldberg and Iaru (2021).

**Multiple divisible/indivisible goods and chores.** The algorithms and complexity of finding fair allocations in settings with multiple divisible/indivisible goods/bads were considered in Oh et al. (2021); Plaut and Roughgarden (2020, 2019); Manurangsi and Suksompong (2021); Chaudhury et al. (2021c); Bilò et al. (2019); Amanatidis et al. (2022); Procaccia (2020); Chaudhury et al. (2020, 2021b); Procaccia and Wang (2014); Kulkarni et al. (2021); Chaudhury et al. (2021a). Tucker-Foltz and Zeckhauser (2023) analyze how the cutter should act in a single-round cut-and-choose on multiple goods where the players' valuations of the goods are drawn from a publicly known distribution.

Ghodsi et al. (2011) studied fairness in cloud computing settings, where there are multiple divisible goods (e.g. CPU and memory) and the users have to run jobs with different resource requirements. Kandasamy et al. (2020) studied players who do not know their own resource requirements.

**Dynamic fair division.** Closest to our setting is the analysis in the book of Aumann and Maschler (1995) (page 243), where two players are dividing a cake with a cherry. Alice (the cutter) has a uniform density and so she does not care for the cherry, while Bob (the chooser) may or may not like the cherry. Alice and Bob declare their actions simultaneously and Alice is only allowed to cut in one of two locations. Additionally, Alice has a prior over the type of Bob she is facing. Aumann and Maschler (1995) analyzes the set of payoffs approachable for Bob using Blackwell approachability. In contrast, we allow arbitrary value densities for the players and do not assume priors and also consider the sequential version of the game.

Tamuz et al. (2018) introduced exploitability in repeated cut-and-choose protocols, with some cuts made by a mediator, designing non-exploitable protocols. Online cake cutting was studied by Walsh (2011), where agents can arrive/depart over time. For dynamic fair division where goods are allocated irrevocably upon arrival, see Kash et al. (2014); Friedman et al. (2015); Benadè et al. (2022).

**Learning in repeated Stackelberg games.** The Stackelberg game was introduced by Stackelberg (1934) to understand the first mover advantage of firms when entering a market. The Stackelberg equilibrium concept has important applications such as security games Tambe (2011); Balcan et al. (2015), online strategic classification Dong et al. (2018), and online principal agent problems Hajiaghayi et al. (2023). Our model can be seen as each player facing an online learning version of a repeated Stackelberg game.

Kleinberg and Leighton (2003) considered a seller's problem of designing an efficient repeated posted price mechanism to buy identical goods when it interacts with a sequence of myopic buyers. Gan et al. (2019); Birmpas et al. (2020); Zhao et al. (2023) considered a repeated Stackelberg game to study how the follower or leader can exploit the opponent in a general game with arbitrary payoffs. Their techniques, however, do not apply to our model as they typically consider the setting of one player knowing the entire payoff matrix trying to deceive the other player given behavioral assumptions.

**Exploiting no-regret agents.** Several works considered the extent to which one player can exploit the knowledge that the other player has a strategy with sublinear regret. The goal is often to approach the Stackelberg value, the maximum payoff that the exploiter could get by selecting an action first and allowing the opponent to best-respond. In simultaneous games, Deng et al. (2019) showed the exploiter can approach their Stackelberg value, assuming knowledge of the other player's payoff function. Haghtalab et al. (2022) showed that, for certain types of sequential games, an exploiting leader can approach their Stackelberg value in the limit. Theorem 1 is a similar statement in our setting, but we bound the exploited agent's behavior with an explicit regret guarantee rather than using discounted future payoffs; also, our setting is not captured by the types of games they consider.

**Fictitious play.** Fictitious play was introduced in Brown (1951). Convergence to Nash equilibria has been shown for zero-sum games (Robinson (1951)) and special cases of general-sum games (Nachbar (1990); Monderer and Shapley (1996b,a)). None of these results directly apply to our setting, but the most relevant is Berger (2005), which covered non-degenerate $2 \times n$ games (i.e. where every action has a unique best response). Our "$2 \times \infty$" game is degenerate, as Bob does not have a unique best response to Alice cutting at $m_B$. Few existing works apply fictitious play to games with continuous action spaces. An example is Perkins and Leslie (2014), which showed that stochastic fictitious play does converge in two-player zero-sum games with continuous action spaces.

Karlin (1959) conjectured that fictitious play converges at a rate of $O(T^{-1/2})$. Brandt et al. (2013) found small games where the convergence rate is $O(T^{-1/2})$, but with very large constants in the $O()$. Daskalakis and Pan (2014) disproved Karlin's conjecture, showing there exist games with $n \times n$ payoff matrices in which convergence takes place at a rate of $\Omega(T^{-1/n})$ using adversarial tie-breaking rules. Panageas et al. (2023) found examples of even slower convergence.

Harris (1998) showed that fictitious play converges at a rate of $O(T^{-1})$ in $2 \times 2$ zero-sum games. Abernethy et al. (2021) considered diagonal payoff matrices with non-adversarial tie-breaking rules, showing convergence rates of $O(T^{-1/2})$. Abernethy et al. (2021) does not give a rate of convergence in our setting because requiring the payoff matrix to be diagonal would correspond to Alice only being allowed to cut at 0 or 1. This assumption is not as natural in our setting. In fact, if Alice can only cut at 0 or 1 the game becomes zero-sum. Furthermore, we allow arbitrary tie-breaking rules.

## 3 Preliminaries

In this section we formally define the notation used in our proofs. All our notation applies to both the sequential and simultaneous settings, unless otherwise stated.

**History.** Recall $T$ is the number of rounds. For each round $t \in [T]$,

- let $a_t \in [0, 1]$ be Alice's cut at time $t$ and $b_t \in \{L, R\}$ be Bob's choice at time $t$, where $L$ stands for the left piece $[0, a_t]$ and $R$ for the right piece $[a_t, 1]$.
- let $A_t = (a_1, \ldots, a_t)$ be the history of cuts until the end of round $t$ and $B_t = (b_1, \ldots, b_t)$ the history of choices made by Bob until the end of round $t$.

A history $H = (A_T, B_T)$ denotes an entire trajectory of play.

**Strategies.** Let $\mathcal{P}$ be the space of integrable value densities over $[0, 1]$. A pure strategy for Alice at time $t$ is a function $S_A^t : [0, 1]^{t-1} \times \{L, R\}^{t-1} \times \mathcal{P} \times \mathbb{N} \to [0, 1]$, such that $S_A^t(A_{t-1}, B_{t-1}, v_A, T)$ is the next cut point made by Alice as a function of the history $A_{t-1}$ of Alice's cuts, the history $B_{t-1}$ of Bob's choices, Alice's valuation $v_A$, and the horizon $T$.

For Bob, we define pure strategies separately for the sequential and simultaneous settings due to the different feedback that he gets:

*Sequential:* A pure Bob strategy at time $t$ is a map $S_B^t : [0, 1]^t \times \{L, R\}^{t-1} \times \mathcal{P} \times \mathbb{N} \to \{L, R\}$. That is, Bob observes Alice's cut point and then responds.

*Simultaneous:* A pure Bob strategy at time $t$ is a map $S_B^t : [0, 1]^{t-1} \times \{L, R\}^{t-1} \times \mathcal{P} \times \mathbb{N} \to \{L, R\}$. Thus here Bob chooses $L/R$ before observing Alice's cut point at time $t$.

A pure strategy for Alice over the entire time horizon $T$ is denoted $S_A = (S_A^1, \ldots, S_A^T)$ and tells Alice what cut to make at each time $t$. A pure strategy for Bob over the entire time horizon $T$ is denoted $S_B = (S_B^1, \ldots, S_B^T)$ and tells Bob whether to play $L/R$ at each time $t$.

A mixed strategy is a probability distribution over the set of pure strategies. [2]

**Rewards and utilities.** Suppose Alice has mixed strategy $S_A$ and Bob has mixed strategy $S_B$. Let $u_A^t$ and $u_B^t$ be the random variables for the utility (payoff) experienced by Alice and Bob, respectively, at round $t$. The utility of player $i \in \{A, B\}$ is denoted $u_i = u_i(S_A, S_B) = \sum_{t=1}^T u_i^t$. The utility of player $i$ from round $t_1$ to $t_2$ is $u_i(t_1, t_2) = \sum_{t=t_1}^{t_2} u_i^t$.

The expected utility of player $i$ is $\mathbb{E}[u_i] = \sum_{t=1}^T \mathbb{E}[u_i^t]$, where the expectation is taken over the randomness of the strategies $S_A$ and $S_B$.

Given a history $H$, let $u_i^t(H)$ be player $i$'s utility in round $t$ under $H$ and let $u_i(H) = \sum_{t=1}^T u_i^t(H)$ be player $i$'s cumulative utility under $H$.

---

[2] In fact, this is equivalent to the behavior strategy in which the player assigns a probability distribution given a history, thanks to Kuhn's theorem Kuhn (1950); Kuhn[1] (1953). The original version of Kuhn's theorem is restricted to games with finite action space, but can be extended to any action space that is isomorphic to unit interval by Aumann (1961); Dresher et al. (2016), which contains our setting.

**Midpoints and Stackelberg value.** Let $m_A \in [0,1]$ be Alice's midpoint of the cake, with $V_A([0, m_A]) = 1/2$, and $m_B \in [0,1]$ be Bob's midpoint, with $V_B([0, m_B]) = 1/2$. Since the densities are bounded from below, the midpoint of each player is uniquely defined. Alice's *Stackelberg value*, denoted $u_A^*$, is the utility Alice gets when she cuts at $m_B$ and Bob chooses his favorite piece breaking ties in favor of Alice (i.e. taking the piece she prefers less).

## 4 Alice exploiting Bob

In this section we give an overview of Theorem 1, which considers the sequential setting and quantifies the extent to which Alice can exploit a Bob that has sub-linear regret with respect to the benchmark of choosing the best piece in each round. Formal proofs for this section are in Appendix A.

We start by defining the notion of Stackelberg regret (Dong et al. (2018); Haghtalab et al. (2022)).

**Definition 1** (Stackelberg regret). *Given a history $H$ of the places Alice cut and the pieces Bob chose in each round, Alice's Stackelberg regret is $Reg_A(H) = \sum_{t=1}^{T} [u_A^* - u_A^t(H)]$, where $u_A^*$ is Alice's Stackelberg value and $u_A^t(H)$ is Alice's utility in round $t$ under history $H$.*

For Bob, we consider the basic notion of static regret, where Bob compares his payoff to what would have happened if Alice's actions remained the same but he chose the best piece in each round.

**Definition 2** (Regret). *Given a history $H$, Bob's regret is*

$$Reg_B(H) = \sum_{t=1}^{T} \Big[ \max\Big\{ V_B([0, a_t]), V_B([a_t, 1]) \Big\} - u_B^t(H) \Big],$$

*recalling that $u_B^t(H)$ is Bob's utility in round $t$ under history $H$.*

Next we provide a proof sketch for Theorem 1, which is divided in the next two propositions, corresponding to the cases where Alice knows $\alpha$ and does not know $\alpha$.

**Proposition 2.** *Let $\alpha \in [0, 1)$. Suppose Bob plays a strategy that ensures his regret is $O(T^\alpha)$ and let $\mathcal{B}^\alpha$ denote the set of all such Bob strategies. Assume Alice knows $\alpha$. Then she has a strategy $S_A = S_A(\alpha)$ that ensures her Stackelberg regret is $O\big(T^{\frac{\alpha+1}{2}} \log T\big)$. The exponent is sharp: Alice's Stackelberg regret is $\Omega\big(T^{\frac{\alpha+1}{2}}\big)$ for some Bob strategy in $\mathcal{B}^\alpha$.*

*Proof sketch.* We sketch both the upper and lower bounds.

**Sketch for the upper bound.** Let $S_B$ denote Bob's strategy, which guarantees his regret is $O(T^\alpha)$. Suppose Alice knows $\alpha$. Then Alice initializes an interval $I = [0, 1]$ and uses the following strategy.

Iteratively, for $i = 0, 1, \dots,$:

(1) Alice discretizes the interval $I = [u, w]$ in a constant number of sub-intervals (set to 6) of equal value to her, by cutting at points $a_{i,j}$ for $j \in [5]$ such that $u < a_{i,1} < a_{i,2} < \dots < a_{i,5} < w$. Denote $a_{i,0} = u$ and $a_{i,6} = w$. An illustration is in Figure 3.

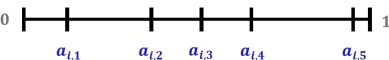

Figure 3: Illustration of step (1) for $i = 0$. Alice divides the interval $[0, 1]$ in 6 disjoint intervals of equal value to her, demarcated by points $a_{0,0} = 0 < a_{0,1} < a_{0,2} < a_{0,3} < a_{0,4} < a_{0,5} < 1 = a_{0,6}$.

(2) Alice selects a number $\eta$, which will be set "large enough" as a function of $T$ and $\alpha$. In the next $5\eta$ rounds, Alice cuts an equal number of times at each point $a_{i,j}$ for $j \in [5]$. That is:

- In each of the next $\eta$ rounds, Alice cuts at $a_{i,1}$ and observes Bob's choices there, computing the majority answer as $c_{i,1} = L$ if Bob picked the left piece more times than the right piece, and $c_{i,1} = R$ otherwise. The next $\eta$ rounds after that Alice switches to cutting at $a_{i,2}$, and so on.

In this fashion, Alice computes $c_{i,j}$ as Bob's majority answer corresponding to cut point $a_{i,j}$ for all $j \in [5]$. Also, by default $c_{i,0} = R$ and $c_{i,6} = L$. An illustration is in Figure 4.

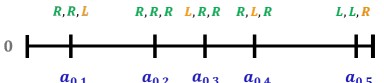

Figure 4: Illustration of step (2) for $i = 0$. Suppose $\eta = 3$. Alice cuts 3 times at each of the points $a_{0,j}$ and observes Bob's choices, which are marked near each such cut point. By default, Alice knows what the answer would be if she cut at 0 or 1, so those are set to $R$ and $L$, respectively. The truthful answers (reflecting Bob's favorite piece according to his actual valuation) are marked with green, while the lying answers are marked with orange.

**(3)** The points $a_{i,j}$ for $j \in \{0, \ldots, 6\}$ are arranged on a line and each is labelled $L$ or $R$, with the leftmost point $a_{i,0} = 0$ labelled $R$ and the rightmost point $a_{i,6} = 1$ labelled $L$. Then there is an index $j \in \{0, \ldots, 5\}$ such that $c_{i,j} = R$ and $c_{i,j+1} = L$.

Alice computes a smaller interval $I_{i+1}$, essentially consisting of $[a_{i,j}, a_{i,j+1}]$ and some extra space around it to make sure that $I_{i+1}$ contains Bob's midpoint as follows. If $j \in \{1, \ldots, 4\}$, set $I_{i+1} = [a_{i,j-1}, a_{i,j+2}]$. If $j = 0$, set $I_{i+1} = [a_{i,0}, a_{i,3}]$. If $j = 5$, set $I_{i+1} = [a_{i,3}, a_{i,6}]$. Then Alice iterates steps $(1-3)$ on the interval $I_1$. An illustration is in Figure 5.

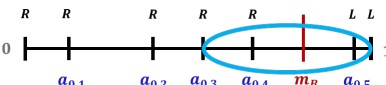

Figure 5: Illustration of step (3) for $i = 0$. Alice labels each point $a_{0,j}$ with the majority answer there. Then she identifies the index $j$ such that the point $a_{0,j}$ is labelled $R$ and the point $a_{0,j+1}$ is labelled $L$. At this stage she is assured that either the interval $[a_{0,j}, a_{0,j+1}]$ or one of the adjacent ones contains Bob's midpoint. Alice sets $I_1 = [a_{0,3}, a_{0,6}]$ and recurses on it.

The full proof explains why the index $j$ from step 3 is unique and why it is in fact necessary to include a slightly larger interval than $[a_{i,j}, a_{i,j+1}]$ in the recursion step, due to Bob potentially having lied if his midpoint was very close to a boundary of $[a_{i,j}, a_{i,j+1}]$ but on the other side.

**Sketch for the lower bound.** The lower bound of $\Omega\left(T^{\frac{\alpha+1}{2}}\right)$ relies on the observation that rounds where Alice cuts near $m_B$ and Bob picks his less-preferred piece cost Bob very little but cost Alice a lot. More precisely, suppose $m_A < m_B$ and Alice cuts at $m_B - \varepsilon$. Then compared to his regret bound, Bob loses $\Theta(\varepsilon)$ if he picks the wrong piece. On the other hand, Alice loses $\Theta(m_B - m_A) = \Theta(1)$ compared to her Stackelberg value.

Bob can use this asymmetry by acting as if his midpoint were $\Theta\left(T^{\frac{\alpha-1}{2}}\right)$ closer to $m_A$ than it really is. Lying $\Theta\left(T^{\frac{\alpha+1}{2}}\right)$ times costs Bob only $\Theta(T^\alpha)$ regret, but costs Alice $\Theta\left(T^{\frac{\alpha+1}{2}}\right)$ regret. To avoid accumulating more regret than this, Bob can afterwards revert to picking his truly preferred piece; the damage to Alice's payoff has already been done. □

**Proposition 3.** *Let $\alpha \in [0, 1)$. Suppose Bob plays a strategy that ensures his regret is $O(T^\alpha)$. Let $\mathcal{B}^\alpha$ denote the set of all such Bob strategies. If Alice does not know $\alpha$, she has a strategy $S_A$ that ensures her Stackelberg regret is $O\left(\frac{T}{\log T}\right)$.*

*The exponent is sharp: if $S_A$ guarantees Alice Stackelberg regret $O(T^\beta)$ against all Bob strategies in $\mathcal{B}^\alpha$ for some $\beta \in [0, 1)$, then $S_A$ has Stackelberg regret $\Omega(T)$ for some Bob strategy in $\mathcal{B}^\beta$.*

*Proof sketch.* Alice's strategy that achieves $O(T/\log T)$ regret follows the same template as her strategy from Proposition 2. The only difference is that she sets $\eta$ differently (and much larger) to cover any possible regret bound Bob could have.

The idea of the lower bound is that, if Alice does not know the value of $\alpha$ in Bob's regret bound, she cannot know when she has true information about Bob's preferences. We use this by having a Bob with $O(T^\beta)$ regret behave exactly like one with $O(T^\alpha)$ regret but a different midpoint. Then Bob can hide his deception from an Alice with $O(T^\beta)$ regret since he can tolerate more regret than her. □

Theorem 1 is implied by Propositions 2 and 3. The players' value densities must be bounded for Theorem 1 to hold; see Remark 1 in Appendix A.2 for a counterexample with unbounded densities.

# 5 Equitable payoffs

Here we sketch the proofs of Theorems 2 and 3. The formal proofs can be found in Appendix B.

Theorem 2 shows how Alice can get at least $1/2$ per round while keeping Bob at $1/2$ per round.

*Proof sketch of Theorem 2.* Alice's strategy uses Blackwell approachability (1956). A challenge is that Blackwell's original version required the number of player types to be finite, but Alice has to be prepared for an uncountably infinite variety of Bob's valuation functions. Another difference is that Alice's action space is also infinite, which turns out to be necessary.

We get around the infinite-Bob issue in two steps. First, Alice defines a countably infinite set $\overline{\mathcal{V}}$ as a stand-in for the full variety of Bobs; $\overline{\mathcal{V}}$ includes arbitrarily good approximations to any valuation.

Second, we replace Blackwell's original finite-dimensional space with a countably-infinite-dimensional one, where the elements of $\overline{\mathcal{V}}$ are the axes. We define an inner product on this space and adapt Blackwell's argument for it. Briefly, Alice's strategy tracks the average payoff to each type of Bob in $\overline{\mathcal{V}}$ and defines $\mathcal{S}$ to be the region of the space where all of them have payoffs at most $1/2$. In each round, she constructs a cut point which moves the Bobs' average payoff closer to $\mathcal{S}$, and in the limit traps them in $\mathcal{S}$.

Under this strategy, Alice's payoff guarantee is mostly a byproduct of Bob's. If Bob and Alice have the same value density, then their payoffs sum to 1, so bounding Bob's payoff to $1/2$ also bounds Alice's to $1/2$. We achieve the substantially better bound on Alice's payoff by explicitly including her value density $v_A$ in the set $\overline{\mathcal{V}}$ of Bobs, thus eliminating any approximation error. $\qquad\square$

Theorem 3 shows how Bob can do the same, albeit only in expectation in the simultaneous setting.

*Proof sketch of Theorem 3.* We cover the simultaneous setting first because it informs the sequential setting. In the simultaneous setting, Bob's algorithm is extremely simple: in each round, randomly select $L$ or $R$ with equal probability. The expected payoffs to each player follow immediately.

Bob's strategy for the sequential setting can be seen as a derandomized version of the simultaneous strategy. The simplest way to derandomize it would be to strictly alternate between $L$ and $R$, but if Bob runs that strategy Alice can easily exploit it. Instead, Bob mentally partitions the cake into $\sqrt{T}$ intervals $I_1, \ldots, I_{\sqrt{T}}$ of equal value to him. He then treats each interval $I_i$ as a separate cake, alternating between $L$ and $R$ for the rounds Alice cuts in $I_i$. Alice can still exploit this strategy on a single interval $I_i$, but doing so can only give her an average payoff of $1/2 + O(V_A(I_i)) \in 1/2 + O(1/\sqrt{T})$. The full proof shows this bound applies for any Alice strategy. $\qquad\square$

# 6 Fictitious play

In this section we include a proof sketch of Theorem 4, which analyzes the fictitious play dynamic. The formal proof can be found in Appendix C.

*Proof sketch of Theorem 4.* To analyze the fictitious play dynamic, we define for each $t = 0, \ldots, T$ two quantities called $\alpha_t$ and $\beta_t$. Let $\alpha_t = r_t - \ell_t$, where $r_t$ is the number of times Bob picked $R$ up to round $t$ and $\ell_t$ is the number of times he picked $L$. Let $\beta_t = \sum_{\tau=1}^{t} \big(2V_B([0, a_\tau]) - 1\big)$.

The quantities $\alpha_t$ and $\beta_t$ control what happens under fictitious play: Alice's decision in round $t + 1$ is based on $\alpha_t$ and Bob's decision in round $t + 1$ is based on $\beta_t$. These decisions in turn affect $\alpha_{t+1}$ and $\beta_{t+1}$, forming a dynamical system that results in a counterclockwise spiral through $\alpha$-$\beta$ space. Figure 6 illustrates the sequences $\alpha_t$ and $\beta_t$ for the instance in Figure 2.

We define $\rho_t = |\alpha_t| + |\beta_t|$ and formalize this spiral, by showing that the sequence $\{\rho\}_{t=0}^{T}$ is non-decreasing and analyzing the change in $(\alpha_t, \beta_t)$ from round to round. Figure 6 illustrates the parameter $\rho_t$ over time, while Figure 7 illustrates the spiral (associated with the same trajectory as in Figure 2 and 6), where the spiral is visualized as a scatter plot of the sequence $(\alpha_t, \beta_t)_{t \geq 1}$.

We first use these dynamics to bound Bob's payoff. Bob's payoff can be almost directly read off due to changes in $\beta_t$ closely matching changes in Bob's payoff. Bob's total payoff to round $t$ turns out to be of the order $t/2 \pm \rho_t$, so bounding the rate at which the spiral expands also bounds Bob's payoff.

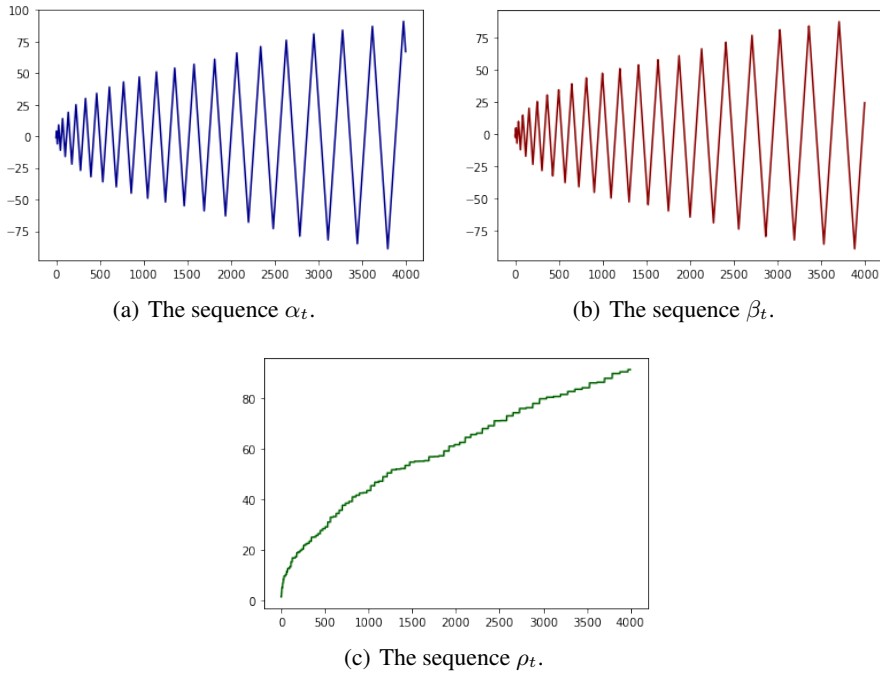

(a) The sequence $\alpha_t$.

(b) The sequence $\beta_t$.

(c) The sequence $\rho_t$.

Figure 6: Illustration of the sequences $\{\alpha_t\}_{t=1}^\infty$, $\{\beta_t\}_{t=1}^\infty$, and $\{\rho_t\}_{t=1}^\infty$ for the instance with trajectories shown in Figure 2. The X axis shows the round number $t$ and the Y axis the variable plotted.

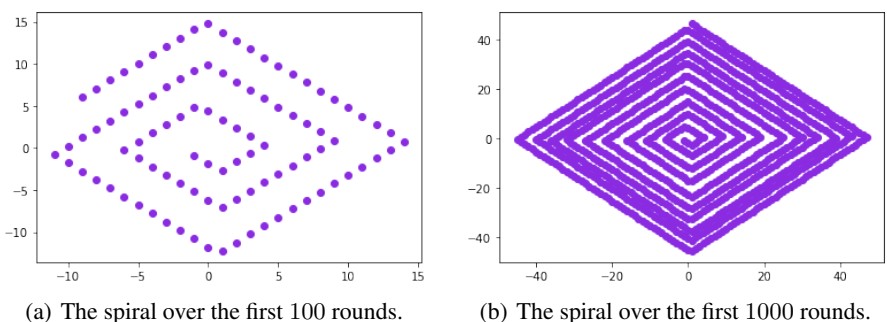

(a) The spiral over the first 100 rounds.

(b) The spiral over the first 1000 rounds.

Figure 7: Scatter plot of the sequence $(\alpha_t, \beta_t)_{t \geq 1}$, illustrating the spiral for the instance with trajectories shown in Figure 2, where the sequences $\alpha_t$ and $\beta_t$ are illustrated separately in Figure 6.

We then use the dynamics to bound the total payoff to Alice and Bob. Alice can only cut in the interior of the cake when $\alpha_t = 0$, which happens less and less often as the spiral expands. The players' total payoff when Alice cuts at one end of the cake is 1, so across $T$ rounds we show the sum of cumulative payoffs of the players is of the order $T \pm \Theta(\sqrt{T})$. Combining the bound on the total payoff with the bound on Bob's payoff gives a bound for Alice's payoff. □

## 7  Concluding remarks

There are several directions for future work. One direction is to consider a wider class of regret benchmarks and understand how the choice of benchmark influences the outcomes reached. Moreover, what payoff profiles are attained when the players use randomized algorithms such as exponential weights to update their strategies? It would also make sense to consider settings where the cake has both good and bad parts. Finally, studying richer feedback models, *e.g.*, when Alice and Bob takes turns cutting and choosing, or allowing Alice to divide the cake into any multiple measurable sets would be intriguing directions.

## Acknowledgements

We would like to thank the reviewers for useful feedback that helped improve the paper. This work was supported by US National Science Foundation CAREER grant CCF-2238372, DARPA QuICC, NSF AF:Small #2218678, NSF AF:Small #2114269, Army-Research Laboratory (ARL) #W911NF2410052, and MURI on Algorithms, Learning and Game Theory.

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

# A    Appendix: Alice exploiting Bob

In this section we present the proofs for Proposition 1, showing how Alice can exploit a myopic Bob that chooses his favorite piece in each round, and for Theorem 1, where Bob is nearly myopic.

Before giving the proofs, we formally define what we mean for Alice or Bob to ensure themselves a certain amount of regret.

**Definition 3** (Ensuring Alice's regret). *Suppose Bob plays a strategy $S_B$. A mixed strategy $S_A$ for Alice ensures her Stackelberg regret is at most $\gamma$ against $S_B$ if $Reg_A(H) \leq \gamma$ for all $T$-round histories $H$ that could have arisen under the strategy pair $(S_A, S_B)$.*

*If instead Alice only knows that Bob plays some strategy from a set $\mathcal{B}$ of strategies, then a mixed strategy $S_A$ for Alice ensures her Stackelberg regret is at most $\gamma$ if it ensures her Stackelberg regret is at most $\gamma$ against all $S_B \in \mathcal{B}$.*

**Definition 4** (Ensuring Bob's regret). *Suppose Alice plays a strategy $S_A$. A mixed strategy $S_B$ for Bob ensures his regret is at most $\gamma$ against $S_A$ if $Reg_B(H) \leq \gamma$ for all $T$-round histories $H$ that could have arisen under the strategy pair $(S_A, S_B)$.*

*In general, a mixed strategy $S_B$ for Bob ensures his regret is at most $\gamma$ if it ensures his regret is at most $\gamma$ against all Alice strategies.*

## A.1    Appendix: Exploiting a Myopic Bob

**Restatement of Proposition 1.** *If Bob plays myopically in the sequential setting, then Alice has a strategy that ensures her Stackelberg regret is $O(\log T)$.*

*Proof.* We consider an explore-then-commit type of algorithm for Alice. In the exploration phase, Alice does binary search to find Bob's midpoint (within accuracy of $1/T$). In the commitment (exploitation) phase, Alice repeatedly cuts at Bob's approximate midpoint. This leads to Alice getting nearly her Stackelberg value in nearly every round. Figure 8 shows a visualization of a cake instance with Alice and Bob's midpoints, respectively, with Alice's search process.

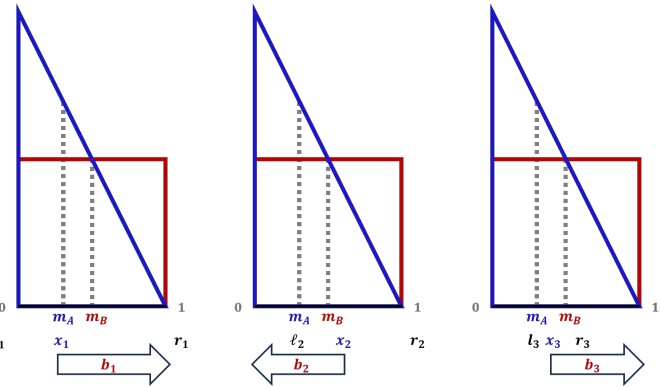

Figure 8: Alice's algorithm against myopic Bob in the exploration phase. Alice's density is shown with blue and her midpoint is $m_A$, while Bob's density is shown with red and his midpoint is $m_B$. The algorithm initialized $\ell_1 = 0$ and $r_1 = 1$ and then re-computes them iteratively depending on Bob's answers. The constructed interval $[\ell_t, r_t]$ shrinks exponentially and becomes closer to $m_B$ as the time $t$ increases.

Alice's algorithm is described precisely in Figure 9.

**Initialization**   Set $\ell_1 = 0, r_1 = 1$ and $\tau = \ln(T)$.

**Exploration**   For $t = 1, 2, \ldots, \tau$ :

- Cut at a point $x_t \in [\ell_t, r_t]$ such that $V_A([\ell_t, x_t]) = V_A([x_t, r_t])$. Then observe Bob's action $b_t$.

- If $b_t = L$, then set $(\ell_{t+1}, r_{t+1}) = (\ell_t, x_t)$.

- Else if $b_t = R$, then set $(\ell_{t+1}, r_{t+1}) = (x_t, r_t)$.

**Exploitation**   For $t = \tau + 1, \ldots, T$:

- If $m_A \leq \ell_\tau$, then cut at $\ell_\tau - 1/T$.

- If $m_A \geq r_\tau$, then cut at $r_\tau + 1/T$.

Figure 9: Algorithm A.1

We show that Algorithm A.1 in Figure 9 with $\tau = \Theta(\ln T)$ gives the desired regret bound in several steps.

**Bob's midpoint lies in the interval** $[r_t, \ell_t]$ **for all** $t$   To this end, we first claim that Bob's midpoint lies in $[\ell_t, r_t]$ at each round $t$. We proceed by induction on $t$. The base case is $t = 1$ clearly holds since $\ell_1 = 0$ and $r_1 = 1$, so $m_B \in [\ell_1, r_1]$. Suppose that $m_B \in [\ell_t, r_t]$ for some $t \geq 1$. Given that Alice cuts at $x_t \in (\ell_t, r_t)$, if $b_t = L$, this implies that $m_B \in [\ell_t, x_t]$. Hence $m_B \in [\ell_t, x_t] = [\ell_{t+1}, r_{t+1}]$. This argument also holds when Bob chooses $R$. Thus by induction, we conclude that $m_B \in [\ell_t, r_t]$ for every $t \in [T]$.

**Alice's midpoint satisfies** $m_A \notin (\ell_t, r_t)$ **for every** $t \geq 2$   Now, during the execution of the algorithm, we will next show that Alice's midpoint $m_A$ satisfies either of $m_A \leq \ell_t$ and $m_A \geq r_t$, *i.e.*, $m_A \notin (\ell_t, r_t)$ for every $t \geq 2$. To see this, recall that in the first round, Alice cuts $x_1 = m_A$. If $b_1 = L$, then $\ell_2 = 0$ and $r_2 = m_A$. In this case, $[\ell_2, r_2] = [0, m_A]$, so $m_A \notin (\ell_2, r_2)$. Afterwards, it still holds since the intervals only shrink, *i.e.*, $(\ell_{t+1}, r_{t+1}) \subset (\ell_t, r_t)$. Similarly, consider the case that $b_1 = R$. Then, we have $\ell_2 = m_A$ and $r_2 = 1$. Thus $[\ell_2, r_2] = [m_A, 1]$, which implies that $m_A \notin (\ell_2, r_2)$. Again since the intervals $(\ell_t, r_t)$ only shrink, we conclude that $m_A \notin (\ell_t, r_t)$ for every $t \geq 2$.

**Interval exponentially shrinks**   We have $V_A([\ell_t, r_t]) = V_A([\ell_{t-1}, r_{t-1}])/2$ for every $t \in [T]$, as we shrink the interval by cutting a point that equalizes Alice's value for both parts within the interval. This implies that $V_A([\ell_\tau, r_\tau]) = 2^{-\tau+1}$.

**Bounding exploitation phase regret**   In the exploitation phase, due to the observation above, we have two cases: (i) $m_A \leq \ell_\tau$ and (ii) $m_A \geq r_\tau$. We will prove that in either case, Alice's single-round regret in the exploitation phase is at most $2^{-\tau+1} + \Delta/T$.

- For the first case of $m_A \leq \ell_\tau$, Alice keeps cutting at $\ell_\tau - 1/T$ for the rest of rounds as per the algorithm's description. Then, Bob will myopically choose $R$ and Alice will obtain $V_A([0, \ell_\tau - 1/T])$. In this case, we have that $m_A \leq m_B$ since $m_B \in [\ell_\tau, r_\tau]$. Then, Alice's single-round regret in the exploitation phase is bounded by

$$V_A([0, m_B]) - V_A([0, \ell_\tau - 1/T]) = V_A([\ell_\tau, m_B]) + V_A([\ell_\tau - 1/T, \ell_\tau])$$

$$\leq V_A([\ell_\tau, r_\tau]) + \frac{\Delta}{T}$$

$$= 2^{-\tau+1} + \frac{\Delta}{T}.$$

- Otherwise suppose $m_A \geq r_\tau$. According to the algorithm, Alice keeps cutting $r_\tau + 1/T$ for all the rest of the rounds, and Bob will respond with $L$. In this case we have $m_A \geq m_B$

since $m_B \in [\ell_\tau, r_\tau]$. Similarly, Alice's single-round regret can be upper-bounded by

$$V_A([m_B, 1]) - V_A([r_\tau + 1/T, 1]) = V_A([m_B, r_\tau]) + V_A([r_\tau, r_\tau + 1/T])$$
$$\leq V_A([\ell_\tau, r_\tau]) + \frac{\Delta}{T}$$
$$= 2^{-\tau+1} + \frac{\Delta}{T}.$$

Hence in both cases, Alice's single-round regret in the exploitation phase is at most $2^{-\tau+1} + \Delta/T$.

**Final regret bound**  Overall, by simply upper-bounding Alice's single-round regret in the exploration phase by 1, we obtain the following upper bound for the total regret:

$$\tau \cdot 1 + (T - \tau) \cdot \left( 2^{-\tau+1} + \frac{\Delta}{T} \right).$$

Plugging $\tau = \ln(T)$, we obtain the regret bound of $O(\ln T)$, which completes the proof.[3]  □

## A.2  Appendix: Exploiting a Nearly Myopic Bob

In this section we prove Theorem 1, which explains the payoffs achievable by Alice when Bob has a strategy with sub-linear regret. We restate it here for reference.

**Restatement of Theorem 1** (Exploiting a nearly myopic Bob). *Let $\alpha \in [0, 1)$. Suppose Bob plays a strategy that ensures his regret is $O(T^\alpha)$. Let $\mathcal{B}^\alpha$ denote the set of all such Bob strategies.*

- *If Alice knows $\alpha$, she has a strategy $S_A = S_A(\alpha)$ that ensures her Stackelberg regret is $O\big(T^{\frac{\alpha+1}{2}} \log T\big)$. The exponent is sharp: Alice's Stackelberg regret is $\Omega\big(T^{\frac{\alpha+1}{2}}\big)$ for some Bob strategy in $\mathcal{B}^\alpha$.*

- *If Alice does not know $\alpha$, she has a strategy $S_A$ that ensures her Stackelberg regret is $O\big(\frac{T}{\log T}\big)$. The exponent is sharp: if $S_A$ guarantees Alice Stackelberg regret $O(T^\beta)$ against all Bob strategies in $\mathcal{B}^\alpha$ for some $\beta \in [0, 1)$, then $S_A$ has Stackelberg regret $\Omega(T)$ for some Bob strategy in $\mathcal{B}^\beta$.*

*Proof of Theorem 1.* The known-$\alpha$ upper bound of $O\big(T^{\frac{\alpha+1}{2}} \log T\big)$ follows from invoking Proposition 4 with $f(T) = T^\alpha$. The $\Omega\big(T^{\frac{\alpha+1}{2}}\big)$ lower bound is Proposition 5.

The lower bound for the case where $\alpha$ is unknown follows from Lemma 3. The upper bound follows from invoking Proposition 4 with $f(T) = \frac{T}{(\log T)^4}$.  □

Both upper bounds follow the same template, which is captured by the following proposition.

**Proposition 4.** *Suppose Bob's strategy has regret $O(f(T))$, for $f(T) \in o\big(\frac{T}{(\log T)^2}\big)$ and $f(T) \geq 1$. If Alice knows $f$, then she has a strategy that guarantees her Stackelberg regret is $O(\sqrt{T \cdot f(T)} \log T)$. In particular, if*

- *Bob's strategy has regret at most $r f(T)$, for some $r > 0$; and*

- *$T$ is large enough so that $T > \exp\big(\frac{4r\Delta}{\delta}\big)$ and $f(T) < \frac{T}{(\ln T)^2}$;*

*then Alice's payoff satisfies:*

$$u_A \geq T \cdot u_A^* - \left( \frac{5}{\ln 2} + 6 \right) \sqrt{f(T) \cdot T} \ln T.$$

Before proving the proposition, we present the algorithm that Alice will run to beat a Bob with a regret guarantee of $O(f(T))$.

---

[3]We do not optimize over $\tau$.

**Alice's strategy when Bob's regret is at most** $r \cdot f(T)$**, for some constant** $r > 0$**:**

*Informational assumption: Alice needs to know $f$ and $T$, but not $r$.*

**Initialization**  Define
$$x_0 = 0; \quad y_0 = 1; \quad \eta = \lceil \sqrt{f(T) \cdot T} \rceil; \quad n = \lfloor -\log_2 \big(3\sqrt{f(T)/T} \ln T\big) \rfloor.$$

**Exploration**  For $i = 0, 1, \ldots, n-1$:

- **Step 1:** Set $a_{i,0} = x_i$ and $a_{i,6} = y_i$. For $j \in [5]$, let $a_{i,j}$ be the point with
$$V_A([x_i, a_{i,j}]) = \frac{j}{6} V_A([x_i, y_i]).$$

- **Step 2:** For $j \in [5]$:
    - Cut at $a_{i,j}$ for $\eta$ rounds. Define $c_{i,j} = L$ if the majority of Bob's answers were left when the cut point was $a_{i,j}$, and $c_{i,j} = R$ otherwise.

- **Step 3:**
    - **(3.a):** If $c_{i,j} = L \; \forall j \in [5]$, then set $x_{i+1} = x_i$ and $y_{i+1} = a_{i,3}$.
    - **(3.b):** If $c_{i,j} = R \; \forall j \in [5]$, then set $x_{i+1} = a_{i,3}$ and $y_{i+1} = y_i$.
    - **(3.c):** Else, there exists a unique $k \in [4]$ such that $c_{i,j} = R$ for all $j \leq k$ and $c_{i,j} = L$ for all $j > k$. Set $x_{i+1} = a_{i,k-1}$ and $y_{i+1} = a_{i,k+2}$.

**Exploitation**  For the rest of the rounds, Alice cuts at $\chi$ based on the following cases:
$$\chi = \begin{cases} x_n & m_A < x_n \\ y_n & m_A > y_n \\ m_A & m_A \in [x_n, y_n]. \end{cases}$$

Figure 10: Algorithm A.2

*Proof of Proposition 4.*  Overall, Alice will use an explore-then-commit style of algorithm:

- In the exploration phase, Alice conducts a variant of binary search to locate Bob's midpoint $m_B$ within an accuracy of $O\big(\sqrt{f(T)/T} \log T\big)$.

- In the exploitation phase, Alice cuts near the estimated midpoint for the rest of the rounds.

The main difficulty that Alice encounters is to precisely locate Bob's midpoint in the exploration phase, since Bob can fool Alice if she cuts sufficiently close to his midpoint. We overcome this challenge by having Alice's algorithm stay far enough from $m_B$ so that Bob is forced to answer truthfully most of the time.

**Notation.**  Let $w = \sqrt{f(T)/T} \ln T$. Then $n = \lfloor -\log_2(3w) \rfloor$. Since $f(T) < \frac{T}{(\ln T)^2}$ by the assumption in the proposition statement, we have $w < 1$ and thereby $n \geq 0$. Also recall the proposition statement assumes that Bob's strategy guarantees him a regret of at most $rf(T)$ for some $r > 0$. Moreover, $T$ was chosen such that $T > \exp\big(\frac{4r\Delta}{\delta}\big)$.

Consider the Alice strategy described in Algorithm A.2 (Fig. 10). By Lemma 1, the exploration phase in Alice's strategy is well-defined.

Next we derive some useful observations and then combine them to upper-bound Alice's regret.

**Useful observations.**  By Lemma 2, we have $m_B \in [x_n, y_n]$. Consider the cut point $\chi$ in the exploitation phase. We write $\text{INTV}[x, y]$ to denote the interval $[x, y]$ if $y \geq x$ and $[y, x]$ if $x > y$.

Then, we obtain

$$\begin{aligned}
V_A(\text{INTV}[\chi, m_B]) &\leq V_A([x_n, y_n]) && \text{(By definition of } \chi) \\
&= 2^{-n} && \text{(By property 2 of Lemma 2)} \\
&\leq 2^{\log_2(3w)+1} && \text{(Plugging in } n \text{ and using } -\lfloor -x \rfloor \leq x+1 \text{ )} \\
&= 6\sqrt{\frac{f(T)}{T}} \ln T, && (1)
\end{aligned}$$

where the last identity in (1) holds by definition of $w$.

To upper-bound the number of times that Bob chooses the piece he likes less in the exploitation phase, we consider the following three cases with respect to $\chi$:

(a) If $\chi = x_n$ then $m_A < x_n$. Thus $x_n \neq 0$, so $V_B([\chi, m_B]) > r\sqrt{f(T)/T}$ by Lemma 2. Since Bob's regret is at most $rf(T)$, it follows that Bob takes the wrong piece at most $\frac{1}{2}\sqrt{f(T) \cdot T}$ times.

(b) If $\chi = y_n$ then $m_A > y_n$. Thus $y_n \neq 1$, so $V_B([m_B, \chi]) > r\sqrt{f(T)/T}$ by Lemma 2. Since Bob's regret is at most $rf(T)$, it follows that Bob takes the wrong piece at most $\frac{1}{2}\sqrt{f(T) \cdot T}$ times.

(c) If $\chi = m_A$, then there is no wrong piece because Alice values both equally. Thus this case does not increase the count of incorrect decisions.

**Putting it all together.** In the exploration phase, Alice accumulates regret at most $n \cdot 5 \left\lceil \sqrt{f(T) \cdot T} \right\rceil$, since that is the length of the exploration phase. In the exploitation phase, the regret comes from two sources:

- The gap between $\chi$ and $m_B$, which is bounded in equation (1).

- The rounds in the exploitation phase in which Bob chooses his least favorite piece. There are at most $\frac{1}{2}\sqrt{f(T) \cdot T}$ such rounds by cases (a-c). Thus Alice's cumulative regret due to these rounds is also at most $\frac{1}{2}\sqrt{f(T) \cdot T}$.

Then Alice's overall regret is at most:

$$\begin{aligned}
n \cdot 5 &\left\lceil \sqrt{f(T) \cdot T} \right\rceil + T \cdot 6\sqrt{f(T)/T} \ln T + \frac{1}{2}\sqrt{f(T) \cdot T} \\
&\leq n \cdot 10\sqrt{f(T) \cdot T} + T \cdot 6\sqrt{f(T)/T} \ln T + \frac{1}{2}\sqrt{f(T) \cdot T} \\
&\qquad\qquad\qquad\qquad\qquad \text{(Since } \lceil x \rceil \leq 2x \;\; \forall x \geq 1) \\
&\leq \left( \frac{5}{\ln 2} + 6 \right) \sqrt{f(T) \cdot T} \ln T, \qquad \text{(Plugging in } n \text{ and rearranging)}
\end{aligned}$$

which is $O\left( \sqrt{f(T) \cdot T} \ln T \right)$. This completes the proof. $\square$

The following lemma shows that the exploration phase is well-defined.

**Lemma 1.** *Alice's strategy from Algorithm A.2 (Fig. 10) has the following properties:*

(i) *If step $(3.c)$ is executed in Alice's exploration phase, then there is a unique index $k \in [4]$ such that $c_{i,j} = R \; \forall j \leq k$ and $c_{i,j} = L \; \forall j > k$.*

(ii) *For each $j \in [5]$, define $\tilde{c}_{i,j} = L$ if Bob prefers $[0, a_{i,j}]$ to $[a_{i,j}, 1]$ and $\tilde{c}_{i,j} = R$ otherwise. If there exists $j \in [5]$ such that $\tilde{c}_{i,j} \neq c_{i,j}$, then $m_B \in (a_{i,j-1}, a_{i,j+1})$.*

(iii) *For all $i \in \{0, \ldots, n-1\}$ and $j \in \{0, 1, \ldots, 5\}$, we have $V_B([a_{i,j}, a_{i,j+1}]) > 2r\sqrt{f(T)/T}$.*

*Proof.* We prove each of the parts $(i - iii)$ required by the lemma.

**Proof of part** $(iii)$. Let $j \in \{0, \ldots, 5\}$. Bob's valuation for the interval $[a_{i,j}, a_{i,j+1}]$ can be lower bounded as follows:

$$V_B([a_{i,j}, a_{i,j+1}]) \geq \delta \cdot (a_{i,j+1} - a_{i,j}) \qquad \text{(Since } v_B(x) \geq \delta \; \forall x \in [0,1])$$

$$\geq \frac{\delta}{\Delta} V_A([a_{i,j}, a_{i,j+1}]) \qquad \text{(Since } v_A(x) \leq \Delta \; \forall x \in [0,1])$$

$$= \frac{\delta}{\Delta} \cdot \frac{1}{6} V_A([x_i, y_i]). \tag{2}$$

By definition, Alice's strategy halves the cake interval considered with each iteration $i \in \{0, \ldots, n-1\}$, that is: $V_A([x_i, y_i]) = 1/2 \cdot V_A([x_{i-1}, y_{i-1}])$. Thus $V_A([x_i, y_i]) = 2^{-i}$ and $2^{-i} \geq 2^{-n}$ for all $i \in \{0, \ldots, n\}$. Combining these observations with inequality (2), we obtain

$$V_B([a_{i,j}, a_{i,j+1}]) \geq 2 \cdot \frac{\delta}{12\Delta} 2^{-n}. \tag{3}$$

Let $w = \sqrt{f(T)/T} \ln T$. Then $n = \lfloor -\log_2(3w) \rfloor$. We have

$$\frac{\delta}{12\Delta} 2^{-n} = \frac{\delta}{12\Delta} 2^{-\lfloor -\log_2(3w) \rfloor} \qquad \text{(By definition of } n)$$

$$\geq \frac{\delta}{12\Delta} 2^{\log_2(3w)} \qquad \text{(Since } -\lfloor -x \rfloor \geq x)$$

$$> \frac{\delta}{4\Delta} \cdot \sqrt{\frac{f(T)}{T}} \cdot \frac{4r\Delta}{\delta} \qquad \text{(By definition of } w \text{ and since } T > \exp\left(\frac{4r\Delta}{\delta}\right))$$

$$= r\sqrt{f(T)/T}. \tag{4}$$

Combining inequalities (3) and (4), we conclude that

$$V_B([a_{i,j}, a_{i,j+1}]) > 2r\sqrt{\frac{f(T)}{T}}. \tag{5}$$

This concludes the proof of part $(iii)$.

**Proof of parts** $(i)$ **and** $(ii)$. For each $i = 0, \ldots, n-1$, we will show that at most one of the majority answers $c_{i,j}$, for $j \in [5]$, is different from Bob's truthful response.

To be precise, recall from the lemma statement that $\tilde{c}_{i,j} \in \{L, R\}$ is Bob's truthful response that maximizes his value when Alice cut at $a_{i,j}$.

Define

$$S_i = \left\{ j \in [5] : c_{i,j} \neq \tilde{c}_{i,j} \right\}. \tag{6}$$

Let INTV$[x, y]$ denote the interval $[x, y]$ if $y \geq x$ and $[y, x]$ if $x > y$. If $j \in S_i$, it must be the case that Bob picked the wrong piece in at least $\frac{1}{2}\sqrt{f(T) \cdot T}$ rounds in which the cut was $a_{i,j}$. Then Bob accumulated at least $V_B(\text{INTV}[m_B, a_{i,j}])$ regret in each such round. Let $\ell \in [4]$. We have

$$\frac{1}{2}\sqrt{f(T) \cdot T} \sum_{j \in S_i} V_B(\text{INTV}[m_B, a_{i,j}]) \leq r \cdot f(T) \quad \text{(Since Bob's total regret is at most } r \cdot f(T))$$

$$< \frac{1}{2} V_B([a_{i,\ell}, a_{i,\ell+1}]) \sqrt{f(T) \cdot T}. \qquad \text{(By (5))}$$

Dividing both sides by $\frac{1}{2}\sqrt{f(T) \cdot T}$ gives:

$$\sum_{j \in S_i} V_B(\text{INTV}[m_B, a_{i,j}]) < V_B([a_{i,\ell}, a_{i,\ell+1}]) \quad \forall \ell \in [4]. \tag{7}$$

We show that $|S_i| \leq 1$. Suppose towards a contradiction that $|S_i| > 1$, meaning there exist indices $j, \ell \in S_i$ with $j \neq \ell$. Then

$$\text{INTV}[a_{i,j}, a_{i,\ell}] \subseteq (\text{INTV}[m_B, a_{i,j}] \cup \text{INTV}[m_B, a_{i,\ell}]).$$

This implies that

$$V_B(\text{INTV}[a_{i,j}, a_{i,\ell}]) \le V_B(\text{INTV}[m_B, a_{i,j}]) + V_B(\text{INTV}[m_B, a_{i,\ell}]) \le \sum_{j \in S_i} V_B(\text{INTV}[m_B, a_{i,j}]),$$

(8)

which contradicts (7). Thus the assumption was false and $|S_i| \le 1$.

For any $j \in S_i$, we must have either $m_B \in (a_{i,j-1}, a_{i,j}]$ or $m_B \in [a_{i,j}, a_{i,j+1})$, as otherwise (7) would be violated. Therefore, if $\tilde{c}_{i,j} \ne c_{i,j}$ for some $j \in [5]$, then $m_B \in (a_{i,j-1}, a_{i,j+1})$. This is part $(ii)$ required by the lemma.

Finally, we prove part $(i)$. We will show there exists a unique $k \in [4]$ such that $c_{i,j} = R \ \forall j \le k$ and $c_{i,j} = L \ \forall j > k$. The proof considers two cases:

- Case $|S_i| = 0$. Then every $c_{i,j}$ truthfully reflects Bob's preferences: $c_{i,j} = R$ if $a_{i,j} < m_B$, and $c_{i,j} = L$ if $a_{i,j} > m_B$; $c_{i,j} \in \{L, R\}$ if $a_{i,j} = m_B$. An illustration can be seen in Figure 11. In all cases, there is a single switch from $R$ to $L$, and so the index $k$ is unique.

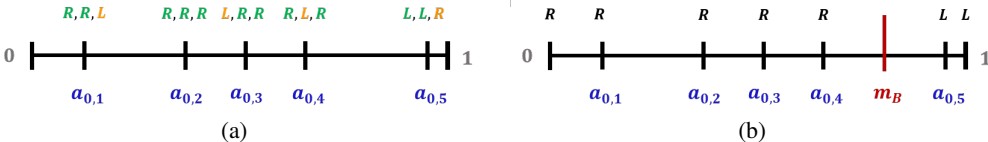

Figure 11: Illustration of Alice's initial cuts for $i = 0$. In this example, she cuts 3 times at each of the points $a_{0,j}$ and observes Bob's choices, which are marked with $L/R$ near each such cut point. By default, Alice knows what the answer would be if she cut at 0 or 1, so those are set to $R$ and $L$, respectively. In Figure (a), the truthful answers (reflecting Bob's favorite piece according to his actual valuation) are marked with green, while the lying answers are marked with orange. The majority answer at each cut point $a_{0,j}$, denoted $c_{0,j}$, is illustrated in Figure (b). In this example, the majority answer at each cut point is consistent with Bob's true preference.

- Case $|S_i| = 1$. Then all but one of the $c_{i,j}$'s truthfully reflect Bob's preferences. An illustration can be seen in Figure 12.

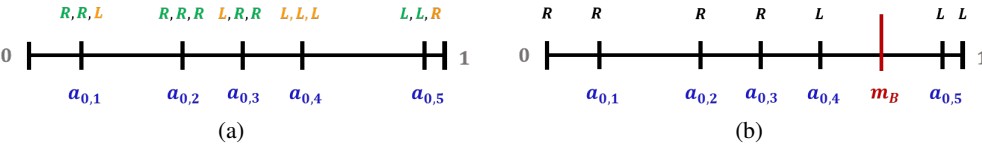

Figure 12: Illustration of Alice's initial cuts for $i = 0$. In this example, she cuts 3 times at each of the points $a_{0,j}$ and observes Bob's choices, which are marked with $L/R$ near each such cut point. By default, Alice knows what the answer would be if she cut at 0 or 1, so those are set to $R$ and $L$, respectively. In Figure (a), the truthful answers (reflecting Bob's favorite piece according to his actual valuation) are marked with green, while the lying answers are marked with orange. The majority answer at each cut point $a_{0,j}$, denoted $c_{0,j}$, is illustrated in Figure (b). In this example, the majority answer at each cut point is consistent with Bob's true preference *except* for cut point $a_{0,4}$ where Bob lied every time and so the majority is incorrect as well.

By part $(ii)$ of the lemma, the only exception occurs at an index $j$ with the property $m_B \in (a_{i,j-1}, a_{i,j+1})$. But then $c_{i,\ell} = \tilde{c}_{i,\ell} = R$ for $\ell \le j - 1$ and $c_{i,\ell} = \tilde{c}_{i,\ell} = L$ for $\ell \ge j + 1$, so regardless of $c_{i,j}$ there will be a single switch from $R$ to $L$.

This concludes that the conditions for $c_{i,j}$ in Step 3.c hold if the conditions in steps 3.a and 3.b do not. This concludes the proof of part $(i)$. $\qquad\square$

The following lemma further reveals several properties of the constructed intervals during the execution of the algorithm.

**Lemma 2.** *In the exploration phase of Algorithm A.2 (Fig. 10), Alice constructs a sequence of intervals*

$$[x_0, y_0], [x_1, y_1], \ldots, [x_n, y_n]$$

*such that the following properties hold:*

- Property 1: $x_0 = 0$ *and* $y_0 = 1$,

- Property 2: $V_A([x_{i+1}, y_{i+1}]) = \frac{1}{2} V_A([x_i, y_i])$ *for* $i = 0, \ldots, n-1$,

- Property 3: $m_B \in [x_i, y_i]$, *for all* $i$,

- Property 4: *If* $x_i \neq 0$, *then* $V_B([x_i, m_B]) > r\sqrt{f(T)/T}$.

- Property 5: *If* $y_i \neq 1$, *then* $V_B([m_B, y_i]) > r\sqrt{f(T)/T}$.

*Proof.* Property 1 holds since $[x_0, y_0] = [0, 1]$ by definition of the algorithm.

Property 2 follows from our choice of $x_{i+1}$ and $y_{i+1}$ always ensuring that $[x_{i+1}, y_{i+1}]$ contains 3 of the 6 intervals of equal value the $a_{i,j}$ divide $[x_i, y_i]$ into.

We will show Properties 3-5 by induction. The base case is $i = 0$. Then $[x_0, y_0] = [0, 1]$. Properties 3-5 are vacuously true for this interval.

Assume that Properties 3-5 hold for $i \in \{0, 1, \ldots, n-1\}$. For each $j \in [5]$, let $\tilde{c}_{i,j}$ represent Bob's truthful answer when the cut point is $a_{i,j}$. Formally, we have $\tilde{c}_{i,j} = L$ if Bob prefers $[0, a_{i,j}]$ to $[a_{i,j}, 1]$ and $\tilde{c}_{i,j} = R$ otherwise.

We show Properties 3-5 also hold for $i+1$ by considering the next three cases:

**Case $c_{i,j} = L$ for all $j$.** In this case, the majority of Bob's answers is $L$ at each cut point used by Alice. An illustration can be seen in Figure 13.

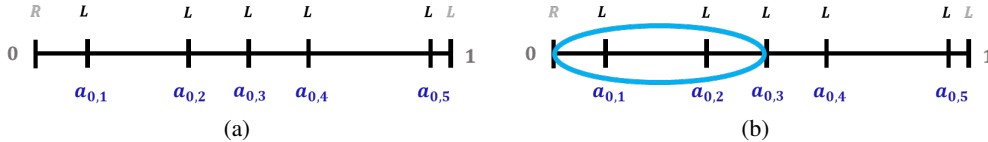

(a)  (b)

Figure 13: Illustration of Alice's initial cuts for $i = 0$. At each cut point $a_{0,j}$, the majority of Bob's answers is $L$ (i.e. $c_{0,j} = L$). Then the algorithm recurses in the interval $[0, a_{0,3}]$.

Then the algorithm recurses in the interval $[x_{i+1}, y_{i+1}]$ given by $x_{i+1} = x_i$ and $y_{i+1} = a_{i,3}$. By Lemma 1, we have $\tilde{c}_{i,j} = c_{i,j}$ for each of $j \in \{2, 3, 4, 5\}$, as otherwise Bob's "true" preferences would alternate between $R$ and $L$ more than once.

We claim that $m_B \in [x_i, a_{i,2})$. To see this, consider two cases:

- Case $\tilde{c}_{i,1} = c_{i,1}$: then $m_B \in [x_i, a_{i,1}]$ since the majority answers are consistent with Bob's true preference.
- Case $\tilde{c}_{i,1} \neq c_{i,1}$: then $m_B \in (x_i, a_{i,2})$ by Lemma 1.

Then $m_B \in [x_i, a_{i,2}) \subset [x_{i+1}, y_{i+1}]$, which proves Property 3 for $i+1$.

If $x_{i+1} = 0$, then Property 4 vacuously follows. Otherwise, we have $x_{i+1} = x_i$. Then by the inductive hypothesis we obtain $V_B([x_{i+1}, m_B]) = V_B([x_i, m_B]) > r\sqrt{f(T)/T}$. Thus Property 4 holds for $i+1$ as well.

Since $m_B \in [x_i, a_{i,2})$, we have

$$V_B([m_B, y_{i+1}]) \geq V_B([a_{i,2}, a_{i,3}]) > 2r\sqrt{f(T)/T},$$

and so Property 5 holds for $i + 1$.

**Case $c_{i,j} = R$ for all $j$.** In this case, the majority of Bob's answers is $R$ at each cut point used by Alice. An illustration can be seen in Figure 14.

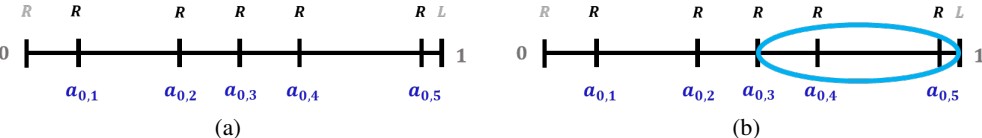

(a)                                                              (b)

Figure 14: Illustration of Alice's initial cuts for $i = 0$. At each cut point $a_{0,j}$, the majority of Bob's answers is $R$ (i.e. $c_{0,j} = R$). Then the algorithm recurses in the interval $[a_{0,3}, 1]$.

Then the algorithm recurses in the interval $[x_{i+1}, y_{i+1}]$ given by $x_{i+1} = a_{i,3}$ and $y_{i+1} = y_i$. By Lemma 1, we have $\tilde{c}_{i,j} = c_{i,j}$ for each of $j \in \{1, 2, 3, 4\}$.

We claim that $m_B \in (a_{i,4}, y_i]$. To see this, consider two cases:

- Case $\tilde{c}_{i,5} = c_{i,5}$: then $m_B \in [a_{i,5}, y_i]$ since the majority answers are consistent with Bob's true preference.
- Case $\tilde{c}_{i,5} \neq c_{i,5}$: then $m_B \in (a_{i,4}, y_i)$ by Lemma 1.

Then $m_B \in (a_{i,4}, y_i] \subset [x_{i+1}, y_{i+1}]$, which proves Property 3 for $i + 1$.

Since $m_B \in (a_{i,4}, y_i]$, we have

$$V_B([x_{i+1}, m_B]) \geq V_B([a_{i,3}, a_{i,4}]) > 2r\sqrt{f(T)/T},$$

and so Property 4 holds for $i + 1$.

If $y_{i+1} = 1$, then Property 5 vacuously follows. Otherwise, we have $y_{i+1} = y_i$. Then by the inductive hypothesis we obtain $V_B([m_B, y_{i+1}]) = V_B([m_B, y_i]) > r\sqrt{f(T)/T}$. Thus Property 5 holds for $i + 1$ as well.

**Case where there is a transition from $R$ to $L$ and the last $R$ is $c_{i,k}$ for some $k \in \{1, \ldots, 4\}$.** An illustration can be seen in Figure 15.

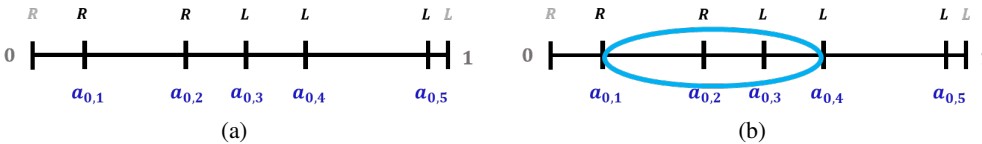

(a)                                                              (b)

Figure 15: Illustration of Alice's initial cuts for $i = 0$. In this example, there is a transition from $R$ to $L$ in the interval $[a_{0,2}, a_{0,3}]$. Then the algorithm recurses in the interval $[a_{0,1}, a_{0,4}]$, which is guaranteed to contain Bob's midpoint.

In this case, Alice recurses on the interval $[x_{i+1}, y_{i+1}]$ given by

$$x_{i+1} = a_{i,k-1} \quad \text{and} \quad y_{i+1} = a_{i,k+2}.$$

If any of the $c_{i,j}$ differ from the $\tilde{c}_{i,j}$, it must be $c_{i,k}$ or $c_{i,k+1}$ because otherwise Bob's true preferences would alternate between $R$ and $L$ more than once, which is impossible. We consider three sub-cases:

1. **Case $\tilde{c}_{i,k} \neq c_{i,k}$.** Each time Bob picked his less-preferred piece when Alice cut at $a_{i,k}$, he lost $2V_B(\textsc{Intv}[a_{i,k}, m_B])$ in value compared to his regret benchmark. Since $\tilde{c}_{i,k} \neq c_{i,k}$, he did so at least $\frac{1}{2}\sqrt{f(T) \cdot T}$ times. In order to have regret at most $rf(T)$, he must have

$$V_B(\textsc{Intv}[a_{i,k}, m_B]) \leq r\sqrt{\frac{f(T)}{T}}. \tag{9}$$

By Lemma 1, we have

$$m_B \in (a_{i,k-1}, a_{i,k+1}) \tag{10}$$

and

$$V_B([a_{i,k-1}, a_{i,k}]) > 2r\sqrt{\frac{f(T)}{T}}. \tag{11}$$

Combining equations (9), (10), and (11) we obtain

$$
\begin{aligned}
V_B([x_{i+1}, m_B]) = V_B([a_{i,k-1}, m_B]) && \text{(Since } x_{i+1} = a_{i,k-1}) \\
\geq V_B([a_{i,k-1}, a_{i,k}]) - V_B(\textsc{Intv}[a_{i,k}, m_B]) && \\
&& \text{(Since } m_B > a_{i,k-1} \text{ by (10))} \\
> r\sqrt{f(T)/T}. && \text{(By (9) and (11))}
\end{aligned}
$$

Thus $V_B([x_{i+1}, m_B]) > r\sqrt{f(T)/T}$, so Property 4 holds for $i+1$.
Since $m_B \in (a_{i,k-1}, a_{i,k+1})$, we have

$$V_B([m_B, y_{i+1}]) \geq V_B([a_{i,k+1}, a_{i,k+2}]) > 2r\sqrt{f(T)/T},$$

proving Property 5 for $i+1$.

2. **Case $\tilde{c}_{i,k+1} \neq c_{i,k+1}$.** Each time Bob picked his less-preferred piece when Alice cut at $a_{i,k+1}$, he lost $2V_B(\textsc{Intv}[a_{i,k+1}, m_B])$ in value compared to his regret benchmark. Since $\tilde{c}_{i,k+1} \neq c_{i,k+1}$, he did so at least $\frac{1}{2}\sqrt{f(T) \cdot T}$ times. In order to have regret at most $rf(T)$, he must have

$$V_B(\textsc{Intv}[a_{i,k+1}, m_B]) \leq r\sqrt{\frac{f(T)}{T}}. \tag{12}$$

By Lemma 1, we have

$$m_B \in (a_{i,k}, a_{i,k+2}) \tag{13}$$

and

$$V_B([a_{i,k}, a_{i,k+1}]) > 2r\sqrt{\frac{f(T)}{T}}. \tag{14}$$

Combining equations (12), (13), and (14) we obtain

$$
\begin{aligned}
V_B([m_B, y_{i+1}]) = V_B([m_B, a_{i,k+2}]) && \text{(Since } y_{i+1} = a_{i,k+2}) \\
\geq V_B([a_{i,k+1}, a_{i,k+2}]) - V_B(\textsc{Intv}[a_{i,k+1}, m_B]) && \\
&& \text{(Since } m_B < a_{i,k+2} \text{ by (13))} \\
> r\sqrt{f(T)/T}. && \text{(By (12) and (14))}
\end{aligned}
$$

Thus $V_B([m_B, y_{i+1}]) > r\sqrt{f(T)/T}$, so Property 5 holds for $i+1$.

Since $m_B \in (a_{i,k}, a_{i,k+2})$, we have

$$V_B([x_{i+1}, m_B]) \geq V_B([a_{i,k-1}, a_{i,k}]) > 2r\sqrt{f(T)/T},$$

which proves Property 4 for $i+1$.

3. **Case $\tilde{c}_{i,j} = c_{i,j}$ for all $j \in [5]$.** Then $m_B \in [a_{i,k}, a_{i,k+1}]$. By Lemma 1, we have

$$V_B([x_i, m_B]) \geq V_B([x_i, a_{i,k}]) = V_B([a_{i,k-1}, a_{i,k}]) > 2r\sqrt{f(T)/T} \qquad (15)$$

and

$$V_B([m_B, y_i]) \geq V_B([a_{i,k+1}, y_i]) = V_B([a_{i,k+1}, a_{i,k+2}]) > 2r\sqrt{f(T)/T}. \qquad (16)$$

Inequality (15) implies Property 4 for $i+1$, while inequality (16) implies Property 5 for $i+1$.

In all three cases, $m_B \in [a_{i,k-1}, a_{i,k+2}] = [x_{i+1}, y_{i+1}]$, showing Property 3 for $i+1$.

Thus, in all three cases, Properties 3-5 hold for $i+1$. By induction, they hold for all $i \in \{0, 1, \ldots, n\}$. This completes the proof. □

**Proposition 5.** *Let $v_A$ be an arbitrary Alice density and $\alpha \in [0, 1)$. Then there exists a value density function $\tilde{v}_B = \tilde{v}_B(v_A)$ and strategy $\tilde{S}_B = \tilde{S}_B(v_A, \alpha)$ that ensures Bob's regret is $O(T^\alpha)$ while Alice's Stackelberg regret is $\Omega\left(T^{\frac{\alpha+1}{2}}\right)$.*

*Proof.* At a high level, the Bob we construct will behave as if his midpoint were at $m_B - T^{\frac{\alpha-1}{2}}$. If Alice calls his bluff, i.e. cuts close to $m_B$ "enough" times, then Bob reverts to being honest by actually selecting his preferred piece.

Formally, let $y$ be an arbitrary point such that $y > m_A$. Define the Bob value density $\tilde{v}_B$ as follows:

$$\tilde{v}_B(x) = \begin{cases} \frac{1}{2y} & \forall x \in [0, y] \\ \frac{1}{2(1-y)} & \forall x \in (y, 1]. \end{cases} \qquad (17)$$

Bob's value density $\tilde{v}_B$ is bounded since $y$ is a fixed constant. Moreover, Bob's midpoint $m_B$ is exactly at $y$. Let $\tilde{S}_B$ be Bob's strategy as defined in Figure 16.

---

**Bob strategy $\tilde{S}_B$:**

**Input:** $\alpha$.
Initialize $c = 0$. For $t \in [T]$:

- If $m_B < a_t \leq 1$ then play $b_t = L$.

- If $0 \leq a_t < m_B - T^{\frac{\alpha-1}{2}}$ then play $b_t = R$.

- If $m_B - T^{\frac{\alpha-1}{2}} \leq a_t \leq m_B$ then:

  If $c \geq T^{\frac{\alpha+1}{2}}$ then play $b_t = R$; Else, play $b_t = L$.

  Update $c \leftarrow c + 1$.

Figure 16: Algorithm A.3

---

Strategy $\tilde{S}_B$ ensures that Bob takes his less-favorite piece at most $T^{\frac{\alpha+1}{2}}$ times, and this only happens when Alice cuts in the interval

$$P = \left[m_B - T^{\frac{\alpha-1}{2}}, m_B\right]. \qquad (18)$$

Since the density $\tilde{v}_B$ is bounded from above by a constant, Bob gives up a value of at most $O(T^{\frac{\alpha-1}{2}})$ in each such round. Thus Bob's regret is $O(T^\alpha)$.

Now let us compute Alice's regret with respect to her Stackelberg value. Suppose that $T$ is sufficiently large, so that $m_B - T^{\frac{\alpha-1}{2}} > m_A$. There are two cases depending on whether Alice triggers the switch in Bob's strategy or not.

- *Alice cuts in $P$ at least $T^{\frac{\alpha+1}{2}}$ times.* Consider the first $T^{\frac{\alpha+1}{2}}$ times Alice cuts in $P$. Whenever Alice cuts at some point $x \in P$, her utility will be $V_A([x,1])$ since Bob plays $L$. Therefore, her payoff in that round is maximized when $x$ is minimized, which occurs at the left-hand endpoint $x = m_B - T^{\frac{\alpha-1}{2}}$. However,

$$m_B - T^{\frac{\alpha-1}{2}} > m_A,$$

and so

$$V_A\left(\left[m_B - T^{\frac{\alpha-1}{2}}, 1\right]\right) < 1/2. \tag{19}$$

Then her Stackelberg regret over all the rounds in which she cuts in $P$ is at least

$$T^{\frac{\alpha+1}{2}} \cdot \left(V_A([0, m_B]) - V_A\left(\left[m_B - T^{\frac{\alpha-1}{2}}, 1\right]\right)\right)$$
$$> T^{\frac{\alpha+1}{2}} \cdot \left(V_A([0, m_A]) + V_A([m_A, m_B]) - \frac{1}{2}\right)$$
$$= T^{\frac{\alpha+1}{2}} V_A([m_A, m_B]) \in \Omega\left(T^{\frac{\alpha+1}{2}}\right).$$

- *Alice cuts in $P$ fewer than $T^{\frac{\alpha+1}{2}}$ times.* In this case, we will show that Alice's payoff per round cannot be more than $V_A\left(\left[0, m_B - T^{\frac{\alpha-1}{2}}\right]\right)$. To see this, we consider two sub-cases.

If Alice cuts at a point $x < m_B - T^{\frac{\alpha-1}{2}}$, then Bob picks R and Alice's utility in that round is

$$V_A([0, x]) < V_A\left(\left[0, m_B - T^{\frac{\alpha-1}{2}}\right]\right). \tag{20}$$

If Alice cuts at a point $x \geq m_B - T^{\frac{\alpha-1}{2}}$, then Bob picks L and Alice's utility in that round is

$$V_A([x, 1]) < \frac{1}{2} < V_A\left(\left[0, m_B - T^{\frac{\alpha-1}{2}}\right]\right). \tag{21}$$

Combining (20) and (21), we obtain that in each round $t \in [T]$, Alice's utility is

$$u_A^t \leq V_A\left(\left[0, m_B - T^{\frac{\alpha-1}{2}}\right]\right). \tag{22}$$

Summing over all rounds $t \in [T]$, we obtain that Alice's Stackelberg regret is

$$\sum_{t=1}^{T} \left(u_A^* - u_A^t\right) \geq \sum_{t=1}^{T} \left(V_A([0, m_B]) - V_A\left(\left[0, m_B - T^{\frac{\alpha-1}{2}}\right]\right)\right) \tag{23}$$
$$= \sum_{t=1}^{T} V_A\left(\left[m_B - T^{\frac{\alpha-1}{2}}, m_B\right]\right) \tag{24}$$
$$\geq \sum_{t=1}^{T} \delta \cdot T^{\frac{\alpha-1}{2}} \in \Omega\left(T^{\frac{\alpha+1}{2}}\right). \tag{25}$$

Thus in both cases, Alice's Stackelberg regret is at least $\Omega\left(T^{\frac{\alpha+1}{2}}\right)$, which concludes the proof. □

**Lemma 3.** *Let $\alpha, \beta \in [0, 1)$. Suppose Alice's density is $v_A$ and her strategy is $S_A$. There exists a Bob $\texttt{Bob}_1 = (v_{B,1}, S_{B,1})$ that depends on $v_A$ and a Bob $\texttt{Bob}_2 = (v_{B,2}, S_{B,2})$ that depends on $v_A$ and $S_A$, such that*

- *$\texttt{Bob}_1$ has regret $O(T^\alpha)$ (i.e. strategy $S_{B,1}$ ensures that a player with density $v_{B,1}$ has regret $O(T^\alpha)$)*

- *$\texttt{Bob}_2$ has regret $O(T^\beta)$; and*

- if $S_A$ ensures Alice Stackelberg regret of $O(T^\beta)$ against Bob$_1$, then $S_A$ has regret at least $T/6$ against Bob$_2$.

*Proof.* Let $x$ be the cake position such that $V_A([0, x]) = 2/3$, and let $y$ be the cake position such that $V_A([0, y]) = 5/6$. The first Bob, Bob$_1$, will have a valuation function $v_{B,1}$ that has midpoint $x$. We define Bob$_1$'s strategy $S_{B,1}$ so that it truthfully picks his preferred piece, *i.e.,*

$$S_{B,1}(A_t, B_{t-1}) = \begin{cases} L & \text{if } a_t \in (x, 1] \\ R & \text{if } a_t \in [0, x]. \end{cases} \tag{26}$$

Against Bob$_1$, Alice's Stackelberg value is $2/3$. Suppose $S_A$ ensures Alice Stackelberg regret at most $rT^\beta$ against Bob$_1$ for some $r > 0$.

Then we define our second Bob, denoted Bob$_2$, having a valuation function $v_{B,2}$ which has a midpoint at $y$. Let $k(t)$ be the number of times Alice cuts in the interval $(x, y]$ in rounds $\{1, \ldots, t\}$. Then Bob$_2$'s strategy will be defined as follows:

$$S_{B,2}(A_t, B_{t-1}) = \begin{cases} S_{B,1}(A_t, B_{t-1}) & \text{if } k(t) \leq 3rT^\beta \\ L & \text{if } k(t) > 3rT^\beta \text{ and } a_t \in (y, 1] \\ R & \text{if } k(t) > 3rT^\beta \text{ and } a_t \in [0, y]. \end{cases} \tag{27}$$

Intuitively, $S_{B,2}$ switches to being honest after Alice cuts in $(x, y]$ sufficiently many times. This transition gives Bob$_2$ a regret guarantee of $O(T^\beta)$.

When Alice plays $S_A$ against Bob$_1$, her total payoff is

$$u_A(S_A, S_{B,1}) \geq 2T/3 - rT^\beta. \tag{28}$$

However, every round she cuts in $(x, y]$, her payoff is less than $1/3$. Therefore, against Bob$_1$, her total payoff can also be bounded by

$$u_A(S_A, S_{B,1}) \leq 2T/3 - k(T)/3. \tag{29}$$

Combining inequalities (28) and (29), we obtain

$$k(T) \leq 3rT^\beta. \tag{30}$$

By definition, strategy $S_{B,2}$ behaves the same as strategy $S_{B,1}$ when $k(T) \leq 3rT^\beta$. By (30), we have $k(T) \leq 3rT^\beta$ when Alice uses strategy $S_A$. Thus, if Alice uses strategy $S_A$, then Bob$_1$ and Bob$_2$ behave exactly the same way. Therefore, Alice receives no more than $2/3$ per round against Bob$_2$, so her regret is at least $T/6 \in \Omega(T)$. $\square$

If Bob's density is not lower bounded by any constant, then Theorem 1 fails as shown in the next remark.

**Remark 1.** *Let $\alpha \in (0, 1)$. There exist value densities $v_A$ and $v_B$, where $v_A(x) \in [\delta, \Delta]\ \forall x \in [0, 1]$ and $v_B(x) \in (0, \Delta]\ \forall x \in [0, 1]$ such that Bob has a strategy $S_B$ which guarantees his regret is at most $T^\alpha$ and Alice's Stackelberg regret is at least $\Omega(T/\log T)$, no matter what strategy she uses.*

*Proof.* The specific valuations will be defined in terms of cumulative valuations. Let Alice's valuation be:

$$V_A([0, x]) = \begin{cases} \frac{1}{2}x & \text{if } x \in [0, 1/2] \\ \frac{3}{2}x - \frac{1}{2} & \text{if } x \in (1/2, 1] \end{cases}$$

Let Bob's valuation be:

$$V_B([0, x]) = \begin{cases} x & \text{if } x \in [0, 1/2] \\ \frac{1}{2} + 2^{-\frac{1}{2x-1}} & \text{if } x \in (1/2, 1] \end{cases} \tag{31}$$

Intuitively, Alice has a well-behaved piecewise uniform density with midpoint $m_A = 2/3$. Bob's density is well-behaved for $x \leq 1/2$, but the density rapidly approaches zero just to the right of $x = 1/2$.

Fix an arbitrary $\alpha \in (0,1)$. There exists a point $y$ on the cake such that $V_B([1/2, y]) = \frac{1}{2}T^{\alpha-1}$. Define Bob's strategy $S_B$ as follows:

$$S_B(A_t, B_{t-1}, v_B, T) = \begin{cases} R & \text{if } a_t < y \\ L & \text{if } a_t \geq y. \end{cases}$$

This strategy would be honest if Bob's midpoint were at $y$ rather than $1/2$. That means it only differs from his preferred piece when $a_t \in (1/2, y]$. The worst outcome for Bob occurs when $a_t = y$. But by construction, even in a round where Alice cuts at $y$, Bob only loses $T^{\alpha-1}$ utility compared to picking his preferred piece. Since there are $T$ rounds, his overall regret is at most $T^{\alpha}$.

Alice, on the other hand, cannot do very well compared to her Stackelberg value. Her Stackelberg value is $3/4$, achieved by cutting at Bob's midpoint of $1/2$ and receiving her preferred piece. But Bob prevents this payoff by pretending his midpoint is at $y$. To obtain the exact location of $y$, note that

$$V_B([1/2, y]) = \frac{1}{2}T^{\alpha-1}.$$

Plugging (31), this is equivalent to

$$2^{-\frac{1}{2y-1}} = \frac{1}{2}T^{\alpha-1}.$$

Solving for $y$, we have

$$y = \frac{1}{2} + \frac{1}{(2-2\alpha)\log_2 T + 2}.$$

For sufficiently large $T$, we have $y \in (m_B, m_A)$. Alice's best cut location in each round is then at $y$ itself, which gives her a per-round payoff of

$$V_A([y, 1]) = \frac{3}{4} - \frac{3}{(4-4\alpha)\log_2 T + 4}.$$

Adding this up over all $T$ rounds gives a total regret of at least, we obtain

$$\frac{3T}{(4-4\alpha)\log_2 T + 4},$$

which is $\Omega(T/\log T)$ as required. This completes the proof. $\square$

# B  Appendix: Equitable payoffs

This appendix has two parts. In Appendix B.1, we prove Theorem 2, which shows that Alice can enforce equitable payoffs. In Appendix B.2, we prove Theorem 3, which shows that Bob can enforce equitable payoffs.

## B.1  Appendix: Alice enforcing equitable payoffs

In this section we prove Theorem 2, the statement of which is included next.

**Restatement of Theorem 2** (Alice enforcing equitable payoffs; formal). *In both the sequential and simultaneous settings, Alice has a pure strategy $S_A$, such that for every Bob strategy $S_B$:*

- *on every trajectory of play, Alice's average payoff is at least $1/2 - o(1)$, while Bob's average payoff is at most $1/2 + o(1)$. More precisely, for all $t \in \{3, \ldots, T\}$:*

$$\frac{u_B(1,t)}{t} \leq \frac{1}{2} + \frac{5\Delta + 11}{\ln(2t/5)} \tag{32}$$

$$\frac{u_A(1,t)}{t} \geq \frac{1}{2} - \frac{4}{\sqrt{t-1}}, \tag{33}$$

*recalling that $\Delta$ is the upper bound on the players' value densities.*

*Moreover, even if Bob's value density is unbounded, his average payoff will still converge to $1/2$.*

The proof of the theorem is deferred until additional definitions have been stated and helpful lemmas have been proved. We first define some notations.

**Definition 5** (Set of valuations $\mathcal{W}_n$). *For each $n \in \mathbb{N}^*$, we define the following set of non-decreasing piecewise linear functions with $n$ pieces:*

$$\mathcal{W}_n = \Big\{ f : [0,1] \to [0,1] \mid f \text{ is non-decreasing with } f(0) = 0, f(1) = 1, f(i/n) \cdot n \in \mathbb{Z}_{\geq 0}, \text{ and}$$

$$f(x) = f\left(\frac{i}{n}\right) + \left(x - \frac{i}{n}\right)\left(f\left(\frac{i+1}{n}\right) - f\left(\frac{i}{n}\right)\right) \forall i \in \{0, \ldots, n-1\} \forall x \in \left[\frac{i}{n}, \frac{i+1}{n}\right] \Big\}.$$

**Definition 6** (Set of functions $\mathcal{V}_n$). *For each $n \in \mathbb{N}^*$, recall $\mathcal{W}_n$ was given in Definition 5 and define $\mathcal{V}_n$ as the following set of functions:*

$$\mathcal{V}_n = \begin{cases} \mathcal{W}_n & \text{if } n \neq 2 \\ \mathcal{W}_2 \cup \{f_A\} & \text{if } n = 2, \text{ where } f_A : [0,1] \to [0,1] \text{ is the function } f_A(x) = V_A([0,x]). \end{cases}$$

**Definition 7** (The set $\overline{\mathcal{V}}$). *Let $\overline{\mathcal{V}} = \bigcup_{n=1}^{\infty} \{(n, V) : V \in \mathcal{V}_n\}$, where $\mathcal{V}_n$ is given by Definition 6.*

**Remark 2.** *By construction, for each $n \in \mathbb{N}^*$, every function $f \in \mathcal{V}_n$ is non-decreasing.*

For each $n \in \mathbb{N}^*$, $\mathcal{W}_n$ contains the nondecreasing piecewise linear functions through a grid with spacing $1/n$. For large $n$, then, $\mathcal{W}_n$ should contain approximations to any given function accurate to roughly $O(1/n)$.

If Alice bases her strategy on limiting the payoff of the Bobs in each $\mathcal{W}_n$, then she will limit Bob's payoff but not necessarily guarantee herself a very good payoff. The inclusion of Alice's valuation function in $\overline{\mathcal{V}}$ rectifies this, allowing us to show a much tighter bound on Alice's payoff.

To formalize the ability of the elements of the $\mathcal{V}_n$ to approximate arbitrary valuation functions, we prove the following three lemmas. The first of them (Lemma 4) is somewhat technical, but most directly shows the richness of the $\mathcal{V}_n$. It will be used to prove Lemmas 5 and 6, which will be used directly to bound the payoff of an unbounded-density and bounded-density Bob, respectively.

**Lemma 4.** *Let $f : [0,1] \to [0,1]$ be continuous and increasing with $f(0) = 0$ and $f(1) = 1$. Suppose there exist $n \in \mathbb{N}^*$ and $\varepsilon \in (0, \infty)$ such that*

$$|f(x) - f(y)| \leq \varepsilon \qquad \forall x, y \in [0,1] \text{ with } |x - y| \leq 1/n. \tag{34}$$

*Then there exists $V_n \in \mathcal{V}_n$, where $\mathcal{V}_n$ is the set of functions from Definition 6, such that $|f(x) - V_n(x)| \leq \varepsilon + 2/n \ \forall x \in [0,1]$.*

*Proof.* For $x \in \mathbb{R}$, let $\lfloor x \rceil$ denote the nearest integer to $x$, breaking ties in favor of $\lceil x \rceil$ when $x = \lfloor x \rfloor + 1/2$.

Recall the set of functions $\mathcal{W}_n$ from Definition 5. Let $V_n$ be the function in $\mathcal{W}_n$ such that:

$$V_n(i/n) = \frac{\lfloor n \cdot f(i/n) \rceil}{n} \qquad \forall i \in [n-1]. \tag{35}$$

Then we claim the function $V_n$ approximates well the function $f$ at the points $i/n$, that is:

$$|V_n(i/n) - f(i/n)| = \left| \frac{\lfloor n \cdot f(i/n) \rceil}{n} - f(i/n) \right| \leq \frac{1}{2n} \qquad \forall i \in \{0, \ldots, n\}, \tag{36}$$

where

- for $i \in [n-1]$ the inequality in (36) follows from (35);
- for $i = 0$ it follows from $V_n(0) = 0 = f(0)$, and
- for $i = n$ it follows from $V_n(1) = 1 = f(1)$.

Let $x \in [0, 1]$. We show three inequalities next:

1. By inequality (34) from the lemma statement with parameters $\lfloor xn \rfloor / n$ and $\lceil xn \rceil / n$, we get

$$f\left( \frac{\lceil xn \rceil}{n} \right) - f\left( \frac{\lfloor xn \rfloor}{n} \right) \leq \varepsilon. \tag{37}$$

2. By inequality (36) with $i = \lceil xn \rceil$, we have

$$\left| V_n\left( \frac{\lceil xn \rceil}{n} \right) - f\left( \frac{\lceil xn \rceil}{n} \right) \right| \leq \frac{1}{2n}. \tag{38}$$

3. By inequality (36) with $i = \lfloor xn \rfloor$, we have

$$\left| f\left( \frac{\lfloor xn \rfloor}{n} \right) - V_n\left( \frac{\lfloor xn \rfloor}{n} \right) \right| \leq \frac{1}{2n}. \tag{39}$$

Summing up inequalities (37), (38), (39) and applying the triangle inequality, we obtain

$$\left| V_n\left( \frac{\lceil xn \rceil}{n} \right) - V_n\left( \frac{\lfloor xn \rfloor}{n} \right) \right| \leq \varepsilon + \frac{1}{n}. \tag{40}$$

We obtain

$$\begin{aligned}
V_n\left( \frac{\lfloor xn \rfloor}{n} \right) - \frac{1}{2n} &\leq f\left( \frac{\lfloor xn \rfloor}{n} \right) && \text{(By (36) with } i = \lfloor xn \rfloor) \\
&\leq f(x) && \text{(Since } f \text{ is non-decreasing)} \\
&\leq f\left( \frac{\lceil xn \rceil}{n} \right) && \text{(Since } f \text{ is non-decreasing)} \\
&\leq V_n\left( \frac{\lceil xn \rceil}{n} \right) + \frac{1}{2n}. && \text{(By (36) with } i = \lceil xn \rceil)
\end{aligned}$$

Denoting $J = \left[ V_n\left( \frac{\lfloor xn \rfloor}{n} \right) - 1/(2n), V_n\left( \frac{\lceil xn \rceil}{n} \right) + 1/(2n) \right]$, we conclude that $f(x) \in J$. Since the function $V_n$ is non-decreasing by Remark 2, we have

$$V_n\left( \frac{\lfloor xn \rfloor}{n} \right) - \frac{1}{2n} \leq V_n(x) - \frac{1}{2n} < V_n(x) < V_n(x) + \frac{1}{2n} \leq V_n\left( \frac{\lceil xn \rceil}{n} \right) + \frac{1}{2n}. \tag{41}$$

Thus $V_n(x) \in J$. Since both $f(x) \in J$ and $V_n(x) \in J$, we can bound $|f(x) - V_n(x)|$ as follows:

$$|f(x) - V_n(x)| \le |J|$$

$$= \left(V_n\left(\frac{\lceil xn \rceil}{n}\right) + \frac{1}{2n}\right) - \left(V_n\left(\frac{\lfloor xn \rfloor}{n}\right) - \frac{1}{2n}\right)$$

$$\le \varepsilon + \frac{2}{n}. \qquad \text{(Since } V_n\left(\frac{\lceil xn \rceil}{n}\right) - V_n\left(\frac{\lfloor xn \rfloor}{n}\right) \le \varepsilon + 1/n \text{ by (40))}$$

Since this holds for all $x \in [0, 1]$, the function $V_n$ required by the lemma exists. $\qquad \square$

**Lemma 5.** *Let $f : [0,1] \to [0,1]$ be continuous and increasing with $f(0) = 0$ and $f(1) = 1$. For each $\varepsilon > 0$, there exists $n \in \mathbb{N}^*$ and a function $V_n \in \mathcal{V}_n$ such that*

$$|V_n(x) - f(x)| \le \varepsilon \ \ \forall x \in [0, 1], \tag{42}$$

*recalling that the set of functions $\mathcal{V}_n$ is given by Definition 6.*

*Proof.* Let $\epsilon > 0$. Since $f$ is continuous and increasing, its inverse $f^{-1}$ is also continuous and increasing. Since $f(0) = 0$ and $f(1) = 1$, we have $f^{-1}(0) = 0$ and $f^{-1}(1) = 1$. Consider a function $g : [0, 1 - \varepsilon/4] \to [0, 1]$ defined as

$$g(x) = f^{-1}(x + \varepsilon/4) - f^{-1}(x). \tag{43}$$

Since $f^{-1}$ is continuous and bounded over a closed interval, so is the function $g$. Therefore, by the extreme value theorem, $g$ attains a global minimum value $\delta^*$. Since $f^{-1}$ is strictly increasing, $g$ is never zero, so

$$\delta^* > 0. \tag{44}$$

Since we will analyze $g(f(x))$, it will be useful to define the set of values $x$ for which $g(f(x))$ is well defined, that is:

$$S_\varepsilon = \{x \in [0, 1] \mid f(x) \in [0, 1 - \varepsilon/4]\}. \tag{45}$$

Since $\delta^*$ is a global minimum of $g$, we have

$$g(f(x)) \ge \delta^* \qquad \forall x \in S_\varepsilon. \tag{46}$$

Using the definition of $g$ (equation (43)) in (46) yields

$$g(f(x)) = f^{-1}(f(x) + \varepsilon/4) - f^{-1}(f(x)) \ge \delta^* \qquad \forall x \in S_\varepsilon. \tag{47}$$

Since $f^{-1}(f(x)) = x$, inequality (47) yields $f^{-1}(f(x) + \varepsilon/4) - x \ge \delta^*$, or equivalently,

$$f^{-1}(f(x) + \varepsilon/4) \ge x + \delta^* \qquad \forall x \in S_\varepsilon. \tag{48}$$

Applying $f$ to both sides of (48) and using monotonicity of $f$, we obtain $f(x) + \varepsilon/4 \ge f(x + \delta^*)$ for all $x \in S_\varepsilon$, or equivalently

$$f(x + \delta^*) - f(x) \le \varepsilon/4 \qquad \forall x \in S_\varepsilon. \tag{49}$$

Since $\varepsilon > 0$ and $\delta^* > 0$ by (44), there exists $n \in \mathbb{N}$ such that

$$n > 1/\delta^* \ \text{ and } \ n > 4/\varepsilon. \tag{50}$$

Note that because the range of $f^{-1}$ is $[0, 1]$, the left side of (48) is at most 1. Therefore, from (48) and the fact that $\delta^* > 1/n$ by (50):

$$1 \ge x + \delta^* > x + 1/n \qquad \forall x \in S_\varepsilon \tag{51}$$

Consider an arbitrary $x, y \in [0, 1]$ with $x \le y \le x + 1/n$. We consider two cases:

- Case $x \in S_\varepsilon$: Since $f$ is strictly increasing and $x + \delta^* \le 1$ by (51), we have:
$$f(y) \le f(x + 1/n) < f(x + \delta^*) \tag{52}$$

Subtracting $f(x)$ from both sides of (52) and applying (49), we obtain

$$f(y) - f(x) < \varepsilon/4 \tag{53}$$

- Case $x \notin S_\varepsilon$: Then $f(x) > 1 - \varepsilon/4$. Since $f(y) \leq 1$, we have:

$$f(y) - f(x) < 1 - (1 - \varepsilon/4) = \varepsilon/4. \tag{54}$$

Combining cases $x \in S_\varepsilon$ and $x \notin S_\varepsilon$ gives $f(y) - f(x) < \varepsilon/4 \ \forall x, y \in [0,1]$ with $x \leq y \leq x + 1/n$. Since $f$ is strictly increasing, we have $f(x) \leq f(y)$ when $x \leq y$, and so

$$|f(x) - f(y)| < \varepsilon/4 \qquad \forall x, y \in [0,1] \text{ with } |x - y| \leq 1/n. \tag{55}$$

By Lemma 4, there exists a function $V_n \in \mathcal{V}_n$ such that:

$$|f(x) - V_n(x)| \leq \varepsilon/4 + 2/n \qquad \forall x \in [0,1]. \tag{56}$$

Since $n > 4/\varepsilon$ by (50), inequality (56) gives

$$|f(x) - V_n(x)| \leq \varepsilon/4 + 2/n < \varepsilon/4 + \varepsilon/2 < \varepsilon \qquad \forall x \in [0,1]. \tag{57}$$

This completes the proof.

$\square$

**Lemma 6.** *Suppose $v_B$ is a value density function for Bob with $v_B(x) \leq \Delta$ for some $\Delta > 0$ and all $x \in [0,1]$. Then for all $n \in \mathbb{N}^*$, there exists a function $V_n \in \mathcal{V}_n$ such that*

$$\left| V_n(x) - V_B([0,x]) \right| \leq \frac{\Delta + 2}{n} \qquad \forall x \in [0,1].$$

*Proof.* Let $n \in \mathbb{N}^*$. Since Bob's density is upper bounded by $\Delta$, we have

$$|V_B([0,x]) - V_B([0,y])| \leq \Delta |x - y| \qquad \forall x, y \in [0,1]. \tag{58}$$

When $|x - y| \leq 1/n$, we get $|V_B([0,x]) - V_B([0,y])| \leq \Delta |x - y| \leq D/n$.

By Lemma 4 applied to the function $f : [0,1] \to [0,1]$ given by $f(x) = V_B([0,x])$, there exists $V_n \in \mathcal{V}_n$ with $|V_n(x) - V_B([0,x])| \leq \Delta/n + 2/n$ for all $x \in [0,1]$. This completes the proof. $\square$

The set of functions $\mathcal{V}_n$ will be used to construct a strategy for Alice. Next we bound the size of $\mathcal{V}_n$ as a function of $n$, as this rate of growth will influence the error bounds on the players' utilities.

**Lemma 7.** $|\mathcal{V}_n| \leq 4^{n-1} \ \forall n \in \mathbb{N}^*$.

*Proof.* We first estimate the size of each set $\mathcal{W}_n$, and then will infer the bound for the size of $\mathcal{V}_n$. Consider the density function corresponding to any particular $V \in \mathcal{W}_n$. Because $V$ is piecewise linear, its density is piecewise constant. Each $V$ is then uniquely determined by a sequence $d_1, d_2, \ldots, d_n$, where $d_i$ is the value density between $\frac{i-1}{n}$ and $\frac{i}{n}$. Each $d_i$ must be a non-negative integer, because each of these intervals has width $1/n$ and sees $V$ rise by an integer multiple of $1/n$. More strongly, because $V(0) = 0$ and $V(1) = 1$, we must have the next relation between the $d_i$'s:

$$\sum_{i=1}^{n} \frac{d_i}{n} = 1 \iff \sum_{i=1}^{n} d_i = n. \tag{59}$$

Thus, the size $|\mathcal{W}_n|$ is the number of possible partitions of $n$ into $n$ parts with nonnegative integer sizes. The size of $\mathcal{W}_n$ can then be counted by a standard combinatorics technique. Represent each choice of $d_1, d_2, \ldots, d_n$ with a sequence of $n$ "stars" and $n - 1$ "bars": $d_1$ stars, then a bar, then $d_2$ stars, then another bar, and so on. Each choice of $d_1, \ldots, d_n$ corresponds to a unique arrangement of stars and bars. The opposite is also true: given an arrangement, the number of stars between each pair of bars can be read off as $d_1, \ldots, d_n$. The size of $\mathcal{V}_n$ is then the number of arrangements, which is $\binom{2n-1}{n}$.

The upper bound can then be shown by induction. As a base case, $\binom{2 \cdot 1 - 1}{1} = \binom{1}{1} = 1 \leq 4^{1-1}$.

Now assume $\binom{2n-1}{n} \leq 4^{n-1}$ for an arbitrary $n \geq 1$. We have

$$\binom{2(n+1)-1}{n+1} = \frac{(2n+1)!}{(n+1)! \cdot n!} = \frac{(2n-1)! \cdot 2n \cdot (2n+1)}{n! \cdot (n-1)! \cdot n(n+1)} = \binom{2n-1}{n} \cdot \frac{2n(2n+1)}{n(n+1)}$$

$$\leq 4^{n-1} \cdot 4\frac{n+\frac{1}{2}}{n+1} \qquad \text{(By the inductive hypothesis)}$$

$$\leq 4^{n-1} \cdot 4 = 4^{(n+1)-1}$$

So by induction, the bound holds for all $n \geq 1$.

Because $\mathcal{V}_n = \mathcal{W}_n$ for all $n \neq 2$, the only case left to verify is $n = 2$. By calculation, $|\mathcal{W}_2| = \binom{3}{2} = 3$, so including the extra function makes $|\mathcal{V}_2| = 4 = 4^{2-1}$. $\qquad\square$

Some other technical lemmas are necessary. The first two relate to the following function, which acts like an infinite-dimensional inner product.

**Definition 8.** *Let $\mathcal{X}$ be the collection of all functions $X : \overline{\mathcal{V}} \to [-1,1]$. For each pair of functions $X, Y : \overline{\mathcal{V}} \to [-1,1]$, define $P : \mathcal{X} \times \mathcal{X} \to \mathbb{R}$ as follows:*

$$P(X,Y) = \sum_{n=1}^{\infty} \frac{1}{2^n |\mathcal{V}_n|} \sum_{V \in \mathcal{V}_n} X(n,V)Y(n,V) \,.$$

**Lemma 8.** *The function $P$ has the properties of an inner product. In particular, for all functions $X, Y, Z : \overline{\mathcal{V}} \to [-1,1]$ and all constants $c \in [-1,1]$, the following holds:*

*(1) $P(X,Y)$ exists, and the infinite sum converges absolutely*

*(2) $P(X,Y) = P(Y,X)$*

*(3) $|P(X,Y)| \leq 1$*

*(4) $P(cX,Y) = cP(X,Y)$*

*(5) $P(X+Z,Y) = P(X,Y) + P(Z,Y)$, assuming $X+Z$ is in the domain of $P$*

*(6) $P(X,X) = 0$ if $X(n,V) = 0$ for all $n$ and all $V$, and $P(X,X) > 0$ otherwise*

*Proof.* We separately prove each of the properties.

For the first property, note that the individual terms go to zero

$$\lim_{n\to\infty} \left| \frac{1}{2^n |\mathcal{V}_n|} \sum_{V \in \mathcal{V}_n} X(n,V)Y(n,V) \right| \leq \lim_{n\to\infty} \frac{1}{2^n |\mathcal{V}_n|} \sum_{V \in \mathcal{V}_n} |X(n,V)Y(n,V)|$$

$$\leq \lim_{n\to\infty} \frac{1}{2^n |\mathcal{V}_n|} \sum_{V \in \mathcal{V}_n} 1 \qquad \text{(By the bounds on $X$ and $Y$)}$$

$$= \lim_{n\to\infty} \frac{1}{2^n |\mathcal{V}_n|} |\mathcal{V}_n| = \lim_{n\to\infty} \frac{1}{2^n}$$

$$= 0 \,.$$

Using the above upper bound for individual terms, the sum of the absolute values does not diverge as follows

$$\sum_{n=1}^{\infty} \left| \frac{1}{2^n |\mathcal{V}_n|} \sum_{V \in \mathcal{V}_n} X(n,V)Y(n,V) \right| \leq \sum_{n=1}^{\infty} \frac{1}{2^n} = 1 \,.$$

So $P(X,Y)$ exists and the infinite sum converges absolutely.

The existence is enough for property (2), which follows directly from the product $X(n,V)Y(n,V)$ being commutative. This calculation directly verifies property (3).

The absolute convergence allows for the linear operations in properties (4) and (5) to factor through the sum.

First, for property (4) we obtain

$$P(cX, Y) = \sum_{n=1}^{\infty} \frac{1}{2^n |\mathcal{V}_n|} \sum_{V \in \mathcal{V}_n} cX(n, V)Y(n, V)$$

$$= c \sum_{n=1}^{\infty} \frac{1}{2^n |\mathcal{V}_n|} \sum_{V \in \mathcal{V}_n} X(n, V)Y(n, V)$$

$$= cP(A, B).$$

For property (5) observe that:

$$P(X + Z, Y) = \sum_{n=1}^{\infty} \frac{1}{2^n |\mathcal{V}_n|} \sum_{V \in \mathcal{V}_n} (X(n, V) + Z(n, V))Y(n, V)$$

$$= \sum_{n=1}^{\infty} \frac{1}{2^n |\mathcal{V}_n|} \sum_{V \in \mathcal{V}_n} X(n, V)Y(n, V) + Z(n, V)Y(n, V)$$

$$= \sum_{n=1}^{\infty} \frac{1}{2^n |\mathcal{V}_n|} \sum_{V \in \mathcal{V}_n} X(n, V)Y(n, V) + \sum_{n=1}^{\infty} \frac{1}{2^n |\mathcal{V}_n|} \sum_{V \in \mathcal{V}_n} Z(n, V)Y(n, V)$$

$$= P(X, Y) + P(Z, Y).$$

For property (6), observe that the expression can be rewritten as:

$$P(X, X) = \sum_{n=1}^{\infty} \frac{1}{2^n |\mathcal{V}_n|} \sum_{V \in \mathcal{V}_n} X(n, V)^2.$$

This is a sum of squares, which will be zero if all the included $X(n, V)$ are zero and positive otherwise. $\square$

**Lemma 9.** *For each $x \in [0, 1]$, let $G_x : \overline{\mathcal{V}} \to [-1/2, -1/2]$ be given by $G_x(n, V) = V(x) - \frac{1}{2}$. Then, for all $Z : \overline{\mathcal{V}} \to [-1, 1]$, the following function is continuous in $x$:*

$$P(G_x, Z) = \sum_{n=1}^{\infty} \frac{1}{2^n |\mathcal{V}_n|} \sum_{V \in \mathcal{V}_n} \left(V(x) - \frac{1}{2}\right) Z(n, V).$$

*Proof.* Alice's valuation function in $\mathcal{V}_2$ needs to be handled separately. Accordingly, write:

$$P(G_x, Z) = \sum_{n=1}^{\infty} \frac{1}{2^n |\mathcal{V}_n|} \sum_{V \in V_n} \left(V(x) - \frac{1}{2}\right) Z(n, V)$$

$$= \frac{1}{2^2 |\mathcal{V}_2|} \left(V_A([0, x]) - \frac{1}{2}\right) Z(n, V_A) + \sum_{n=1}^{\infty} \frac{1}{2^n |\mathcal{V}_n|} \sum_{V \in \mathcal{W}_n} \left(V(x) - \frac{1}{2}\right) Z(n, V).$$

As long as each of these two parts is continuous, the sum will be too. The first part is continuous because $V_A([0, x])$ is continuous. To prove the second part, for notational simplicity, define

$$P'(G_x, Z) = \sum_{n=1}^{\infty} \frac{1}{2^n |\mathcal{V}_n|} \sum_{V \in \mathcal{W}_n} \left(V(x) - \frac{1}{2}\right) Z(n, V)$$

The continuity of $P'$ will be shown by directly appealing to the definition of a limit. First, note that each $V \in \mathcal{W}_n$ is made of $n$ linear segments, each of width $\frac{1}{n}$ and height at most 1. Therefore, if $V \in \mathcal{W}_n$, for all $x, y \in [0, 1]$:

$$|V(y) - V(x)| \leq |y - x|n \tag{60}$$

Fix an arbitrary $\varepsilon > 0$ and $x \in [0, 1]$. Choose $\delta = \varepsilon/2$. For any $y$ such that $|y - x| < \delta$:

$$
\begin{aligned}
|P'(G_y, Z) - P'(G_x, Z)| &= |P'(G_y - G_x, Z)| && \text{(By property (5))} \\
&= \left| \sum_{n=1}^{\infty} \frac{1}{2^n |\mathcal{V}_n|} \sum_{V \in \mathcal{W}_n} (V(y) - V(x)) Z(n, V) \right| \\
&\leq \sum_{n=1}^{\infty} \frac{1}{2^n |\mathcal{V}_n|} \sum_{V \in \mathcal{W}_n} |V(y) - V(x)| \cdot |Z(n, V)| && \text{(Triangle inequality)} \\
&\leq \sum_{n=1}^{\infty} \frac{1}{2^n |\mathcal{V}_n|} \sum_{V \in \mathcal{W}_n} n|y - x| \cdot 1 && \text{(By ineq (60) and } Z \leq 1) \\
&\leq |y - x| \sum_{n=1}^{\infty} \frac{n}{2^n} && \text{(Since } |\mathcal{W}_n| \leq |\mathcal{V}_n|) \\
&= |y - x| \cdot 2 \\
&< 2\delta && \text{(By choice of } \delta) \\
&= \varepsilon \, .
\end{aligned}
$$

Therefore, for all $x \in [0, 1]$:

$$
\lim_{y \to x} P'(G_y, Z) = P'(G_x, Z),
$$

which is the definition of $P'(G_x, Z)$ being continuous in $x$. This concludes that $P(G_x, Z)$ is continuous in $x$. □

Finally, the last one is a strengthening of Lemma 1 from Blackwell (1956), under stronger hypotheses.

**Lemma 10.** *Suppose a sequence of nonnegative values $\delta_1, \delta_2, \ldots$ satisfies, for all $t \geq 1$:*

$$
\delta_{t+1} \leq \frac{1}{(t+1)^2} + \left( \frac{t-1}{t+1} \right) \delta_t \, .
$$

*Then $\lim_{t \to \infty} \delta_t = 0$. In particular, for all $t \geq 2$:*

$$
\delta_t \leq \frac{1}{t(t-1)} \sum_{i=2}^{t} \frac{i-1}{i} \leq \frac{1}{t} \, .
$$

*Proof.* The latter bound will be shown by induction. As a base case, consider $t = 2$. The condition for $t = 1$ gives:

$$
\delta_2 \leq \frac{1}{(1+1)^2} + \left( \frac{1-1}{1+1} \right) \delta_1 = \frac{1}{4} = \frac{1}{2(2-1)} \sum_{i=2}^{2} \frac{i-1}{i} \, . \tag{61}
$$

So the base case holds. Now suppose the conclusion holds for some $t \geq 2$. Assume that the claim holds for $\delta_t$. We can expand the upper bound of $\delta_{t+1}$ as follows.

$$
\delta_{t+1} \leq \frac{1}{(t+1)^2} + \left( \frac{t-1}{t+1} \right) \delta_t \tag{62}
$$

$$
\leq \frac{1}{(t+1)^2} + \left( \frac{t-1}{t+1} \right) \frac{1}{t(t-1)} \sum_{i=2}^{t} \frac{i-1}{i} \qquad \text{(By the induction hypothesis)}
$$

$$
= \frac{1}{(t+1)^2} + \frac{1}{t(t+1)} \sum_{i=2}^{t} \frac{i-1}{i} \tag{63}
$$

$$
= \frac{1}{(t+1)t} \sum_{i=2}^{t+1} \frac{i-1}{i} \, . \tag{64}
$$

Now it suffices to show that the last term in (64) is upper bounded by $1/t$, which follows from the following inequality:

$$\frac{1}{t(t-1)} \sum_{i=2}^{t} \frac{i-1}{i} \leq \frac{1}{t(t-1)} \sum_{i=2}^{t} 1 = \frac{1}{t(t-1)} \cdot (t-1) = \frac{1}{t}. \tag{65}$$

Thus $\delta_{t+1} \leq 1/t$, which completes the proof. $\qquad\square$

Now the groundwork is laid to prove Theorem 2. The idea is to limit every strategy and every valuation in every $\mathcal{V}_n$ to an average payoff of $1/2$, which is sufficient to limit Bob equipped with arbitrary valuation as well.

*Proof of Theorem 2.* We first define the Alice's strategy in an analytical manner, and then prove that it is well-defined. Then we will show that this strategy will force Bob's payoff to be at most $1/2$ on average, by showing both that Bob's valuation can be well-approximated by functions in $\overline{\mathcal{V}}$ and that such valuation functions are limited to a payoff of $1/2$ on average. Finally, we establish explicit convergence rates for Bob's payoff if his valuation is bounded, as well as the convergence of Alice's payoff to $1/2$.

**Defining Alice's strategy** $S_A$   Consider Alice's decision in round $T$. In each round $t < T$, let the payoff to Bob whose cumulative valuation function is $V$ be $u_{t,V}$. For each $x \in [0,1]$, let $G_x : \overline{\mathcal{V}} \to [-1,1]$ be a function defined as $G_x(n, V) = V(x) - 1/2$. Let $U_t(n, V) = u_{t,V} - 1/2$ for $t = 1, \ldots, T-1$, and let $\overline{U}_t(n, V) = \sum_{i=1}^{t} U_i(n, V)/t$. Let $W_t(n, V) = \max\{0, \overline{U}_t(n, V)\}$ for $t = 1, \ldots, T-1$. In round $T$, Alice's strategy $S_A$ is to cut at a point $x$ such that $P(G_x, W_{T-1}) = 0$.

We then show that $S_A$ is well-defined. It suffices to prove that such an $x$ always exists. To this end, observe that

$$\begin{aligned}
P(G_0, W_{T-1}) &= \sum_{n=1}^{\infty} \frac{1}{2^n |\mathcal{V}_n|} \sum_{V \in \mathcal{V}_n} \left( V(0) - \frac{1}{2} \right) W_{T-1}(n, V) \\
&= \sum_{n=1}^{\infty} \frac{1}{2^n |\mathcal{V}_n|} \sum_{V \in \mathcal{V}_n} -\frac{1}{2} W_{T-1}(n, V) \\
&\leq \sum_{n=1}^{\infty} \frac{1}{2^n |\mathcal{V}_n|} \sum_{V \in \mathcal{V}_n} 0 \qquad \text{(Since } W_{T-1}(\cdot) \text{ is nonnegative)} \\
&= 0.
\end{aligned}$$

Using similar algebra for $P(G_1, W_{T-1})$, we obtain

$$\begin{aligned}
P(G_1, W_{T-1}) &= \sum_{n=1}^{\infty} \frac{1}{2^n |\mathcal{V}_n|} \sum_{V \in \mathcal{V}_n} \left( V(1) - \frac{1}{2} \right) W_{T-1}(n, V) \\
&= \sum_{n=1}^{\infty} \frac{1}{2^n |\mathcal{V}_n|} \sum_{V \in \mathcal{V}_n} \frac{1}{2} W_{T-1}(n, V) \\
&\geq \sum_{n=1}^{\infty} \frac{1}{2^n |\mathcal{V}_n|} \sum_{V \in \mathcal{V}_n} 0 \qquad \text{(Since } W_{T-1}(\cdot) \text{ is nonnegative)} \\
&= 0
\end{aligned}$$

By Lemma 9, $P(G_x, W_{T-1})$ is continuous in $x$, so by the Intermediate Value Theorem there exists $x$ such that $P(G_x, W_{T-1}) = 0$.

**Bounding Bob's payoff**   Define

$$\mathcal{S} = \left\{ X : -\frac{1}{2} \leq X(n, V) \leq 0 \text{ for all } (n, V) \in \overline{\mathcal{V}} \right\}.$$

For each round $t$, let $\delta_t$ be the distance from $\mathcal{S}$ to $\overline{U}_t$, defined as:

$$\delta_t = \inf_{X \in S} P(\overline{U}_t - X, \overline{U}_t - X). \tag{66}$$

For $t < T$, let $Y_t : \overline{\mathcal{V}} \to [-1/2, 1/2]$ be the function defined by:

$$Y_t(n, V) = \min\{0, \overline{U}_t(n, V)\}$$

By Claim 3, we have $\delta_t \leq 1/t$ for all $t \geq 2$.

Now we will show that this strategy guarantees that, for any Bob's valuation function $v_B$ and any strategy $S_B$ Bob employs, his average payoff is at most $1/2$. Consider an arbitrary $\varepsilon > 0$. It will be shown that, for some sufficiently large $T$, Bob's average payoff up to any point after round $T$ is at most $1/2 + \varepsilon$.

In the general case, this convergence can be established from Lemma 5. Consider $N, T \in \mathbb{N}^*$ such that

- $N$ is such that there exists $V' \in \mathcal{V}_N$ with $|V'(x) - V_B([0, x])| < \varepsilon/2$ for all $x \in [0, 1]$.

- $T$ is sufficiently large so that

$$\delta_t < \frac{\varepsilon^2}{4 \cdot 2^N |\mathcal{V}_N|} \qquad \forall t \geq T. \tag{67}$$

For example, taking $T = \lceil \frac{4 \cdot 2^N |\mathcal{V}_N|}{\varepsilon^2} \rceil$ would suffice.

We will show that for each $t \geq T$, we have $\overline{U}_t(N, V') < \varepsilon/2$. This trivially holds if $\overline{U}_t(N, V') \leq 0$, so assume $\overline{U}_t(N, V') > 0$. We then obtain

$$\overline{U}_t(N, V') = \sqrt{2^N |\mathcal{V}_N| \cdot \frac{1}{2^N |\mathcal{V}_N|} \overline{U}_t(N, V')^2}$$

$$= \sqrt{2^N |\mathcal{V}_N| \cdot \frac{1}{2^N |\mathcal{V}_N|} \left(\overline{U}_t(N, V') - Y_t(N, V')\right)^2}, \tag{68}$$

where (68) follows from the fact that $Y_t(N, V') = \min\{0, \overline{U}_t(N, V')\} = 0$. Using (68), we can upper bound $\overline{U}_t(N, V')$ as follows:

$$\overline{U}_t(N, V') \leq \sqrt{2^N |\mathcal{V}_N| \sum_{n=1}^{\infty} \frac{1}{2^n |\mathcal{V}_n|} \sum_{V \in \mathcal{V}_n} \left(\overline{U}_t(n, V) - Y_t(n, V)\right)^2}$$

$$= \sqrt{2^N |\mathcal{V}_N| P(\overline{U}_t - Y_t, \overline{U}_t - Y_t)}$$

$$= \sqrt{2^N |\mathcal{V}_N| \delta_t} \tag{69}$$

$$< \sqrt{2^N |\mathcal{V}_N| \cdot \frac{\varepsilon^2}{4 \cdot 2^N |\mathcal{V}_N|}} = \frac{\varepsilon}{2}. \qquad \text{(By (67))}$$

Therefore, we have $\overline{U}_t(N, V') < \varepsilon/2$ for all $t \geq T$. Translating back into payoffs, this gives a Bob whose cumulative valuation function is $V'$ an average payoff of less than $1/2 + \varepsilon/2$. By the choice of $V'$, we have

$$|V'(x) - V_B([0, x])| < \varepsilon/2 \qquad \text{for all } x \in [0, 1]. \tag{70}$$

so the average payoff to a Bob whose valuation function is $v_B$ is less than $1/2 + \varepsilon$. Since this construction works for all $\varepsilon > 0$, Bob's average payoff satisfies the following inequality as required:

$$\frac{u_B(1, t)}{t} \leq \frac{1}{2} + o(1).$$

**Explicit bounds** It remains to prove inequality (32) when Bob's density is upper bounded by $\Delta$, and prove Alice's explicit payoff in inequality (33). Choose an integer $N$ large enough that

$$\frac{\Delta + 2}{N} < \frac{\varepsilon}{2}.$$

For instance, taking $N = \lceil 2(\Delta + 2)/\varepsilon \rceil$ is sufficient. By Lemma 6, there exists $V' \in \mathcal{V}_N$ such that

$$|V'(x) - V_B([0, x])| < (\Delta + 2)/N \quad \text{for all } x \in [0, 1].$$

Choose $T$ in exactly the same way as the general case, i.e. $T = \lceil 4 \cdot 2^N |\mathcal{V}_N|/\varepsilon^2 \rceil$. By exactly the same algebra as the general case, we can conclude that $\overline{U}_t(N, V') < \varepsilon/2$ for all $t \geq T$. By the choice of $V'$, we have

$$|V'(x) - V_B([0, x])| < \varepsilon/2 \qquad \text{for all } x \in [0, 1], \tag{71}$$

so the average payoff to Bob is at most $\frac{1}{2} + \varepsilon$.

In this case, Bob's average payoff will be within $\varepsilon$ of $1/2$ by time

$$T_\varepsilon = \left\lceil \frac{4 \cdot 2^N |\mathcal{V}_N|}{\varepsilon^2} \right\rceil, \qquad \text{where} \quad N = \lceil 2(\Delta + 2)/\varepsilon \rceil. \tag{72}$$

Hence, for such $T_\varepsilon$, we have

$$T_\varepsilon \leq \left\lceil \frac{4 \cdot 2^N \cdot 4^{N-1}}{\varepsilon^2} \right\rceil.$$

Since $\lceil x \rceil \leq 2x$ for $x \geq 1/2$, this implies

$$T_\varepsilon \leq 2 \frac{8^N}{\varepsilon^2}.$$

Due to our choice of $N$, we can further obtain

$$T_\varepsilon \leq 2 \cdot \frac{8^{\frac{2(\Delta+2)}{\varepsilon}+1}}{\varepsilon^2}. \tag{73}$$

Taking the natural logarithm of both sides in (73), we have

$$\ln(T_\varepsilon) \leq \ln(16) + \frac{2(\Delta + 2)}{\varepsilon} \ln(8) + 2 \ln\left(\frac{1}{\varepsilon}\right).$$

Using the fact that $\ln(x) \leq x - 1$ for every $x > 0$, it follows that

$$\ln(T_\varepsilon) \leq \ln(16) + \frac{2(\Delta + 2)}{\varepsilon} \ln(8) + 2 \left(\frac{1}{\varepsilon} - 1\right). \tag{74}$$

Rearranging (74), we finally obtain

$$\varepsilon \leq \frac{2 \ln(8)(\Delta + 2) + 2}{\ln(T_\varepsilon) - \ln(16) + 2}.$$

Note that this requires the step of dividing by $\ln(T_\varepsilon) - \ln(16) + 2$ and it needs this quantity to be positive, which is indeed positive for $T \geq 3$. Rounding the terms gives the desired regret bound for Bob.

To provide Alice's payoff bound in inequality (33), consider a hypothetical Bob whose valuation function $\tilde{v}_B$ is exactly the same as Alice's valuation function $v_A$. For all rounds $\tau$, the payoff $\tilde{u}_B^\tau$ to this Bob then satisfies:

$$u_A^\tau + \tilde{u}_B^\tau = V_A([0, a_\tau]) + V_A([a_\tau, 1])$$
$$\text{(One player gets } V_A([0, a_\tau]) \text{ and the other gets } V_A([a_\tau, 1]))$$
$$= 1 \tag{75}$$

For any $t$, summing $u_A^\tau + \tilde{u}_B^\tau$ over $\tau \in [t]$ and dividing by $t$ then gives:

$$\frac{u_A(1, t)}{t} + \frac{\tilde{u}_B(1, t)}{t} = 1 \tag{76}$$

So it suffices to upper-bound this particular Bob's payoff to lower-bound Alice's payoff.

Choose an arbitrary $\varepsilon > 0$. Let $T_\varepsilon = \left\lceil \frac{2^2|\mathcal{V}_2|}{\varepsilon^2} \right\rceil$, ensuring $\delta_t < \frac{\varepsilon^2}{2^2|\mathcal{V}_2|}$ for all $t \geq T_\varepsilon$. By construction, Alice's valuation function $V_A([0, x]) \in \mathcal{V}_2$. Taking $N = 2$ and $V' = V_A$, we can then use identical algebra as the general-Bob case up to (69) to conclude that, for $t \geq T_\varepsilon$:

$$\overline{U}_t(2, V_A) \leq \sqrt{2^2|\mathcal{V}_2|\delta_t} \qquad\qquad \text{(Copying over (69))}$$

$$< \sqrt{2^2|\mathcal{V}_2| \cdot \frac{\varepsilon^2}{2^2|\mathcal{V}_2|}} = \varepsilon \qquad\qquad (77)$$

So this Bob's payoff is upper-bounded by $1/2 + \varepsilon$, and so by (76) Alice's payoff is lower-bounded by $1/2 - \varepsilon$. To solve for $\varepsilon$, observe that $T_\varepsilon \leq 16/\varepsilon^2 + 1$. Solving this bound on $T_\varepsilon$ for $\varepsilon$ gives

$$\varepsilon \leq \frac{4}{\sqrt{T_\varepsilon - 1}},$$

and it finishes the proof. $\qquad\qquad\qquad\qquad\qquad\qquad\qquad\qquad\qquad\qquad\qquad\qquad\square$

We finally provide the claims and their proofs used throughout the main proof.

**Claim 1.** *In the setting of Theorem 2, if $\overline{U}_t \notin \mathcal{S}$, then $\operatorname{argmin}_{X \in \mathcal{S}} P(\overline{U}_t - X, \overline{U}_t - X)$ exists. Furthermore:*

$$\operatorname*{argmin}_{X \in \mathcal{S}} P(\overline{U}_t - X, \overline{U}_t - X) = Y_t = \min\{0, \overline{U}_t(n, V)\}$$

*Proof of the claim.* Let $Y_t = \min\{0, \overline{U}_t(n, V)\} \in \mathcal{S}$. For any $X \in \mathcal{S}$,

$$P(\overline{U}_t - X, \overline{U}_t - X) = \sum_{n=1}^\infty \frac{1}{2^n|\mathcal{V}_n|} \sum_{V \in \mathcal{V}_n} \left(\overline{U}_t(n, V) - X(n, V)\right)^2$$

$$= \sum_{n=1}^\infty \frac{1}{2^n|\mathcal{V}_n|} \left( \sum_{\substack{V \in V_n \\ \overline{U}_t(n,V) > 0}} \left(\overline{U}_t(n, V) - X(n, V)\right)^2 + \sum_{\substack{V \in V_n \\ \overline{U}_t(n,V) \leq 0}} \left(\overline{U}_t(n, V) - X(n, V)\right)^2 \right)$$

$$(78)$$

Since $X \in \mathcal{S}$, we have that $X(n, V) \leq 0$ for every $n$ and $V$, due to our construction of S.

Therefore, replacing them with 0 brings them closer to any positive value and so decreases the first sum of squares. The second sum of squares is nonnegative, so it can be reduced by replacing it with 0. Applying these simplifications, we obtain

$$\sum_{n=1}^\infty \frac{1}{2^n|\mathcal{V}_n|} \left( \sum_{\substack{V \in V_n \\ \overline{U}_t(n,V) > 0}} \left(\overline{U}_t(n, V) - X(n, V)\right)^2 + \sum_{\substack{V \in V_n \\ \overline{U}_t(n,V) \leq 0}} \left(\overline{U}_t(n, V) - X(n, V)\right)^2 \right)$$

$$\geq \sum_{n=1}^\infty \frac{1}{2^n|\mathcal{V}_n|} \left( \sum_{\substack{V \in V_n \\ \overline{U}_t(n,V) > 0}} \left(\overline{U}_t(n, V) - 0\right)^2 + 0 \right)$$

$$= D(\overline{U}_t - Y_t, \overline{U}_t - Y_t), \qquad\qquad \text{(Only the } \overline{U}(n, V) > 0 \text{ terms remain)}$$

and it proves the claim. $\qquad\qquad\qquad\qquad\qquad\qquad\qquad\qquad\qquad\qquad\qquad\qquad\square$

**Claim 2.** *In the setting of Theorem 2, the following properties hold when $\overline{U}_t \notin \mathcal{S}$.*

1. $P(Y_t, W_t) = 0$

2. $P(U_{t+1}, W_t) = 0$

3. $P(\overline{U}_t, W_t) > 0$

4. $P(X, W_t) \leq 0$ *for all* $X \in \mathcal{S}$

*Proof of the claim.* All of these properties can be explained by expanding the dot product $P$ and referring to the strategy $S_A$. The first one is the most straightforward: $Y_t$ and $W_t$ are never nonzero in the same coordinate, so $P(Y_t, W_t) = 0$. For the second one, by definition, in round $t + 1$ Alice cuts at a point $x$ such that $P(G_x, W_t) = 0$. If Bob selects the left piece, $U_{t+1} = G_x$. If Bob instead selects the right piece:

$$U_{t+1}(n, V) = (1 - V(x)) - \frac{1}{2}$$
$$= \frac{1}{2} - V(x)$$
$$= -G_x(n, V).$$

So by property 4 of Lemma 8, we have $P(U_{t+1}, W_t) = -P(G_x, W_t) = 0$. Because $P$ is linear as per property 5 in Lemma 8, any mixed strategy over these outcomes must also satisfy $P(U_{t+1}, W_t) = 0$.

For part (3) of the lemma, expanding the dot product gives:

$$P(X, W_t) = \sum_{n=1}^{\infty} \frac{1}{2^n |\mathcal{V}_n|} \sum_{V \in \mathcal{V}_n} X(n, V) \cdot \max\{0, \overline{U}_t(n, V)\}$$
$$= \sum_{n=1}^{\infty} \frac{1}{2^n |\mathcal{V}_n|} \sum_{V \in \mathcal{V}_n, \overline{U}_t(n,V) > 0} X(n, V) \overline{U}_t(n, V).$$

As there exists $(n, V)$ such that $\overline{U}_t(n, V) > 0$, this sum is strictly positive when $X = \overline{U}_t(n, V)$, and so $P(\overline{U}_t, W_t) > 0$. Similarly, for $X \in \mathcal{S}$, we have $X(n, V) \leq 0$ for all $(n, V)$, so $P(X, W_t) \leq 0$. $\qquad\square$

**Claim 3.** *In the setting of Theorem 2, the sequence $\{\delta_t\}_{t=1}^{\infty}$ defined in (66) satisfies the inequality $\delta_t \leq 1/t$ for all $t \geq 2$.*

*Proof.* We will show that our construction of $\delta_t$ satisfies the recursion formula defined in Lemma 10. We first focus on the case such that $\delta_t > 0$, which is equivalent to $\overline{U}_t \notin \mathcal{S}$, and then prove that it also holds for $\delta_t = 0$. Given that $\delta_t > 0$, we observe that

$$\delta_{t+1} = \inf_{X \in \mathcal{S}} P(\overline{U}_{t+1} - X, \overline{U}_{t+1} - X) \leq P(\overline{U}_{t+1} - Y_t, \overline{U}_{t+1} - Y_t). \tag{79}$$

By rewriting the right hand side of (79), we obtain

$$\delta_{t+1} \leq P((\overline{U}_{t+1} - \overline{U}_t) + (\overline{U}_t - Y_t), (\overline{U}_{t+1} - \overline{U}_t) + (\overline{U}_t - Y_t)) \tag{80}$$

Distributing over $P$ in the expression on the right hand side of (80), we get that (80) is equivalent to

$$\delta_{t+1} \leq P(\overline{U}_{t+1} - \overline{U}_t, \overline{U}_{t+1} - \overline{U}_t) + 2P(\overline{U}_{t+1} - \overline{U}_t, \overline{U}_t - Y_t) + P(\overline{U}_t - Y_t, \overline{U}_t - Y_t)$$
$$= P(\overline{U}_{t+1} - \overline{U}_t, \overline{U}_{t+1} - \overline{U}_t) + 2P(\overline{U}_{t+1} - \overline{U}_t, \overline{U}_t - Y_t) + \delta_t, \tag{81}$$

where the inequality follows from Claim 1.

Noting that $\overline{U}_{t+1} - \overline{U}_t = (U_{t+1} - \overline{U}_t)/(t + 1)$, we have

$$P(\overline{U}_{t+1} - \overline{U}_t, \overline{U}_t - Y_t) = \frac{1}{t+1} P(U_{t+1} - \overline{U}_t, \overline{U}_t - Y_t)$$
$$= \frac{1}{t+1} P((U_{t+1} - Y_t) + (Y_t - \overline{U}_t), \overline{U}_t - Y_t)$$
$$= \frac{1}{t+1} \left( P(U_{t+1} - Y_t, \overline{U}_t - Y_t) + P(Y_t - \overline{U}_t, \overline{U}_t - Y_t) \right).$$

By using $\overline{U}_t = W_t + Y_t$, we can expand it by

$$P(\overline{U}_{t+1} - \overline{U}_t, \overline{U}_t - Y_t) = \frac{1}{t+1}\left(P(U_{t+1}, W_t) - P(Y_t, W_t) - P(Y_t - \overline{U}_t, \overline{U}_t - Y_t)\right).$$

$$= \frac{1}{t+1}\left(P(U_{t+1}, W_t) - P(Y_t, W_t) - \delta_t\right) \qquad \text{(By Claim 1)}$$

$$= \frac{1}{t+1}\left(0 - 0 - \delta_t\right) \qquad \text{(By Claim 2)}$$

$$= -\frac{1}{t+1} \cdot \delta_t. \tag{82}$$

Further, observe that

$$P(\overline{U}_{t+1} - \overline{U}_t, \overline{U}_{t+1} - \overline{U}_t) = \frac{1}{(t+1)^2}P(U_{t+1} - \overline{U}_t, U_{t+1} - \overline{U}_t)$$

$$\leq \frac{1}{(t+1)^2}, \tag{83}$$

where the inequality follows from Property (3) in Lemma 8.

Putting (81), (82) and (83) together, we obtain the following inequality for $\delta_t > 0$:

$$\delta_{t+1} \leq P(\overline{U}_{t+1} - \overline{U}_t, \overline{U}_{t+1} - \overline{U}_t) + 2P(\overline{U}_{t+1} - \overline{U}_t, \overline{U}_t - Y_t) + \delta_t$$

$$\leq \frac{1}{(t+1)^2} - \frac{2}{t+1}\delta_t + \delta_t$$

$$= \frac{1}{(t+1)^2} + \left(1 - \frac{2}{t+1}\right)\delta_t.$$

Now, we show that the same inequality holds for the case such that $\delta_t = 0$ too. Since $\overline{U}_t \in \mathcal{S}$, we obtain

$$\delta_{t+1} = \inf_{X \in \mathcal{S}} P(\overline{U}_{t+1} - X, \overline{U}_{t+1} - X)$$

$$\leq P(\overline{U}_{t+1} - \overline{U}_t, \overline{U}_{t+1} - \overline{U}_t)$$

$$\leq \frac{1}{(t+1)^2} \qquad \text{(By (83))}$$

$$= \frac{1}{(t+1)^2} + \left(1 - \frac{2}{t+1}\right)\delta_t.$$

By Lemma 10, we therefore have that $\delta_t \leq 1/t$ for $t \geq 2$. $\qquad \square$

## B.2 Appendix: Bob enforcing equitable payoffs

In this section, we prove Theorem 3, which shows how Bob can enforce equitable payoffs.

**Restatement of Theorem 3** (Bob enforcing equitable payoffs; formal)**.**

- In the sequential setting: *Bob has a pure strategy $S_B$, such that for every Alice strategy $S_A$, on every trajectory of play, Bob's average payoff is at least $1/2 - o(1)$, while Alice's average payoff is at most $1/2 + o(1)$. More precisely,*

$$\frac{u_B}{T} \geq \frac{1}{2} - \frac{1}{\sqrt{T}} \qquad \text{and} \qquad \frac{u_B}{T} \leq \frac{1}{2} + \left(\frac{\Delta}{2\delta} + 2\right)\frac{1}{\sqrt{T}},$$

*recalling that $\delta$ and $\Delta$ are, respectively, the lower and upper bounds on the players' value densities.*

- In the simultaneous setting: *Bob has a mixed strategy $S_B$, such that for every Alice strategy $S_A$, both players have average payoff $1/2$ in expectation.*

*Proof of Theorem 3.* Bob's strategy for the simultaneous setting follows from Proposition 6. His strategy for the sequential setting follows from Proposition 7. □

**Proposition 6.** *In the simultaneous setting, Bob has a mixed strategy $S_B$ such that, for every Alice strategy $S_A$:*

$$\frac{\mathbb{E}[u_A]}{T} = \frac{\mathbb{E}[u_B]}{T} = \frac{1}{2}.$$

*Proof.* Bob's strategy is very simple: in each round, randomly pick $L$ or $R$ with equal probability.

To analyze the expected payoffs, consider an arbitrary player $i \in \{A, B\}$ and arbitrary round $t$. Bob is equally likely to pick $L$ or $R$ in round $t$, so each player is equally likely to receive $[0, a_t]$ or $[a_t, 1]$ in round $t$. Therefore, their expected payoff is:

$$
\begin{aligned}
\mathbb{E}[u_i^t] &= \frac{1}{2} V_i([0, a_t]) + \frac{1}{2} V_i([a_t, 1]) \\
&= \frac{1}{2} V_i([0, 1]) &&\text{(Since valuations are additive)} \\
&= \frac{1}{2}. &&(84)
\end{aligned}
$$

Summing (84) over the $T$ rounds gives the desired expected payoffs for each player. □

**Proposition 7.** *Bob has a pure strategy $S_B$, such that for every Alice strategy $S_A$, the cumulative utilities in the sequential game are bounded by:*

$$u_A(S_A, S_B) \le T/2 + \left(\frac{\Delta}{2\delta} + 2\right) \cdot \sqrt{T} \qquad and \qquad u_B(S_A, S_B) \ge T/2 - \sqrt{T}.$$

*Proof.* Bob devises his strategy $S_B$ by considering a division of the cake into $P = \lceil \sqrt{T} \rceil$ consecutive intervals $I_1, \ldots, I_P$ of equal value to him. That is, Bob chooses points $0 = z_0 \le z_1 \ldots \le z_P = 1$ such that

$$
I_j = \begin{cases} [z_{j-1}, z_j) & \text{if } 1 \le j \le P - 1; \\ [z_{P-1}, z_P] & \text{if } j = P; \end{cases} \qquad \text{and} \qquad V_B(I_j) = 1/P \ \ \forall j \in [P]. \quad (85)
$$

An illustration of the division into intervals used by Bob can be seen in Figure 17.

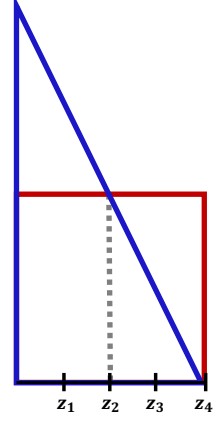

Figure 17: Example of a Bob density and the discretization used by Bob when $T = 10$. The number of intervals is $P = \lceil \sqrt{T} \rceil = 4$. The intervals are $I_j = [z_{j-1}, z_j)$ for $j \in [3]$ and $I_4 = [z_3, z_4]$, with $V_B(I_j) = 1/4 \ \forall j \in [4]$.

Bob's strategy $S_B$ is defined as follows. Bob keeps a counter $j$ associated with each interval $I_j$, such that the value of the counter at time $t$, denoted $c_{j,t}$, represents how many times Alice has cut inside the interval $I_j$ in the first $t$ rounds (including round $t$). For each time $t \in [T]$:

- Let $I_j$ be the interval that contains Alice's cut at time $t$ (that is, $a_t \in I_j$), for $j \in [P]$.

- If $c_{j,t}$ is even then Bob plays L; if $c_{j,t}$ is odd, then Bob plays R.

Informally, Bob alternates between L and R inside each interval $I_j$. We argue that this Bob strategy ensures his payoff is at least $1/2 - o(1)$ per round, while Alice cannot get more than $1/2 + o(1)$ per round.

For each $i \in [P]$ and $j \in [c_{i,T}]$, let $r_{i,j}$ be the first time when the number of cuts in $I_i$ reached $j$. That is,

$$r_{i,j} = \min\{t \in \mathbb{N} \mid c_{i,t} = j \text{ and } a_t \in I_i\}.$$

By definition of the sequence $\{r_{i,\ell}\}$, for each $i, j \in \mathbb{N}$ with $2j \le c_{i,T}$:

- Alice cut in the interval $I_i$ in both rounds $r_{i,2j-1}$ and $r_{i,2j}$ (meaning $a_{r_{i,2j-1}}, a_{r_{i,2j}} \in I_i$);

- Bob played different actions in the rounds $r_{i,2j-1}$ and $r_{i,2j}$.

We view rounds $r_{i,2j-1}$ and $r_{i,2j}$ as a pair. For each $i \in [P]$, there is at most one round $r_{i,j}$ that does not have a pair, namely round $r_{i,c_{i,T}}$: the last round Alice cut in $I_i$. However, this loss only represents at most $P$ rounds in total, which will translate to a sub-linear loss for either Alice's or Bob's utility estimates.

Now we can bound the cumulative utility of each player.

**Bob's payoff.** For each $i, j \in \mathbb{N}$ with $2j \le c_{i,T}$, Bob's payoff across the two rounds $r_{i,2j-1}$ and $r_{i,2j}$ is bounded by

$$u_B^{r_{i,2j-1}} + u_B^{r_{i,2j}} \ge 1 - V_B(I_i) = 1 - \frac{1}{P}. \tag{86}$$

Since the rounds $r_{i,j}$ with $i \in [P]$ and $2j \le c_{i,T}$ represent a subset of the total set of rounds $[T]$, Bob's cumulative payoff is at least

$$
\begin{aligned}
\sum_{t=1}^{T} u_B^t &\ge \sum_{i=1}^{P} \left( \sum_{j \in \mathbb{N}: 2j \le c_{i,T}} u_B^{r_{i,2j-1}} + u_B^{r_{i,2j}} \right) \\
&\ge \sum_{i=1}^{P} \left( \sum_{j \in \mathbb{N}: 2j \le c_{i,T}} \left( 1 - \frac{1}{P} \right) \right) &&\text{(By (86))} \\
&\ge \left( 1 - \frac{1}{P} \right) \frac{T-P}{2} &&\text{(Since the sum is over at least } \tfrac{T-P}{2} \text{ pairs of rounds)}
\end{aligned}
$$

Since $P = \lceil \sqrt{T} \rceil \ge \sqrt{T}$, we get

$$
\begin{aligned}
u_B = \sum_{t=1}^{T} u_B^t &\ge \left( 1 - \frac{1}{P} \right) \frac{T-P}{2} = \frac{T}{2} - \frac{\lceil \sqrt{T} \rceil}{2} - \frac{T}{2\lceil \sqrt{T} \rceil} + \frac{1}{2} \\
&\ge \frac{T}{2} - \frac{\sqrt{T}+1}{2} - \frac{\sqrt{T}}{2} + \frac{1}{2} \\
&= \frac{T}{2} - \sqrt{T}. \tag{87}
\end{aligned}
$$

**Alice's payoff.** For each $i, j \in \mathbb{N}$ with $2j \le c_{i,T}$, Alice's payoff across rounds $r_{i,2j-1}$ and $r_{i,2j}$ is

$$u_A^{r_{i,2j-1}} + u_A^{r_{i,2j}} \le 1 + V_A(I_i) \le 1 + \frac{\Delta}{\delta} V_B(I_i) = 1 + \frac{\Delta}{\delta} \cdot \frac{1}{P}. \tag{88}$$

There is at most one round without a pair for each interval $I_i$, namely round $r_{i,c_{i,T}}$: the last time Alice cut in $I_i$. For each such round without a pair, we upper bound Alice's payoff by 1. Then we can upper bound Alice's cumulative utility by

$$\sum_{t=1}^{T} u_A^t \leq P + \sum_{i=1}^{P} \left( \sum_{j\in\mathbb{N}:2j\leq c_{i,T}} u_A^{r_{i,2j-1}} + u_A^{r_{i,2j}} \right)$$

$$\leq P + \sum_{i=1}^{P} \left( \sum_{j\in\mathbb{N}:2j\leq c_{i,T}} \left( 1 + \frac{\Delta}{\delta} \cdot \frac{1}{P} \right) \right) \qquad \text{(By (88))}$$

$$\leq P + \frac{T}{2} \left( 1 + \frac{\Delta}{P\delta} \right) \qquad \text{(Since the sum is over at most } T/2 \text{ pairs)}$$

$$= \frac{T}{2} + \frac{T}{P} \cdot \frac{\Delta}{2\delta} + P$$

$$\leq \frac{T}{2} + \sqrt{T} \left( \frac{\Delta}{2\delta} + 2 \right) \qquad (P = \lceil \sqrt{T} \rceil \leq 2\sqrt{T})$$

This completes the proof. $\qquad\qquad\qquad\qquad\qquad\qquad\qquad\qquad\qquad\qquad\qquad\qquad$ $\square$

# C  Appendix: Fictitious play

In this section, we analyze the fictitious play dynamics. We first formally define fictitious play in terms of the *empirical frequency* and *empirical distribution*:

- The empirical frequency of Alice's play up to (but not including) time $t$ is:

$$\phi_A^t(x) = \sum_{\tau=1}^{t-1} \mathbb{1}_{\{a_\tau = x\}} \quad \forall x \in [0,1].$$

- The empirical frequency of Bob's play up to (but not including) time $t$ is:

$$\phi_B^t(x) = \sum_{\tau=1}^{t-1} \mathbb{1}_{\{b_\tau = x\}} \quad \forall x \in \{L, R\}.$$

- The *empirical distribution* of player $i$'s play up to (but not including) time $t$ is:

$$\mathfrak{p}_i^t(x) = \frac{\phi_i^t(x)}{(t-1)},$$

Where $x \in [0,1]$ for Alice and $x \in \{L, R\}$ for Bob.

**Definition 9.** *(Fictitious play) In round $t = 1$, each player simultaneously selects an arbitrary action. In every round $t = 2, \ldots, T$, each player simultaneously best responds to the empirical distribution of the other player up to time $t$. If there are multiple best responses, the player chooses one arbitrarily.*

We now give the proof of Theorem 4 using lemmas that will be presented later, and then prove the required lemmas.

**Restatement of Theorem 4.** *When both Alice and Bob run fictitious play, regardless of tie-breaking rules, their average payoff will converge to $1/2$ at a rate of $O(1/\sqrt{T})$. Formally:*

$$\left| \frac{u_A}{T} - \frac{1}{2} \right| \leq \frac{2\sqrt{10}}{\sqrt{T}} \quad and \quad \left| \frac{u_B}{T} - \frac{1}{2} \right| \leq \frac{\sqrt{10}}{\sqrt{T}} \quad \forall T \geq 5. \tag{89}$$

*Proof.* The bounds on Bob's payoff follow immediately from Lemma 21, which states that

$$\frac{T}{2} - \sqrt{10T} \leq \sum_{t=1}^{T} u_B^t \leq \frac{T}{2} + \sqrt{10T}. \tag{90}$$

By Lemma 22,

$$T - \sqrt{10T} \leq \sum_{t=1}^{T} \left( u_A^t + u_B^t \right) \leq T + \sqrt{10T}. \tag{91}$$

Subtracting equation (90) from (91) gives

$$\frac{T}{2} - 2\sqrt{10T} \leq \sum_{t=1}^{T} u_A^t \leq \frac{T}{2} + 2\sqrt{10T}. \tag{92}$$

Since $u_A = \sum_{t=1}^{T} u_A^t$ and $u_B = \sum_{t=1}^{T} u_B^t$, both players' utilities are bounded as required, which completes the proof. $\square$

To prove the required lemmas, we first introduce several notations. For ease of exposition, we often use round $t = 0$ as a fake round in which nothing actually happens and all the defined quantities are zero.

**Definition 10.** *For $t \in \{0, 1, \ldots, T\}$:*

- *Let $r_t$ be the number of rounds in which Bob has picked $R$, up to and including round $t$.*

- *Let $\ell_t$ be the number of rounds in which Bob has picked $L$ up to and including round $t$.*

- *Let $\alpha_t = r_t - \ell_t$.*

- *Let $\beta_t = \sum_{i=1}^{t}(2V_B([0, a_i]) - 1)$.*

- *Let $\rho_t = |\alpha_t| + |\beta_t|$, which we call the **radius**.*

We remark that in the fake round of $t = 0$, all these variables have the value zero. Alice's action under fictitious play entirely depends on the variable $\alpha_t$ while Bob's action entirely depends on $\beta_t$. Overall, we will bound the payoffs using the growth of the radius $\rho_t$ over $t = 0, 1, \ldots, T$. To this end, we introduce two lemmas that will play an essential role throughout the proof.

First, the following lemma formally argues that the action taken by each player is guided by their corresponding variable $\alpha$ and $\beta_t$, respectively.

**Lemma 11.** *Let $t \in \{0, 1, \ldots, T-1\}$. Then the following hold for round $t + 1$:*

- *Alice's action with respect to $\alpha_t$:*

  - *If $\alpha_t > 0$, Alice will cut at 1.*
  - *If $\alpha_t < 0$, Alice will cut at 0.*
  - *If $\alpha_t = 0$, any action would incur the same payoff, so she could cut anywhere.*

- *Bob's action with respect to $\beta_t$:*

  - *If $\beta_t > 0$, Bob will pick L.*
  - *If $\beta_t < 0$, Bob will pick R.*
  - *If $\beta_t = 0$, any action would incur the same payoff, so he could pick either L or R.*

*Proof.* First, consider Alice's choice. For $t = 0$, there is no history, so any choice has the same value to her; accordingly, $\alpha_0 = 0$. For $t \geq 1$, the expected value she assigns to any particular cut location $x$ is:

$$
\begin{aligned}
\frac{1}{t}\left(r_t V_A([0, x]) + \ell_t V_A([x, 1])\right) &= \frac{1}{t}\left(r_t V_A([0, x]) + \ell_t(1 - V_A([0, x]))\right) \\
&= \frac{r_t - \ell_t}{t} V_A([0, x]) + \frac{\ell_t}{t} \\
&= \frac{\alpha_t}{t} V_A([0, x]) + \frac{\ell_t}{t}.
\end{aligned} \tag{93}
$$

From (93), we get that Alice's cut decision is entirely based on $\alpha_t$. If $\alpha_t < 0$, she will minimize $V_A([0, x])$, which means she cuts at 0. If $\alpha_t > 0$, she will maximize $V_A([0, x])$, which means she cuts at 1. If $\alpha_t = 0$, then her choice of $x$ doesn't affect her expected value.

Now consider Bob's choice. For $t = 0$, there is no history, so Bob is indifferent between $L$ and $R$; accordingly, $\beta_0 = 0$. For $t \geq 1$, the expected value he assigns to choosing $L$ is:

$$
E_L = \frac{1}{t}\sum_{i=1}^{t} V_B([0, a_i]). \tag{94}
$$

The expected value he assigns to choosing $R$ is:

$$
E_R = \frac{1}{t}\sum_{i=1}^{t} V_B([a_i, 1]). \tag{95}
$$

Combining (94) and (95), the difference between his expected value for choosing $L$ and $R$ is:

$$
E_L - E_R = \left(\frac{1}{t}\sum_{i=1}^{t} V_B([0, a_i])\right) - \left(\frac{1}{t}\sum_{i=1}^{t} V_B([a_i, 1])\right) = \frac{1}{t}\sum_{i=1}^{t}(2V_B([0, a_i]) - 1) = \frac{\beta_t}{t}. \tag{96}
$$

From (96), we get that Bob's decision is entirely based on $\beta_t$. If $\beta_t < 0$, he values $R$ more than $L$, so he picks $R$. If $\beta_t > 0$, he values $L$ more than $R$, so he picks $L$. If $\beta_t = 0$, he values each equally. □

The following lemma further describes the evolution of $\alpha_t$ and $\beta_t$ given the actions taken by the players.

**Lemma 12.** *For each round $t \in [T]$:*

- *Alice's action affects $\beta_t$ as follows:*

    - *If Alice cuts at $0$ in round $t$, then $\beta_t = \beta_{t-1} - 1$.*
    - *If Alice cuts at $1$ in round $t$, then $\beta_t = \beta_{t-1} + 1$.*
    - *If she cuts at $x \in (0, 1)$, then $|\beta_t - \beta_{t-1}| < 1$.*

- *Bob's action affects $\alpha_t$ as follows:*

    - *If Bob picks $L$ in round $t$, then $\alpha_t = \alpha_{t-1} - 1$.*
    - *If Bob picks $R$ in round $t$, then $\alpha_t = \alpha_{t-1} + 1$.*

*Proof.* First, we consider the impact of Alice's cut point $a_t$ on $\beta_t$. Explicitly writing out the difference $\beta_t - \beta_{t-1}$ gives:

$$\beta_t - \beta_{t-1} = \sum_{i=1}^{t}(2V_B([0, a_i]) - 1) - \sum_{i=1}^{t-1}(2V_B([0, a_i]) - 1) = 2V_B([0, a_t]) - 1 . \quad (97)$$

If $a_t = 0$, then $V_B([0, a_t]) = V_B([0, 0]) = 0$, so by (97) we have $\beta_t - \beta_{t-1} = -1$ as desired.

If $a_t = 1$, then $V_B([0, a_t]) = V_B([0, 1]) = 1$, so by (97) we have $\beta_t - \beta_{t-1} = 1$.

If $a_t \in (0, 1)$, then there is at least some cake on each side of $a_t$, so we have $V_B([0, a_t]) \in (0, 1)$. By (97), we have $\beta_t - \beta_{t-1} \in (-1, 1)$, so $|\beta_t - \beta_{t-1}| < 1$ as stated by the lemma.

Second, we consider the impact of Bob's choice on $\alpha_t$. Explicitly writing out the difference $\alpha_t - \alpha_{t-1}$, we obtain the following:

$$\alpha_t - \alpha_{t-1} = (r_t - \ell_t) - (r_{t-1} - \ell_{t-1}) = (r_t - r_{t-1}) - (\ell_t - \ell_{t-1}) = \begin{cases} 1 & b_t = R \\ -1 & b_t = L \end{cases} \quad (98)$$

This completes the proof. $\qquad \square$

Let us elaborate more the dynamics of each player based on Lemma 11 and 12. If either of $\alpha_t$ or $\beta_t$ is exactly 0, the corresponding player will use their tie-breaking rules. Ignoring these cases, these choices lead to movement through $(\alpha_t, \beta_t)$ space that spirals counter-clockwise around the origin at exactly 45 degree angles. Figure 18-(a) describes the overall dynamics of the variables $(\alpha_t, \beta_t)$.

The following lemma shows a symmetry that will help reduce the number of cases in the subsequent analysis. Figure 18-(b) depicts the rotational symmetry of fictitious play dynamics in the $\alpha$-$\beta$-plane shown by the lemma. Specifically, the symmetry will allow us to assume $\alpha_t \geq 0$ without loss of generality when analyzing $\rho_t$ and $u_B^t$.

**Lemma 13.** *Consider an arbitrary pair of tie-breaking rules for both players. Consider the resulting sequence for the variables $\alpha_t, \beta_t$ and $u_B^t$ for $t = 0, \ldots, T$. Then, there exists another choice of tie-breaking rules that would result in the sequence of variables $\tilde{\alpha}_t, \tilde{\beta}_t,$ and $\tilde{u}_B^t$ for $t = 0, \ldots, T$ such that*

$$\tilde{\alpha}_t = -\alpha_t; \quad \tilde{\beta}_t = -\beta_t; \quad \tilde{u}_B^t = u_B^t . \quad (99)$$

*Proof.* The proof will proceed by induction.

**Base case** All of $\alpha_0, \tilde{\alpha}_0, \beta_0, \tilde{\beta}_0, u_B^0,$ and $\tilde{u}_B^0$ are zero, so (99) trivially holds.

**Inductive hypothesis** Assume that (99) holds for some $t \geq 0$.

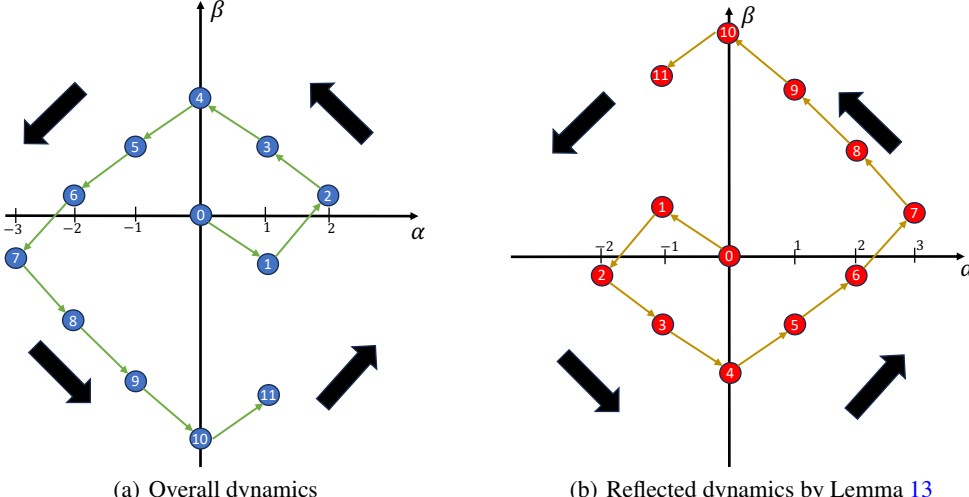

(a) Overall dynamics  (b) Reflected dynamics by Lemma 13

Figure 18: Figure (a) represents the overall illustration of the dynamics of the action quantities $\alpha_t$ and $\beta_t$ for $t = 0, 1, \ldots, T$. The $x$-axis denotes the quantity of $\alpha_t$ and the $y$-axis denotes that of $\beta_t$. Blue circles represent the sequence of points in the plane, where the number inside the circle denotes the index $t$. Note that $\alpha_t$ only takes integer values, while $\beta_t$ possibly takes noninteger values at some rounds. Figure (b) also depicts an overall dynamics implemented by another pair of tiebreaking rules guaranteed by Lemma 13. Note that each point is reflected with respect to the origin point. Importantly, Lemma 13 guarantees that $\rho_t$ and $u_B^t$ for $t = 0, 1, \ldots, T$ remain exactly the same for both dynamics.

**Inductive step**   We show that (99) holds for $t + 1$. Suppose that, under the original tie-breaking rules, Alice cut at $a_{t+1}$ and Bob picked $b_{t+1} \in \{L, R\}$ in round $t + 1$. First, we analyze for Alice. Let $a'_{t+1} \in [0, 1]$ be the unique point for which

$$V_B([0, a_{t+1}]) = V_B([a'_{t+1}, 1]). \tag{100}$$

The point $a'_{t+1}$ is uniquely defined since Bob's density is strictly positive.

We show that there exists another choice of tie-breaking rules under which, starting from $\tilde{\alpha}_t$ and $\tilde{\beta}_t$, Alice cuts at $a'_{t+1}$ in round $t + 1$. We split into cases based on $\alpha_t$:

($\alpha_t > 0$): Then $a_{t+1} = 1$ by Lemma 11. Since $V_B([0, a_{t+1}]) = V_B([0, 1])$, we have $a'_{t+1} = 0$ by definition. By the inductive hypothesis, we have $\tilde{\alpha}_t = -\alpha_t$, and so $\tilde{\alpha}_t < 0$. Then, regardless of tie-breaking rules, Alice cuts at $0 = a'_{t+1}$.

($\alpha_t < 0$): Then $a_{t+1} = 0$ by Lemma 11. Since $V_B([0, a_{t+1}] = 0 = V_B([1, 1])$, we have $a'_{t+1} = 1$ by definition. By the inductive hypothesis, $\tilde{\alpha}_t = -\alpha_t$, and so $\tilde{\alpha}_t > 0$. Then, regardless of tie-breaking rules, Alice cuts at $1 = a'_{t+1}$.

($\alpha_t = 0$): Then Alice can break ties any way she likes by Lemma 11. However, $\tilde{\alpha}_t = -\alpha_t = 0$, so any cut point is a valid choice for her new tie-breaking rule. In particular, she can cut at $a'_{t+1}$.

Second, we analyze for Bob. Let $b'_{t+1} \in \{L, R\}$ be such that $b'_{t+1} \neq b_{t+1}$. We show that there exist tie-breaking rules under which, starting from $\tilde{\alpha}_t$ and $\tilde{\beta}_t$, Bob chooses $b'_{t+1}$. We split into cases based on $\beta_t$:

($\beta_t > 0$): Then $b_{t+1} = L$ by Lemma 11; therefore, $b'_{t+1} = R$. By the inductive hypothesis, $\tilde{\beta}_t = -\beta_t$, so $\tilde{\beta}_t < 0$. Then, regardless of tie-breaking rules, Bob picks $R = b'_{t+1}$.

($\beta_t < 0$): Then $b_{t+1} = R$ by Lemma 11; therefore, $b'_{t+1} = L$. By the inductive hypothesis, $\tilde{\beta}_t = -\beta_t$, so $\tilde{\beta}_t > 0$. Then, regardless of tie-breaking rules, Bob picks $L = b'_{t+1}$.

($\beta_t = 0$): Then Bob can break ties any way he likes by Lemma 11. However, $\tilde{\beta}_t = -\beta_t = 0$, so either $L$ or $R$ is a valid choice for his new tie-breaking rule. In particular, he can choose $b'_{t+1}$.

Finally, we show that these opposite choices have exactly the desired effect on $\tilde{\alpha}_{t+1}$, $\tilde{\beta}_{t+1}$, and $\tilde{u}_B^{t+1}$. Covering each in turn:

- Since Bob picks the opposite side under the trajectory associated with $\tilde{\alpha}$ and $\tilde{\beta}$, the change from $\alpha_t$ to $\alpha_{t+1}$ is in the opposite direction as the change from $\tilde{\alpha}_t$ to $\tilde{\alpha}_{t+1}$ by Lemma 12. By the inductive hypothesis, we have $\tilde{\alpha}_t = -\alpha_t$, and so $\tilde{\alpha}_{t+1} = -\alpha_{t+1}$.

- Since Alice picks the mirror image of her cut point under the trajectory associated with $\tilde{\alpha}$ and $\tilde{\beta}$, the change from $\beta_t$ to $\beta_{t+1}$ is exactly opposite to the change from $\tilde{\beta}_t$ to $\tilde{\beta}_{t+1}$. Specifically,

$$\beta_{t+1} = \beta_t + \Big(2V_B([0, a_{t+1}]) - 1\Big), \tag{101}$$

while

$$\begin{aligned}
\tilde{\beta}_{t+1} &= \tilde{\beta}_t + \Big(2V_B([0, a'_{t+1}]) - 1\Big) \\
&= \tilde{\beta}_t + 2(1 - V_B([a'_{t+1}, 1])) - 1 \\
&= \tilde{\beta}_t + 2(1 - V_B([0, a_{t+1}])) - 1 \\
&= \tilde{\beta}_t - \Big(2V_B([0, a_{t+1}]) - 1\Big).
\end{aligned} \tag{102}$$

By the inductive hypothesis, we have $\tilde{\beta}_t = -\beta_t$. Using equations (101) and (102), we obtain $\tilde{\beta}_{t+1} = -\beta_{t+1}$.

- Under Alice's new cut point $a'_{t+1}$, Bob's valuation of the left and right sides of the cake swap, i.e., $V_B([0, a'_{t+1}]) = V_B([a_{t+1}, 1])$. But he also chooses the opposite side, so he gets exactly the same payoff as under the original tie-breaking rules in round $t + 1$. Therefore, $\tilde{u}_B^{t+1} = u_B^{t+1}$.

By induction, the claim holds for all $t$, which completes the proof. $\qquad\square$

In addition, it is helpful to distinguish between the rounds that cross an axis in the $\alpha$-$\beta$ plane and those that do not. The following formalizes the definition of such rounds, which are depicted in Figure 19-(a).

**Definition 11.** *An* axis-crossing round *is a round $t$ where at least one of the following occurs:*

- $\alpha_t = 0$

- $\beta_{t+1} > 0$*, but $\beta_t \leq 0$*

- $\beta_{t+1} < 0$*, but $\beta_t \geq 0$.*

Importantly, we will show that $\rho_t$ can strictly increase only if the current round is axis-crossing, while it is non-decreasing over the entire game. The following lemma formalizes this observation. We provide an example in Figure 19-(b).

**Lemma 14.** *Let $t \in \{0, 1, \ldots, T\}$. The radius $\rho_t$ satisfies the following properties:*

- *(a) $\rho_t = \rho_{t+1}$ if $t$ is not axis-crossing*

- *(b) $\rho_t \leq \rho_{t+1} \leq 2 + \rho_t$ if $t$ is axis-crossing*

- *(c) $\rho_0 = 0$*

- *(d) $\rho_t \geq 1$ for $t \geq 1$.*

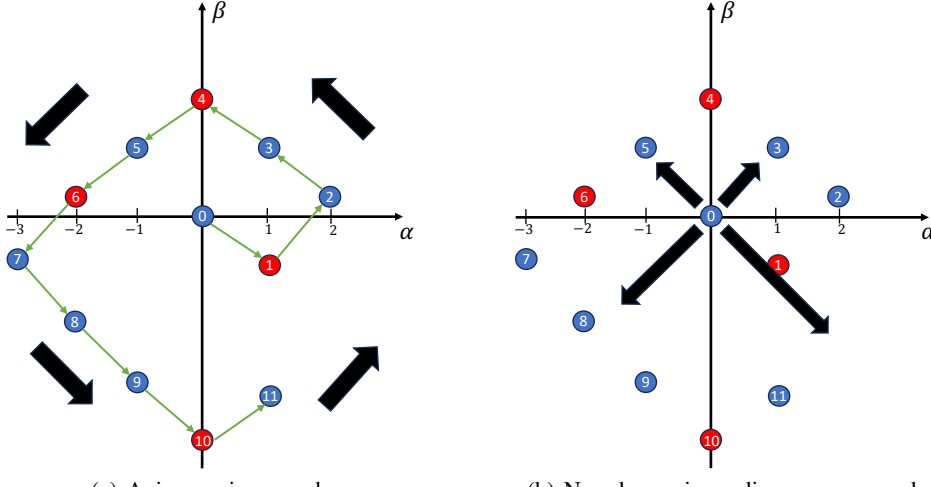

(a) Axis-crossing rounds
(b) Non-decreasing radius $\rho_t$ over rounds

Figure 19: Overall dynamics of $\alpha_t$ and $\beta_t$ with non axis-crossing rounds (blue circles) and axis-crossing rounds (red circles). Figure (b) shows that the radius $\rho_t = |\alpha_t| + |\beta_t|$ is nondecreasing in $t$ as shown by Lemma 14. In particular, $\rho_t$ remains the same for non axis-crossing rounds but possibly increases for axis-crossing rounds.

*In particular, the radius $\rho_t$ is non-decreasing in $t$.*

*Proof.* First, we will show (a) and (b). Consider an arbitrary round $t \in \{0, \ldots, T\}$. By Lemma 12, we have $|\alpha_{t+1} - \alpha_t| \leq 1$ and $|\beta_{t+1} - \beta_t| \leq 1$, so by the triangle inequality

$$\left| \rho_{t+1} - \rho_t \right| = \left| |\alpha_{t+1}| + |\beta_{t+1}| - |\alpha_t| - |\beta_t| \right| \leq 2 \,. \tag{103}$$

Therefore, $\rho_{t+1} \leq 2 + \rho_t$, as required by (b). Thus for (a) and (b) it remains to show that $\rho_{t+1} \geq \rho_t$ $\forall t \in \{0, \ldots, T\}$ and that $\rho_{t+1} = \rho_t$ if $t$ is not axis-crossing.

By Lemma 13, it suffices to consider $\alpha_t \geq 0$. More precisely, this is because if $\alpha_t \leq 0$, then there exists a tie-breaking rule with associated $\tilde{\alpha}_t$ satisfying $\tilde{\alpha}_t = -\alpha_t$ for every $t = 0, 1, \ldots, T$, so $\tilde{\alpha}_t \geq 0$. Crucially, the lemma ensures the sequences $\tilde{\alpha}_t$ and $\alpha_t$ have the same radius $\rho_t$.

We consider a few cases based on $\alpha_t$ and $\beta_t$, considering only $\alpha_t \geq 0$. Since $\alpha_t$ is an integer, also divide into $\alpha_t \geq 1$ and $\alpha_t = 0$:

$(\alpha_t \geq 1 \text{ and } \beta_t > 0)$**:** By Lemma 11 Alice will cut at 1 and Bob will pick L. Then by Lemma 12, we have $\alpha_{t+1} = \alpha_t - 1$ and $\beta_{t+1} = \beta_t + 1$. Therefore:

$$\begin{aligned} \rho_{t+1} = |\alpha_{t+1}| + |\beta_{t+1}| = \alpha_t - 1 + \beta_t + 1 && \text{(Since } \alpha_{t+1} \geq 0) \\ = |\alpha_t| + |\beta_t| && \text{(Because } \alpha_t, \beta_t \geq 0) \\ = \rho_t \,. && \text{(104)} \end{aligned}$$

$(\alpha_t \geq 1 \text{ and } \beta_t = 0)$**:** Then Alice will cut at 1 and Bob will pick whichever piece he likes. Therefore $\alpha_{t+1} = \alpha_t \pm 1$ and $\beta_{t+1} = \beta_t + 1 = 1$. Since $\beta_t \leq 0$ but $\beta_{t+1} > 0$, round $t$ is axis-crossing. What remains is to show that $\rho_{t+1} \geq \rho_t$ in this case:

$$\begin{aligned} \rho_{t+1} = |\alpha_{t+1}| + |\beta_{t+1}| \geq |\alpha_t| - 1 + 1 && \text{(Because } |x \pm y| \geq |x| - |y| \text{ for all } x, y) \\ = |\alpha_t| + |\beta_t| && \text{(Since } \beta_t = 0) \\ = \rho_t \,. && \text{(105)} \end{aligned}$$

$(\alpha_t \geq 1 \text{ and } -1 < \beta_t < 0)$**:** Then Alice will cut at 1 and Bob will pick R. Therefore, $\alpha_{t+1} = \alpha_t + 1$ and $\beta_{t+1} = \beta_t + 1$. Since $\beta_t < 0$ but $\beta_{t+1} > 0$, round $t$ is axis-crossing. What

remains is to show that $\rho_{t+1} \geq \rho_t$:

$$\rho_{t+1} = |\alpha_{t+1}| + |\beta_{t+1}| = \alpha_t + 1 + |\beta_t + 1|$$
$$\geq \alpha_t + 1 + |\beta_t| - 1 \qquad \text{(Since } |x + y| \geq |x| - |y| \text{ for all } x, y)$$
$$= \rho_t . \tag{106}$$

($\alpha_t \geq 1$ **and** $\beta_t \leq -1$)**:** Then Alice will cut at 1 and Bob will pick R. Therefore, $\alpha_{t+1} = \alpha_t + 1$ and $\beta_{t+1} = \beta_t + 1$. In this case, $\rho_{t+1} = \rho_t$:

$$\rho_{t+1} = |\alpha_{t+1}| + |\beta_{t+1}| = \alpha_t + 1 - (\beta_t + 1) \qquad \text{(Because } \beta_t + 1 \leq 0)$$
$$= |\alpha_t| + |\beta_t| \qquad \text{(Since } \beta_t \leq 0)$$
$$= \rho_t . \tag{107}$$

($\alpha_t = 0$ **and** $\beta_t \geq 0$)**:** Then $t$ is an axis-crossing round. Bob could pick either L or R, but either way $|\alpha_{t+1}| = 1$. Alice could cut anywhere, but since $|\beta_{t+1} - \beta_t| \leq 1$ we can still conclude $|\beta_t| - |\beta_{t+1}| \leq 1$. Therefore:

$$\rho_{t+1} = |\alpha_{t+1}| + |\beta_{t+1}| \geq 1 + |\beta_t| - 1$$
$$= |\alpha_t| + |\beta_t| \qquad \text{(Since } \alpha_t = 0)$$
$$= \rho_t . \tag{108}$$

($\alpha_t = 0$ **and** $\beta_t < 0$)**:** Then we can use the symmetry of Lemma 13 to consider $\beta_t > 0$ instead, which has already been covered.

In all cases, properties (a) and (b) must hold.

Now we show that the properties (c) and (d) hold. Property (c) follows from $\alpha_0 = \beta_0 = 0$. Because $\rho_{t+1} \geq \rho_t$ for all $t$, property (d) would follow from showing $\rho_1 \geq 1$, which can be seen true from the following inequality:

$$\rho_1 = |\alpha_1| + |\beta_1| = 1 + |\beta_1| \qquad \text{(Since } \alpha_0 = 0, \text{ so } \alpha_1 = \pm 1)$$
$$\geq 1 . \tag{109}$$

This finishes the proof. $\qquad \square$

**Lemma 15.** *Suppose $t - 1$ is an axis-crossing round and $\tau > t - 1$ is the next axis-crossing round after $t - 1$. Then, there exists at least $\rho_t - 2$ and at most $\rho_t$ rounds between them,* i.e.,

$$\rho_t - 2 \leq \tau - t \leq \rho_t . \tag{110}$$

*Proof.* By Lemma 13, we can assume $\alpha_{t-1} \geq 0$. Then it suffices to consider only four types that the axis-crossing round $t - 1$ could have:

(i) $\alpha_{t-1} \geq 1$, $\beta_{t-1} \leq 0$, and $\beta_t > 0$;

(ii) $\alpha_{t-1} \geq 1$, $\beta_{t-1} \geq 0$, and $\beta_t < 0$;

(iii) $\alpha_{t-1} = 0$ and $\beta_{t-1} \geq 1$;

(iv) $\alpha_{t-1} = 0$ and $0 \leq \beta_{t-1} < 1$.

We show separately for each of the types **(i)**-**(iv)**.

**(i)** $\alpha_{t-1} \geq 1$**,** $\beta_{t-1} \leq 0$**, and** $\beta_t > 0$**.** Because $\alpha_{t-1} > 0$, Alice will cut at 1 in round $t$ by Lemma 11, so $\beta_t = \beta_{t-1} + 1$. Since $\beta_{t-1}$ and $\beta_t$ have different signs, it must be that $-1 < \beta_{t-1} \leq 0$ and $0 < \beta_t \leq 1$. As long as $\alpha$ remains positive, Alice will keep cutting at 1 and increasing $\beta$ by Lemma 11 and Lemma 12, so the next axis-crossing round $\tau$ cannot be one where $\beta$ changes sign. Thus, it must be the one satisfying $\alpha_\tau = 0$. Until then, Bob will keep picking L, so $\alpha$ will decrease by 1 every round. Therefore, this implies that $\tau = t + \alpha_t$. To show $\tau - t \leq \rho_t$ as required by (110), we have

$$\tau - t = \alpha_t \leq |\alpha_t| + |\beta_t| \leq \rho_t.$$

To prove $\rho_t - 2 \leq \tau - t$, observe that $\alpha_t \geq \alpha_{t-1} - 1$. Since $\alpha_{t-1} \geq 1$, we have $\alpha_t \geq 0$. Thus

$$
\begin{aligned}
\tau - t &= \alpha_t \\
&\geq |\alpha_t| + |\beta_t| - 1 &&\text{(Since } 0 < \beta_t \leq 1 \text{ and } \alpha_t \geq 0) \\
&= \rho_t - 1 > \rho_t - 2\,. &&(111)
\end{aligned}
$$

**(ii)** $\alpha_{t-1} \geq 1$, $\beta_{t-1} \geq 0$, **and** $\beta_t < 0$. Because $\alpha_{t-1} > 0$, Alice will cut at 1 in round $t$, so $\beta_t = \beta_{t-1} + 1$. But then $\beta_t > \beta_{t-1} \geq 0$, contradicting $\beta_t < 0$. Therefore, this case cannot happen.

**(iii)** $\alpha_{t-1} = 0$ **and** $\beta_{t-1} \geq 1$. By Lemma 11, Alice can cut wherever she likes, but Bob will pick L. Wherever Alice cuts, we will have $\alpha_t = -1$ and $\beta_t \geq 0$. In order to return to $\alpha = 0$, Bob must start picking R, but he cannot do so until $\beta \leq 0$ by Lemma 11. Therefore, the next axis-crossing round $\tau$ will be the one where $\beta_{\tau+1} < 0$ and $\beta_\tau \geq 0$. Until then, Alice will keep cutting at 0, so $\beta$ will decrease by 1 every round. Therefore, $\tau = t + \lfloor \beta_t \rfloor$, and this implies that

$$
\tau - t = \lfloor \beta_t \rfloor \leq |\alpha_t| + |\beta_t| = \rho_t\,.
$$

Again to show $\rho_t - 2 \leq \tau - t$, observe that

$$
\begin{aligned}
\tau - t &= \lfloor \beta_t \rfloor \\
&\geq |\alpha_t| - 1 + |\beta_t| - 1 &&\text{(Since } \alpha_t = -1 \text{ and } \beta_t \geq 0) \\
&= \rho_t - 2\,.
\end{aligned}
$$

**(iv)** $\alpha_{t-1} = 0$ **and** $0 \leq \beta_{t-1} < 1$. Under these constraints, we have $\rho_{t-1} = |\alpha_{t-1}| + |\beta_{t-1}| < 1$, so part (d) of Lemma 14 implies that $t - 1 = 0$. Therefore, we have $\alpha_{t-1} = \beta_{t-1} = 0$. Accounting for all possible choices Alice and Bob can make, it must be the case that $\alpha_t = \pm 1$ and $-1 \leq \beta_t \leq 1$, which implies that $1 \leq \rho_t \leq 2$. Thus it suffices to show that $\tau - t \leq 1$. We have a few cases:

- If $\alpha_t = 1$ and $\beta_t > 0$, then Bob will pick L in round $t + 1$ by Lemma 11. Therefore, $\alpha_{t+1} = 0$, so $\tau = t + 1$.
- If $\alpha_t = 1$ and $-1 < \beta_t \leq 0$, then Alice will cut at 1 in round $t + 1$ by Lemma 11. Therefore, $\beta_{t+1} > 0$, so round $t$ is axis-crossing and $\tau - t = 0$.
- If $\alpha_t = 1$ and $\beta_t = -1$, then Alice will cut at 1 and Bob will pick R in round $t + 1$ by Lemma 11. That will lead to $\alpha_{t+1} = 2$ and $\beta_{t+1} = 0$, so round $t + 1$ is axis-crossing. Therefore, $\tau - t = 1$.
- If $\alpha_t = -1$, we can reduce to the $\alpha_t = 1$ case by Lemma 13.

Thus for all types **(i)**-**(iv)**, it follows that $\rho_t - 2 \leq \tau - t \leq \rho_t$ as desired. $\qquad\square$

The next two lemmas show different conditions under which $\rho$ must increase. The first (Lemma 16) is more technical in nature, while the second (Lemma 17) is key to bounding the players' total payoff.

**Lemma 16.** *Suppose there exists a round $t$ such that $\beta_t \notin \mathbb{Z}$. Let $\tau > t$ be the first round after $t$ such that $\beta_t \beta_\tau \leq 0$, i.e., $\beta_\tau$ is zero or has the opposite sign of $\beta_t$. Then the following inequality holds:*

$$
\rho_\tau \geq \lfloor \rho_t \rfloor + 1\,.
$$

*Proof.* Again by Lemma 13, we can assume without loss of generality that $\beta_t \geq 0$. Since $\beta_t \notin \mathbb{Z}$, we can further assume $\beta_t > 0$.

Suppose that $\beta_\tau \notin \mathbb{Z}$. Because $\tau > t$ is the first round after $t$ with $\beta_\tau \leq 0$, we must have $\beta_{\tau-1} > 0$. Since $\beta_\tau \notin \mathbb{Z}$, we also have $\beta_\tau < 0$. Combining $|\beta_\tau - \beta_{\tau-1}| \leq 1$ and $\beta_\tau < 0$ yields $0 < \beta_{\tau-1} < 1$. Since $\beta_{\tau-1} > 0$, Bob picked L in round $\tau$ by Lemma 11, so we have $\alpha_\tau = \alpha_{\tau-1} - 1$. As $\beta_\tau < \beta_{\tau-1}$,

Alice must have not cut at 1 in round $\tau$ by Lemma 12. This implies that $\alpha_{\tau-1} \leq 0$. Finally, we obtain

$$
\begin{aligned}
\rho_\tau = |\alpha_\tau| + |\beta_\tau| &\geq |\alpha_{\tau-1} - 1| + 0 && \text{(Since } \alpha_\tau = \alpha_{\tau-1} - 1) \\
&= 1 - \alpha_{\tau-1} && \text{(Since } \alpha_{\tau-1} \leq 0) \\
&= 1 + |\alpha_{\tau-1}| + \lfloor |\beta_{\tau-1}| \rfloor && \text{(Since } \alpha_{\tau-1} \leq 0 \text{ and } 0 < \beta_{\tau-1} < 1) \\
&= 1 + \lfloor \rho_{\tau-1} \rfloor && \text{(Since } \alpha_{\tau-1} \in \mathbb{Z}) \\
&\geq 1 + \lfloor \rho_t \rfloor. && \text{(By Lemma 14)}
\end{aligned}
$$

On the other hand, suppose $\beta_\tau \in \mathbb{Z}$. By Lemma 14, we have $\rho_\tau \geq \rho_t$. Since $\beta_t \notin \mathbb{Z}$ but $\alpha_t \in \mathbb{Z}$ and $\alpha_\tau \in \mathbb{Z}$, we have $\rho_\tau \in \mathbb{Z}$ but $\rho_t \notin \mathbb{Z}$. Therefore, $\lfloor \rho_t \rfloor < \rho_t \leq \rho_\tau$. Since both $\lfloor \rho_t \rfloor \in \mathbb{Z}$ and $\rho_\tau \in \mathbb{Z}$, we have $\rho_\tau \geq \lfloor \rho_t \rfloor + 1$. $\qquad\square$

**Lemma 17.** *Let $t$ be a round in which Alice cuts at $a_t \in (0, 1)$. Let $\tau - 1$ be the first axis-crossing round strictly after $t - 1$. Then $\rho_\tau \geq \lfloor \rho_{t-1} \rfloor + 1$.*

*Proof.* By Lemma 11, Alice will cut at 0 or 1 if $\alpha_{t-1} \neq 0$. As Alice does not cut at 0 or 1 in round $t$, this implies that $\alpha_{t-1} = 0$. First, if $\beta_{t-1} = 0$, then Lemma 14 implies $t - 1 = 0$, so $\rho_\tau \geq 1 = \lfloor \rho_{t-1} \rfloor + 1$ as required.

Otherwise, by Lemma 13, we can assume without loss of generality that $\beta_{t-1} > 0$. Because $\rho_{t-1} \geq |\beta_{t-1}| > 0$, Lemma 14 implies $t - 1 \geq 1$. By part (d) of Lemma 14, we further have

$$
1 \leq \rho_{t-1} = |\alpha_{t-1}| + |\beta_{t-1}| = \beta_{t-1}.
$$

Since Alice does not cut at 0 in round $t$, we have $\beta_t > \beta_{t-1} - 1$ by Lemma 11, which implies that $\beta_t > 0$. As $\beta_{t-1} > 0$, Bob picked L in round $t$ by Lemma 11, so we have $\alpha_t < 0$. Again by Lemma 11, for $\alpha$ to return to 0, Bob would have to start picking R, but he won't until $\beta$ stops being positive. Therefore, the next axis-crossing round after $t$ will be one where $\beta$ crosses into being non-positive from positive, so $\beta_\tau < 0$.

Let $s > t - 1$ be the first round after $t - 1$ such that $\beta_s \leq 0$. Because $\beta_\tau < 0$, we have $s \leq \tau$. Because $\beta_t > 0$, we have $s > t$.

If $\beta_{t-1} \notin \mathbb{Z}$, then applying Lemma 16 to $t - 1$ yields $\rho_s \geq \lfloor \rho_{t-1} \rfloor + 1$. By Lemma 14, we have $\rho_\tau \geq \rho_s$, and so $\rho_\tau \geq \rho_s \geq \lfloor \rho_{t-1} \rfloor + 1$ as required.

If $\beta_t \notin \mathbb{Z}$, then applying Lemma 16 to $t$ yields $\rho_s \geq \lfloor \rho_t \rfloor + 1$. By Lemma 14, we have $\rho_\tau \geq \rho_s$ and $\rho_t \geq \rho_{t-1}$, and so $\rho_\tau \geq \rho_s \geq \lfloor \rho_t \rfloor + 1 \geq \lfloor \rho_{t-1} \rfloor + 1$ as required.

Else, we have $\beta_{t-1}, \beta_t \in \mathbb{Z}$. Since Alice did not cut at 0 or 1 in round $t$, by Lemma 12 we have $|\beta_t - \beta_{t-1}| < 1$. Since $\beta_t, \beta_{t-1} \in \mathbb{Z}$, we get $\beta_{t-1} = \beta_t$. Moreover, since $\alpha_{t-1} = 0$, we have $|\alpha_t| = |\alpha_{t-1} \pm 1| = 1$. This implies that

$$
\begin{aligned}
\rho_\tau &\geq \rho_t && \text{(By Lemma 14)} \\
&= \rho_{t-1} + 1 && \text{(Since } \beta_t = \beta_{t-1} \text{ and } |\alpha_t| = |\alpha_{t-1}| + 1) \\
&\geq \lfloor \rho_{t-1} \rfloor + 1.
\end{aligned}
$$

This completes the proof. $\qquad\square$

The following lemma bounds Bob's total payoff using the radius $\rho_t$.

**Lemma 18.** *For every round $t \geq 0$, the following inequalities hold:*

$$
-\rho_t \leq \sum_{i=1}^{t} (2u_B^i - 1) \leq \rho_t. \tag{112}
$$

*Proof.* We will first show the following stronger set of inequalities by induction on $t$:

$$
0 \leq |\alpha_t| + \sum_{i=1}^{t} (2u_B^i - 1) \leq \rho_t. \tag{113}
$$

As a base case, consider $t = 0$. Before anything has happened, all three sides of (113) are zero, so the inequalities trivially hold.

Now assume that (113) holds for some $t \geq 0$, and we will prove that it still holds for round $t + 1$. We consider three cases separately in what follows, depending on the values of $\alpha_t$ and $\beta_t$. Note that again by Lemma 13, it suffices to only consider cases where $\alpha_t \geq 0$.

- If $\alpha_t > 0$, Alice will cut at 1 in round $t + 1$ by Lemma 11. If Bob picks $L$, then he will receive a payoff of 1 and $\alpha_t$ will decrease by 1 due to Lemma 12. If Bob picks $R$, then he will receive a payoff of 0 and $\alpha_t$ will increase by 1 due to Lemma 12. In either case, the changes to $|\alpha_t| + \sum_{i=1}^{t}(2u_B(i) - 1)$ cancel out, which implies that

$$|\alpha_t| + \sum_{i=1}^{t}(2u_B^i - 1) = |\alpha_{t+1}| + \sum_{i=1}^{t+1}(2u_B^i - 1). \tag{114}$$

Further, by Lemma 14 we have $\rho_{t+1} \geq \rho_t$. Together with the induction hypothesis (113), this concludes

$$0 \leq |\alpha_{t+1}| + \sum_{i=1}^{t+1}(2u_B^i - 1) \leq \rho_{t+1},$$

and thus the induction holds for the first case.

- If $\alpha_t = 0$ and $\beta_t \geq 0$, then regardless of whether Bob picks L or R in round $t + 1$ we have $|\alpha_{t+1}| = |0 \pm 1| = 1$ as $\alpha$ necessarily changes by 1. Therefore, we have

$$\left(|\alpha_{t+1}| + \sum_{i=1}^{t+1}(2u_B^i - 1)\right) - \left(|\alpha_t| + \sum_{i=1}^{t}(2u_B^i - 1)\right) = 1 + 2u_B^{t+1} - 1$$

$$= 2u_B^{t+1} \tag{115}$$

Also, the change in $\rho$ can be bounded as follows:

$$
\begin{aligned}
\rho_{t+1} - \rho_t &= |\alpha_{t+1}| + |\beta_{t+1}| - |\alpha_t| - |\beta_t| \\
&= 1 + \left|\sum_{i=1}^{t+1}(2V_B([0, a_i]) - 1)\right| - 0 - \beta_t && (\beta_t \geq 0) \\
&= 1 + |(2V_B([0, a_{t+1}]) - 1) + \beta_t| - \beta_t \\
&\geq 1 + 2V_B([0, a_{t+1}]) - 1 && \text{(Removing the absolute value)} \\
&= 2V_B([0, a_{t+1}]) \tag{116}
\end{aligned}
$$

If Bob picked $L$ in round $t + 1$, this is exactly the same as (115). If Bob picked $R$, then by Lemma 11 and the assumption that $\beta_t \geq 0$ we must have $\beta_t = 0$. The change in the radius can be bounded as follows:

$$
\begin{aligned}
\rho_{t+1} - \rho_t &= |\alpha_{t+1}| + |\beta_{t+1}| - |\alpha_t| - |\beta_t| \\
&= 1 + |2V_B([0, a_{t+1}]) - 1| - 0 - 0 && (\beta_t = 0) \\
&= 1 + |1 - 2V_B([a_{t+1}, 1])| && (V_B([0, a_{t+1}]) + V_B([a_{t+1}, 1]) = 1) \\
&= 1 + |-(1 - 2V_B([a_{t+1}, 1]))| \\
&\geq 1 - 1 + 2V_B([a_{t+1}, 1]) && \text{(Removing the absolute value)} \\
&= 2u_B^{t+1} && \text{(Since Bob picked } R\text{)}
\end{aligned}
$$

In either case, the radius increased by at least as much as the middle of (113). More precisely, by the induction hypothesis (113), we obtain

$$
\begin{aligned}
|\alpha_{t+1}| + \sum_{i=1}^{t}(2u_B^i - 1) &= |\alpha_t| + \sum_{i=1}^{t}(2u_B^i - 1) + 2u_B^{t+1} - 1 + |\alpha_{t+1}| - |\alpha_t| \\
&\geq 0 + 2u_B^{t+1} && (\alpha_t = 0 \text{ and } |\alpha_{t+1}| = 1) \\
&\geq 0. && (u_B^{t+1} \geq 0)
\end{aligned}
$$

Note further that

$$|\alpha_{t+1}| + \sum_{i=1}^{t}(2u_B^i - 1) = |\alpha_t| + \sum_{i=1}^{t}(2u_B^i - 1) + 2u_B^{t+1} - 1 + |\alpha_{t+1}| - |\alpha_t|$$

$$\leq \rho_t + 2u_B^{t+1} \qquad\qquad \text{(By the induction hypothesis)}$$

$$= \rho_{t+1}. \qquad\qquad (\rho_{t+1} - \rho_t = 2u_B^{t+1})$$

This finishes the proof of the induction for the second case.

- If $\alpha_t = 0$ and $\beta_t < 0$, then by Lemma 13 we can consider $\alpha_t = 0$ and $\beta_t > 0$ instead, which has already been shown above.

In all cases, the inductive step holds. Therefore, by induction principle, (113) holds for all $t$. Subtracting $|\alpha_t|$ from all three sides of it, we obtain

$$-|\alpha_t| \leq \sum_{i=1}^{t}(2u_B^i - 1) \leq |\beta_t|.$$

Since $\rho_t = |\alpha_t| + |\beta_t|$, this immediately implies the desired bounds. $\qquad\square$

Now that we have bounds on the relevant events in terms of $\alpha$, $\beta$, and $\rho$, we can bound them as functions of $T$ to obtain our final result.

**Lemma 19.** *Let $n \geq 7$. Let $t_1, t_2, \ldots, t_n$ be a sequence of rounds such that for all $i \in [n-1]$ the following inequality holds:*

$$\rho_{t_{i+1}} \geq \lfloor \rho_{t_i} \rfloor + 1. \tag{117}$$

*Then $t_n - t_1 > \frac{1}{10}n^2$.*

*Proof.* First, we will show by induction that, for $i \in [n]$, we have $\rho_{t_i} \geq i - 1$. For the base case, $\rho_{t_1} \geq \rho_0 = 0$, so the inequality trivially holds.

Now assume $\rho_{t_i} \geq i - 1$ for some $i \in [n-1]$ and we will prove that the induction step holds for the case $i + 1$. By (117), observe that

$$\rho_{t_{i+1}} \geq \lfloor \rho_{t_i} \rfloor + 1$$
$$\geq \lfloor i - 1 \rfloor + 1$$
$$= (i + 1) - 1.$$

Thus, by the induction principle we have

$$\rho_{t_i} \geq i - 1 \text{ for } i = 1, \ldots, n \tag{118}$$

Now consider an arbitrary $i \in [n-1]$. Note that $\rho_{t_{i+1}} \geq \lfloor \rho_{t_i} \rfloor + 1$ implies $\rho_{t_{i+1}} > \rho_{t_i}$. Therefore, by Lemma 14, there must be an axis-crossing round $c_i$ among the interval $[t_i, t_{i+1})$. This is because if this is not true, we have $\rho_{t_{i+1}} = \rho_{t_i}$ which contradicts $\rho_{t_{i+1}} > \rho_{t_i}$. Repeating the same argument for each $i$, we conclude that there exists a sequence of rounds $c_1, c_2, \ldots, c_{n-1}$ each of which is axis-crossing, and satisfies the following inequality for every $i \in [n-1]$:

$$t_i \leq c_i < t_{i+1} \tag{119}$$

In addition, for any $i \in [n-1]$, we observe that

$$c_{i+1} - (c_i + 1) \geq \rho_{c_i+1} - 2 \qquad \text{(By Lemma 15, with } \tau = c_{i+1} \text{ and } t = c_i + 1)$$
$$\geq \rho_{t_i} - 2 \qquad\qquad\qquad\qquad\qquad \text{(By (119))}$$
$$\geq i - 3 \qquad\qquad\qquad\qquad\qquad\quad \text{(By (118))}$$

Slightly rearranging, we obtain

$$c_{i+1} - c_i \geq i - 2 \ \ \forall i \in [n-1]. \tag{120}$$

Combining (119) and (120), we finally obtain

$$t_n - t_1 > c_{n-1} - c_1 \qquad \text{(By (119))}$$
$$= \sum_{i=1}^{n-2}(c_{i+1} - c_i)$$
$$\geq \sum_{i=1}^{n-2}(i - 2) \qquad \text{(By (120))}$$
$$= \frac{1}{2}n^2 - \frac{7}{2}n + 5.$$

For $n \geq 7$, we have $\frac{1}{2}n^2 - \frac{7}{2}n + 5 \geq \frac{1}{10}n^2$,[4] and so $t_n - t_1 > n^2/10$, as required. $\qquad \square$

The following lemma will finally be combined with Lemma 18 to obtain the desired bound for Bob's payoff.

**Lemma 20.** *For $T \geq 5$, the final radius $\rho_T$ satisfies the following:*

$$\rho_T \leq 2\sqrt{10T}.$$

*Proof.* Let $t_1 = 0$, and for $i \geq 2$ recursively define $t_i$ be the first round after $t_{i-1}$ satisfying $\rho_{t_i} \geq \lfloor \rho_{t_{i-1}} \rfloor + 1$. Let $n$ be the last index of such $t_i$ given the time horizon $T$.

As an immediate corollary of Lemma 19, we have

$$n \leq \max\{7, \sqrt{10T}\}.$$

For $T \geq 5$, this implies that

$$n \leq \sqrt{10T} \qquad (121)$$

By Lemma 14, we also know that $\rho$ can increase by at most 2 per round. Furthermore, since $t_n$ is the last element in the sequence $\{t_i\}_{i \in [n]}$, we have $\rho_T < \lfloor \rho_{t_n} \rfloor + 1$. Relaxing this slightly, we have

$$\rho_T \leq \rho_{t_n} + 2 \qquad (122)$$

Therefore, we obtain the following inequalities:

$$\rho_T \leq 2 + \rho_{t_n} \qquad \text{(By (122))}$$
$$= 2 + \sum_{i=1}^{n-1}(\rho_{i+1} - \rho_i)$$
$$\leq 2 + 2(n-1) \qquad \text{(By Lemma 14)}$$
$$= 2n \qquad (123)$$

Putting (121) and (123) together, we obtain

$$\rho_T \leq 2\sqrt{10T},$$

which finishes the proof of the lemma. $\qquad \square$

Using the above bound on the radius together with Lemma 18, Bob's payoff can now be bounded as follows.

**Lemma 21.** *For $T \geq 5$, Bob's total payoff satisfies:*

$$\frac{T}{2} - \sqrt{10T} \leq \sum_{t=1}^{T} u_B^t \leq \frac{T}{2} + \sqrt{10T}.$$

---

[4]We omit the elementary calculus.

*Proof.* We start from the inequality (112). Halving all three sides of (112) for $t = T$ and adding $T/2$, we obtain

$$\frac{T}{2} - \frac{1}{2}\rho_T \leq \sum_{t=1}^{T} u_B^t \leq \frac{T}{2} + \frac{1}{2}\rho_T.$$

Using the upper bound $\rho_T \leq 2\sqrt{10T}$ from Lemma 20 gives the desired bounds. $\square$

Combined with Lemma 21, Alice's payoff can eventually be bounded using the following lemma, which effectively bounds the summation of Alice and Bob's total payoff.

**Lemma 22.** *For $T \geq 5$, the summation of total payoff to Alice and Bob satisfies the following:*

$$T - \sqrt{10T} \leq \sum_{t=1}^{T} \left( u_A^t + u_B^t \right) \leq T + \sqrt{10T}.$$

*Proof.* Given the time horizon $T$, let $s_1, s_2, \ldots, s_k$ be the rounds in which Alice cuts at a point other than 0 or 1, where $k$ denotes the number of such rounds. These are the only rounds in which the total payoff $u_A^t + u_B^t$ is not necessarily 1. The total payoff in these rounds can be bounded as

$$0 \leq u_A^{s_i} + u_B^{s_i} \leq 2,$$

for any $i \in [k]$. Thus, we obtain

$$T - k \leq \sum_{t=1}^{T} u_A^t + u_B^t \leq T + k.$$

Therefore, it suffices to prove that $k \leq \sqrt{10T}$.

If $k < 7$, the proof follows as $k < 7 \leq \sqrt{10T}$.

Otherwise, we have $k \geq 7$. For each $s_i$, let $\tau_i$ be the round after the next axis-crossing round after $s_i - 1$. Consider $s_i$ for an arbitrary $i \in [k]$. Due to Lemma 11, if $\alpha_{s_i-1} \neq 0$ then Alice cuts at 0 or 1 in round $s_i$, which contradicts our definition of round $s_i$. Thus we have $\alpha_{s_i-1} = 0$. This argument holds for arbitrary $i \in [k]$, so both $s_i - 1$ and $s_{i+1} - 1$ are axis-crossing rounds. Moreover, since both $\alpha_{s_i-1} = 0$ and $\alpha_{s_{i+1}-1} = 0$, there must have been some rounds between $s_i - 1$ and $s_{i+1} - 1$ where Bob picked $L$ and some where he picked $R$. Therefore, $\beta$ must have changed sign at least once, so there is another axis-crossing round in between $s_i - 1$ and $s_{i+1} - 1$ where $\beta$ changed sign. This implies that for every $i = 1, \ldots, k - 1$, the next axis crossing round $\tau_i$ after $s_i - 1$ should exist at least before $s_{i+1} - 1$, *i.e.,*

$$\tau_i \leq s_{i+1} - 1. \tag{124}$$

Due to the monotonicity of $\rho$ by Lemma 14, for any $i \in \{1, \ldots, k - 1\}$ we have

$$\rho_{s_{i+1}-1} \geq \rho_{\tau_i} \qquad \text{(By (124))}$$
$$\geq \lfloor \rho_{s_i-1} \rfloor + 1. \qquad \text{(By Lemma 17)}$$

By Lemma 19, we have:

$$(s_k - 1) - (s_1 - 1) > \frac{1}{10}k^2. \tag{125}$$

Re-arranging (125) and using the fact that $s_k - s_1 \leq T$, we obtain $k < \sqrt{10T}$. This finishes the proof of the case $k \geq 7$, which completes the lemma. $\square$

