# OpenReview forum: "Dueling over Dessert, Mastering the Art of Repeated Cake Cutting"
_NeurIPS.cc/2024/Conference — NeurIPS 2024 poster_

### Official Review · Reviewer_17dK · 2024-07-07

**Soundness:** 4
**Presentation:** 4
**Contribution:** 4
**Rating:** 8
**Confidence:** 3

**Summary:**

This paper studies the game of repeated cake cutting. The cake is modeled as the unit interval $[0,1]$, and in each round $t \in \\{1,\ldots,T\\}$, player $A$ chooses a point $a_t$ to cut the cake into two pieces $[0,a_t]$ and $(a_t, 1]$, and player $B$ chooses one of the two pieces, either after observing $a_t$ (the sequential setting) or without observing $a_t$ (the simultaneous setting). The paper proves some properties of this game.

**Strengths:**

At a high level, this is a very well-written paper of theoretical nature that is easy to understand. The literature review is exhaustive. It properly defines and proves all the claims. The results proved in the paper seem new to me, and the proofs are elegant. The setting studied is interesting and will be useful to the community.

**Weaknesses:**

I do not find any obvious weaknesses in this paper. It a paper of theoretical nature and requiring experiments would be meaningless.

**Questions:**

1. Could valuations be defined more generally as countably additive, instead of just finitely additive? This could allow the analysis to be applicable to more general games where player $A$ could divide up the cake into any two disjoint measurable sets.
2. Why is equitable allocation called "fair"? I could imagine allocation $(Z_A, Z_B)$ to players such that $V_A(Z_A) = 1/3 = V_B(Z_B)$ and thus is equitable, but allocation $(Z_B, Z_A)$ which has $V_A(Z_B) = 1- V_A(Z_A) = 2/3 = 1 - V_B(Z_B) = V_B(Z_A)$ and thus is also equitable and "fairer". Should fairness be defined as $V_A(Z_A),V_B(Z_B) \ge 1/2$?

---

> ### Author Rebuttal · Authors · 2024-08-06
>
> Thank you for your review.
>
> For question 2: Equitability requires that each player gets the same value. As you pointed out, not all equitable allocations are equally good. When $V_A(Z_A) \geq 1/2$ and $V_B(Z_B) \geq 1/2$, the allocation is also proportional. Proportionality is a fairness notion that requires that when there are two players, each player gets utility at least $1/2$. Thus in the example you mention, one allocation is equitable but not proportional, and the other one is both equitable and proportional. We will clarify this point.
>
> For question 1: Yes, the valuation of a player for a measurable set $S$ is generally defined as the integral of that player’s value density function over $S$. We will clarify this.
>
> Regarding the direction of Alice dividing the cake using more than one cut, we include an example that we also wrote for the third reviewer (xPAG).
> Suppose Alice can cut at two points $a_{t,1}$ and $a_{t,2}$ each day $t$ (placing each of the resulting three pieces in one of two bins labeled L and R, respectively). Suppose Bob is myopic, choosing his favorite bin each day.
>
> Then Alice can
>
> (1) discretize the cake to an $\epsilon$ grid according to her valuation, and then
>
> (2) sample Bob’s behavior at each choice of cut points on the $\epsilon$-grid.
>
> Steps (1-2) can be done in $O(1/\epsilon \cdot \log(1/\epsilon))$ rounds by checking every possibility for the first cut point on the $\epsilon$-grid and using binary search for the second.
> Once Alice finds the best pair of cut points on the $\epsilon$-grid (from her perspective), she can cut there for the remainder of time and get within $O(\epsilon)$ of her Stackelberg value (defined with respect to two cuts) for the rest of the rounds.
> Setting $\epsilon \approx 1/\sqrt{T}$ gives $O(\sqrt{T} \cdot \log(T))$ regret.
>
> For $n$ cuts, a similar strategy can achieve $O(T^{1-1/n} \cdot \log(T))$ regret.
> Making such intuition precise and extending the analysis to a sublinear-regret Bob are interesting directions for future work.
>
> Allowing Alice to divide the cake into any two measurable sets is also interesting, though our ideas here are much more speculative. Perhaps Alice can still discretize the cake precisely enough to approximate optimal divisions, or maybe a Bob with a sufficiently pathological valuation function could never be completely exploited.

---

> > ### Comment · Reviewer_17dK · 2024-08-11
> >
> > Thank you for the response. I will keep my high score for the paper.

---

### Official Review · Reviewer_xPAG · 2024-07-11

**Soundness:** 3
**Presentation:** 3
**Contribution:** 3
**Rating:** 7
**Confidence:** 1

**Summary:**

The paper deal with a repeated division problem where at each round a new cake (modeled as the interval $[0,1]$), identical to previous ones, arrives. Alice acts first and cuts the cake in two parts. Then, Bob chooses the piece he prefers, leaving the remainder for Alice. Alice (resp. Bob) valuation preferences are determined by the integral over her (resp. his) *private* valuation density function over the piece of cake she (resp. he) receives.
The authors analyze two versions of this game:
1) sequential, where Bob sees Alice’s cut before choosing
2) simultaneous, where Bob chooses without seeing Alice’s cut
Interestingly, in the sequential setting, the authors shows that if Bob chooses his favorite piece in (a nearly) myopic way, then Alice can exploit this greedy tendency to build a strategy that has sublinear Stackelberg regret. Furthermore, they show how both players can devise strategies that force the other player into a dynamic yielding equitable outcomes for both.

**Strengths:**

The writing style of the paper is rigorous, with a clear introduction to the problem and related literature.
The relevant definitions and claims are clearly presented in a mathematical fashion.
The sequential version of the problem is especially interesting from practical, modeling, mathematical, and philosophical points of view. In particular, it is very insightful that Bob's greedy behavior leads to undesirable outcomes that primarily damage himself, yet both players have a way to enforce fairness.

**Weaknesses:**

The analysis is limited to the case where the interval $[0,1]$ is split into two sub-intervals $[0,a]$ and $(a,1]$, for any $a \in [0,1]$ of Alice's choice. This simplification may restrict the applicability of the results to more complex real-world scenarios where resources might need to be divided into more than two parts.

**Questions:**

What if Alice can select not just a point, but say a pluri-interval and its complement (maybe with a bounded number of pieces known in advance). What if Alice can split the set $[0,1]$ in any measurable set and its complement? Does the problem turn out to be utterly intractable in these cases?

**Limitations:**

The authors correctly states the assumptions under which their results hold.

---

> ### Author Rebuttal · Authors · 2024-08-06
>
> Thank you for your review. The question you raise about more general cutting models is interesting and likely to be tractable for many of them.
>
> For example, suppose Alice can cut at two points $a_{t,1}$ and $a_{t,2}$ each day $t$ (placing each of the resulting three pieces in one of two bins labeled L and R, respectively). Suppose Bob is myopic, choosing his favorite bin each day.
>
> Then Alice can
>
> - (1) discretize the cake to an $\epsilon$ grid according to her valuation, and then
>
> - (2) sample Bob’s behavior at each choice of cut points on the $\epsilon$-grid.
>
> Steps (1-2) can be done in $O(1/\epsilon \cdot \log(1/\epsilon))$ rounds by checking every possibility for the first cut point on the $\epsilon$-grid and doing binary search for the second.
> Once Alice finds the best pair of cut points on the $\epsilon$-grid (from her perspective), she can cut there for the remainder of time and get within $O(\epsilon)$ of her Stackelberg value (defined with respect to two cuts) for the rest of the rounds.
> Setting $\epsilon \approx 1/\sqrt{T}$ gives $O(\sqrt{T} \cdot \log(T))$ regret.
>
> For $n$ cuts, a similar strategy can achieve $O(T^{1-1/n} \cdot \log(T))$ regret.
> Making such intuition precise and extending the analysis to a sublinear-regret Bob are interesting directions for future work.
>
> Allowing Alice to divide the cake into any two measurable sets is also interesting, though our ideas here are much more speculative. Perhaps Alice can still discretize the cake precisely enough to approximate optimal divisions, or maybe a Bob with a sufficiently pathological valuation function could never be completely exploited.

---

### Official Review · Reviewer_6j8m · 2024-07-12

**Soundness:** 4
**Presentation:** 4
**Contribution:** 3
**Rating:** 7
**Confidence:** 4

**Summary:**

The paper considers a problem of sequential cake cutting. Each day for $T$ days, 2 players, Alice and Bob, must divide the cake. The cake is a unit interval $[0, 1]$ which they each value with some density function that in total adds up to $1$. They have the same preferences across days. The first player, Alice, makes a single cut into two contiguous pieces. Bob chooses left or right. They consider both when Bob first sees the cut and when they act simultaneously.

Ideally, Alice knows Bob's midpoint and can cut right there, and Bob, indifferent between the two pieces, will take the one less valuable for Alice (assuming reasonable tie-breaking). This is the maximum value she can guarantee and is called her Stackelberg value. The paper measures Bob and Alice's performance by regret. Alice's by how far she is off from achieving her Stackelberg at each step, and Bob's from how far off he is from picking the best of the two cuts at each step.

First, they show that if Bob always picks the more valuable piece (so has regret 0), then Alice can exploit achieving $O(\log T)$ regret. If Bob is close to this by having regret $O(T^\alpha)$ for $\alpha < 1$, if Alice knows $\alpha$, she can get regret $O(T^{\frac{\alpha + 1}{2}} \log T)$, and otherwise, she can get regret $O(T/\log T)$. Neither of these can be improved by polynomial factors.

On top of these, Alice has a strategy such that no matter what Bob plus, she gets on average $1/2 - o(1)$ and Bob gets $1/2 + o(1)$, and Bob has a strategy such that no matter what Alice plays, he gets $1/2 - o(1)$ and Alice gets $1/2 + o(1)$. If they play simultaneously, Bob also has a randomized strategy so that they both get $1/2$ in expectation (just picking left and right uniformly at random).

Finally, assuming players best respond assuming the other player will pick a random strategy they have played so far, both players will have average payoff approaching 1/2 at a rate of $O(1/\sqrt{T})$

**Strengths:**

- The model is simple but satisfying
- The results are interesting and presented nicely

**Weaknesses:**

Although the model is very cute, it's unclear how realistic it is for agents to be playing this exact same game against each other repeatedly.

Also, there is no field for smaller comments, but you may consider citing "Playing Divide-and-Choose Given Uncertain Preferences" from EC'23 which seems relevant to the game-theoretic aspects of cut and choose in cake cutting

**Questions:**

Do you have some real-world motivation where such repeated games with the same players/valuations make sense?

**Limitations:**

Yes

---

> ### Author Rebuttal · Authors · 2024-08-06
>
> Thank you for your review.
>
> Regarding playing the game repeatedly, a high level scenario is where the salespeople of a roofing company are paid by commission for solar panel installation and maintenance services. Each day, they might divide areas of town among themselves for door-to-door sales. Due to local connections or varying tactics, different salespeople would have different expected profits in different areas.
> More generally, one can consider daily task allocation in businesses or recurring distribution of computational resources.
>
> Thank you for bringing the EC 23 paper on cut-and-choose in a Bayesian setting to our attention. We had seen the paper and thought we had included it, but accidentally forgot to; we will certainly cite and discuss it in the final version.

---

> > ### Comment · Reviewer_6j8m · 2024-08-07
> >
> > Thank you. That is a reasonable example. I leave my original score unchanged.

---

### Official Review · Reviewer_zKTZ · 2024-07-13

**Soundness:** 4
**Presentation:** 3
**Contribution:** 4
**Rating:** 7
**Confidence:** 4

**Summary:**

This paper considers the problem of repeated cake cutting among two agents. In this problem the same cake appears at each round and Alice cuts the cake based on her utility function over the cake. Bob has to choose one of the two parts after seeing the cut.

The authors show that if Bob almost always chooses his preferred piece then Alice can learn his utility function and exploit him. Then they show that in both the sequential and simultaneous settings Alice has an strategy which gives her an average payoff of $1/2-o(1)$. They add a similar bound for Bob in the sequential setting.

At the end they analyze the fictitious play in this setting and show that the convergence rate is $O(1/\sqrt T)$.

**Strengths:**

The paper considers an interesting classic problem and takes it to the next level. The problem is natural and well motivated. The results are strong and use wide range of techniques. It is well written and easy to follow. They consider a nice problem and draw a complete picture.

**Weaknesses:**

My main concern is that there are too many results in this paper and basically the body just gives and overview of the paper. Due to the page limit they had to shove everything to the appendix.
The related work section is too long. You can include a shorter version in the main body and an extended version in the appendix. This gives you space to bring the preliminaries to the body.

This is a solid piece of work but I'm not sure if NeurIPS is the right venue for such papers.

**Questions:**

If A and B take turns in cutting the cake, can we see it as two separate instances of this problem?

**Limitations:**

-

---

> ### Author Rebuttal · Authors · 2024-08-06
>
> Thank you for your review. We will bring the preliminaries in the main body of the paper and shorten related work as necessary to make this possible.
>
> If Alice and Bob take turns cutting and choosing, the feedback model is richer. Alice observes how Bob behaves (1) as a chooser, thus learning about his preferences observing him in this role, but also (2) as a cutter, which gives her additional information about his preferences through a different feedback model.
>
> For this reason, we believe the problem you mention is not separable (i.e. it’s not equivalent to running two separate instances of the problem). Formalizing this intuition would take additional analysis; we will mention the question in future work, thank you for suggesting it.

---

### Decision · Program_Chairs · 2024-09-25

**Decision:**

Accept (poster)

**Comment:**

This is a high quality algorithmic game theory paper that was truly appreciated by the reviewers. While the area chair had some doubts about its relevance to the conference itself, the reviewers defended the paper vigorously and convinced AC that the paper deserves to be accetped.